# Offline Model-based Optimization for Real-World Molecular Discovery

**Dong-Hee Shin** [1] [*]  **Young-Han Son** [1] [*]  **Hyun Jung Lee** [1]  **Deok-Joong Lee** [1]  **Tae-Eui Kam** [1]

## Abstract

Molecular discovery has attracted significant attention in scientific fields for its ability to generate novel molecules with desirable properties. Although numerous methods have been developed to tackle this problem, most rely on an online setting that requires repeated online evaluation of candidate molecules using the oracle. However, in real-world molecular discovery, the oracle is often represented by wet lab experiments, making this online setting impractical due to the significant time and resource demands. To fill this gap, we propose the *Molecular Stitching (MolStitch)* framework, which utilizes a fixed offline dataset to explore and optimize molecules without the need for repeated oracle evaluations. Specifically, MolStitch leverages existing molecules from the offline dataset to generate novel 'stitched molecules' that combine their desirable properties. These stitched molecules are then used as training samples to fine-tune the generative model using preference optimization techniques. Experimental results on various offline multi-objective molecular optimization problems validate the effectiveness of MolStitch. The source code is available online.

## 1. Introduction

In recent years, a diverse array of *in silico* generative models has been developed to tackle molecular discovery (Bilodeau et al., 2022). These computational approaches have demonstrated impressive success across various benchmarks, leading to a growing interest in integrating them into real-world applications such as drug discovery. Despite this success, most existing *in silico* generative models often operate under an online optimization setting, where numerous candidate molecules are generated iteratively and those molecules are evaluated immediately using the oracle function. However,

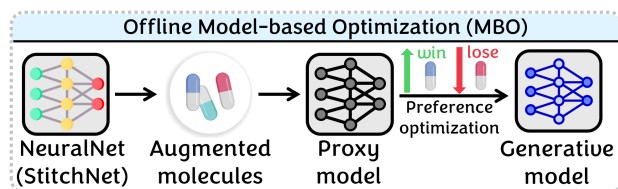

*Figure 1.* Proposed offline MBO process for molecular discovery.

in real-world molecular discovery, the oracle function is typically represented by wet lab experiments, which are resource-intensive and can take weeks or even months for evaluation (Payton et al., 2023). This creates a significant bottleneck, as *in silico* models cannot receive real-time evaluation feedback from wet lab. Instead, these models must wait for the wet lab experiments to complete before further optimization can occur, leading to substantial delays.

To address these challenges, a promising research direction is to enable the optimization and refinement of the *in silico* generative model without relying on online evaluation feedback. To this end, we propose to explore an offline optimization setting for real-world molecular discovery. Specifically, offline optimization seeks to fully leverage the information contained within an offline dataset, using it to refine the generative model in the absence of online evaluation feedback. Detailed explanations of an offline setting is in Appendix A.

One of the most promising approaches for solving the offline optimization problem is offline model-based optimization (MBO) (Trabucco et al., 2022). In this approach, a proxy, typically parameterized as a deep neural network $\hat{f}_\theta(\cdot)$, is trained to approximate the oracle function by fitting it to an offline dataset. Once trained, the proxy acts as a surrogate to guide the optimization of the generative model. For instance, gradient ascent (Zinkevich, 2003) can be applied to the generative model's parameters based on the proxy's predictions, refining the generative model to produce candidate molecules with increasingly desirable properties.

The offline MBO approach exhibits strong performance by generating synthetic data guided by the proxy. However, several challenges remain: the vanilla proxy is trained using a supervised regression loss, which may struggle to accurately approximate the true values from the oracle function as the problem becomes more complex (Fu & Levine, 2021). This challenge is compounded when the proxy encounters out-of-distribution (OOD) data, leading to significant discrepancies

---
[*]Equal contribution  [1]Department of Artificial Intelligence, Korea University, Seoul, Republic of Korea. Correspondence to: Tae-Eui Kam <kamte@korea.ac.kr>.

*Proceedings of the 42nd International Conference on Machine Learning*, Vancouver, Canada. PMLR 267, 2025. Copyright 2025 by the author(s).

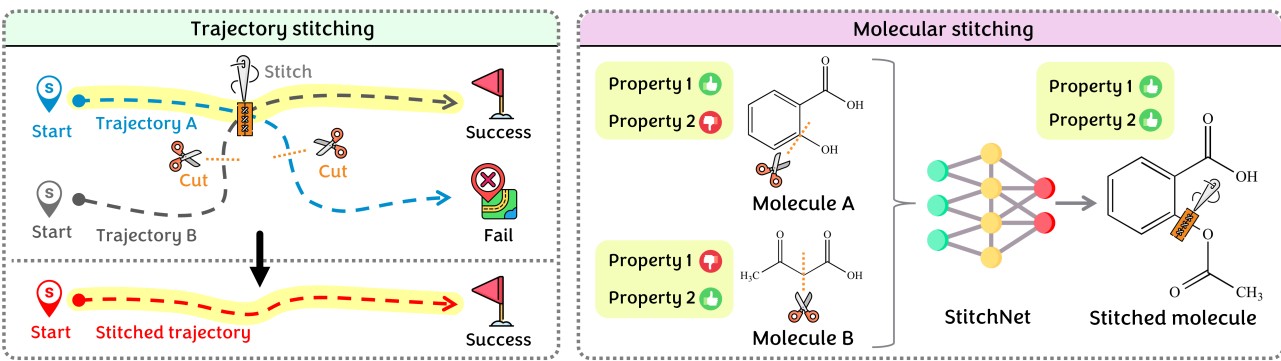

*Figure 2.* An illustration of trajectory stitching in reinforcement learning (left) and molecular stitching in molecular discovery (right).

between the true values and the proxy's predictions (Qi et al., 2022). To tackle these issues, recent studies have proposed various strategies to enhance the robustness and accuracy of the proxy such as introducing conservative estimates (Trabucco et al., 2021), employing a local smoothness (Yu et al., 2021), and adopting ensemble methods (Chen et al., 2023a).

While these advanced offline MBO methods significantly enhance the proxy, they may not fully exploit the valuable information inherent in the offline dataset, as this data is typically used solely for training the proxy. In the field of offline reinforcement learning (RL), researchers have introduced trajectory stitching techniques (Li et al., 2024; Kim et al., 2024b) to directly leverage the existing offline data by creating synthetic trajectories through segment combination. As depicted in Figure 2, consider two distinct trajectories in the offline dataset: trajectory A has a strong start but ends at the wrong destination, whereas trajectory B starts poorly yet successfully reaches the goal. By applying trajectory stitching, these trajectories can be merged to form a new stitched trajectory that combines the strong start of trajectory A with the successful goal achievement of trajectory B.

In this paper, we propose the Molecular Stitching (MolStitch) framework that tackles the offline molecular optimization problem. Drawing inspiration from trajectory stitching, our framework involves stitching molecules from the offline dataset. For instance, if molecule A possesses desirable property 1 but lacks property 2, while molecule B has the opposite characteristics, we aim to 'stitch' these molecules together to produce a new stitched molecule that exhibits both desirable properties. In other words, our framework utilizes the existing molecules in the offline dataset to generate novel stitched molecules, allowing the generative model to learn from these newly synthesized data samples.

To leverage stitched molecules as augmented synthetic data, they must be effectively evaluated to provide constructive feedback to the generative model. However, this evaluation process poses a challenge, as these molecules are unfamiliar to the proxy. To address this, we reformulate the proxy's task from property score regression to pairwise classification.

In particular, we develop a rank-based proxy that learns the ranking relationship between two molecules based on desired properties and classifies which molecule is more favorable. This transformation simplifies the task for the proxy, thereby enabling it to provide more reliable feedback.

Beyond the offline setting, real-world molecular discovery also requires the optimization of multiple molecular objectives (properties). For example, a successful drug must satisfy several criteria—it must be bioactive, safe, synthesizable, and more—rather than excelling in just one aspect (Fromer & Coley, 2023). This challenge is referred to as a multi-objective molecular optimization (MOMO) problem. A prevalent approach to tackling MOMO is scalarization, where multiple objectives are combined into a single objective by assigning weights that reflect their relative importance (Gunantara, 2018). However, in the offline setting, the exact importance of each objective is often unknown, and adjusting weights based on immediate feedback is limited (Xue et al., 2024). To tackle this, we incorporate priority sampling using a Dirichlet distribution (Minka, 2000) into our framework. Specifically, instead of manually selecting weights, we employ priority sampling to generate a variety of weight configurations during the molecular stitching process, resulting in a diverse set of stitched molecules.

The main contributions of our framework are outlined as:

- We investigate two critical aspects for real-world molecular discovery: *offline molecular optimization* and *multi-objective molecular optimization (MOMO)*. To this end, we propose MolStitch, the novel framework specifically designed to solve the offline MOMO problem.

- MolStitch includes novel components: *StitchNet* for utilizing the offline dataset to generate 'stitched molecules', *Rank-based proxy* for evaluating molecules, and *preference optimization* to fine-tune the generative model.

- We introduce *priority sampling* using a Dirichlet distribution to efficiently generate diverse weight configurations. This allows for effective exploration of trade-offs among multiple objectives in the offline MOMO problem.

## 2. Preliminaries

**Multi-objective optimization.** Let $\mathcal{M}$ denote the space of all possible molecules $m$, and let $f_1, f_2, \ldots, f_k : \mathcal{M} \to \mathbb{R}$ be $k$ real-valued molecular objective functions, each representing a molecular property to be optimized. The multi-objective molecular optimization problem can be stated as:

$$\underset{m \in \mathcal{M}}{\text{Maximize}} \quad \mathbf{F}(m) = \{f_1(m), f_2(m), \ldots, f_k(m)\}. \quad (1)$$

In this problem, it is challenging to identify and generate a single molecule that simultaneously maximizes all objective functions. This challenge arises because improving one molecular property may lead to the deterioration of other properties due to inherent trade-offs between them (Fromer & Coley, 2023). Thus, the goal of this problem is to generate a diverse set of molecules that reside on the Pareto front.

**Definition 2.1** (**Pareto front**). The Pareto front, denoted as $\mathbf{PF}$, is the set of all Pareto optimal molecules $m^*$ in the objective space. Mathematically, it can be expressed as:

$$\mathbf{PF} = \{\mathbf{F}(m^*) \mid m^* \in \mathcal{PS}\}, \quad (2)$$

where $\mathcal{PS}$ is the Pareto set that contains all $m^*$, defined as:

$$\mathcal{PS} = \big\{ m^* \in \mathcal{M} \mid \nexists m \in \mathcal{M} : \mathbf{F}(m) \succeq \mathbf{F}(m^*) \\ \wedge \mathbf{F}(m) \neq \mathbf{F}(m^*) \big\}. \quad (3)$$

**Generative model.** Let $G_\phi$ denote a generative model that produces molecules in an auto-regressive manner. The generation process for a molecule $m$ of total length $T$ is defined:

$$G_\phi(m) = \prod_{t=1}^{T} G_\phi(m^t | m^{t-1}, m^{t-2}, \ldots, m^1), \quad (4)$$

where $m^t$ represents the $t$-th component (or token) in the sequence that constitutes the molecule $m$.

**Offline optimization.** Let $\mathcal{D} = \{(m_n, \mathbf{F}(m_n))\}_{n=1}^{N}$ be the offline dataset, where $m_n \in \mathcal{M}$ represents a pre-collected molecule and $\mathbf{F}(m_n)$ represents the corresponding true molecular objective scores. The goal of this offline molecular optimization is to generate new molecules that potentially outperform the best-known Pareto optimal molecules in $\mathcal{D}$. To explore molecular space beyond the dataset $\mathcal{D}$, a common strategy involves constructing a proxy, $\hat{f}_\theta(\cdot) : \mathcal{M} \to \mathbb{R}$, to evaluate molecules. The most direct approach is the vanilla proxy, which approximates the scores of true objective functions $\mathbf{F}(\cdot)$. The vanilla proxy is trained to minimize the discrepancy between the predicted and true objective scores:

$$\theta^* = \arg\min_\theta \frac{1}{N} \sum_{n=1}^{N} \left\| \hat{f}_\theta(m_n) - \mathbf{F}(m_n) \right\|^2. \quad (5)$$

To optimize the generative model $G_\phi$ such that it produces molecules with improved objective scores, the vanilla proxy

can be implemented. Specifically, the $G_\phi$ can be updated by maximizing the expected performance of the generated molecules based on the vanilla proxy's predictions such as:

$$\phi^* = \arg\max_\phi \mathbb{E}_{m \sim G(\phi)} \big[ \hat{f}_\theta(m) \big]. \quad (6)$$

However, this approach may face challenges as the complexity of the problem increases. The vanilla proxy might produce unreliable predictions when encountering molecules outside its training data distribution, leading to potentially misguided optimization. Moreover, this approach may not fully leverage the valuable insights within the offline dataset.

## 3. Method

In this section, we present our MolStitch framework for tackling the offline MOMO problem. **There are three distinct neural networks in our framework:** *the generative model*, *StitchNet*, and *the proxy model*. The generative model is designed to generate molecules in textual formats, such as SMILES (Weininger, 1988). StitchNet takes two parent molecules as input and outputs a novel stitched molecule that combines desirable properties from both inputs. The proxy serves as a surrogate for evaluating molecules by classifying which molecule has more desirable properties.

### 3.1. Stage 1: Unsupervised Pre-training for StitchNet

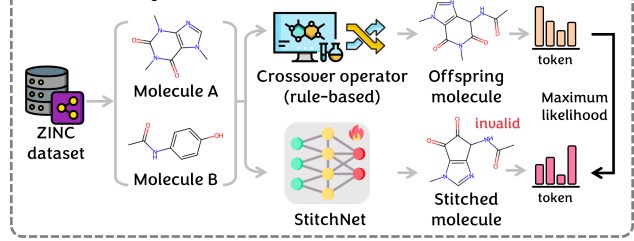

*Figure 3.* In the first pre-training stage, StitchNet gradually learns chemical grammar by imitating rule-based crossover operator.

In the first stage of our framework, we perform unsupervised pre-training for StitchNet using the publicly available ZINC dataset (Sterling & Irwin, 2015). Specifically, we randomly sample two parent molecules from the ZINC dataset and apply a rule-based crossover operator (Jensen, 2019) to generate an offspring molecule, as illustrated in Figure 3. Note that this crossover operator adheres to chemical rules and constraints, ensuring that the resulting offspring molecules are chemically valid and potentially exhibit desirable properties (Kamphausen et al., 2002). We then train StitchNet using a maximum likelihood objective to generate a stitched molecule that closely resembles the offspring molecules derived from the crossover operator. This pre-training encourages StitchNet to internalize chemical grammar, thereby enabling it to generate chemically valid stitched molecules.

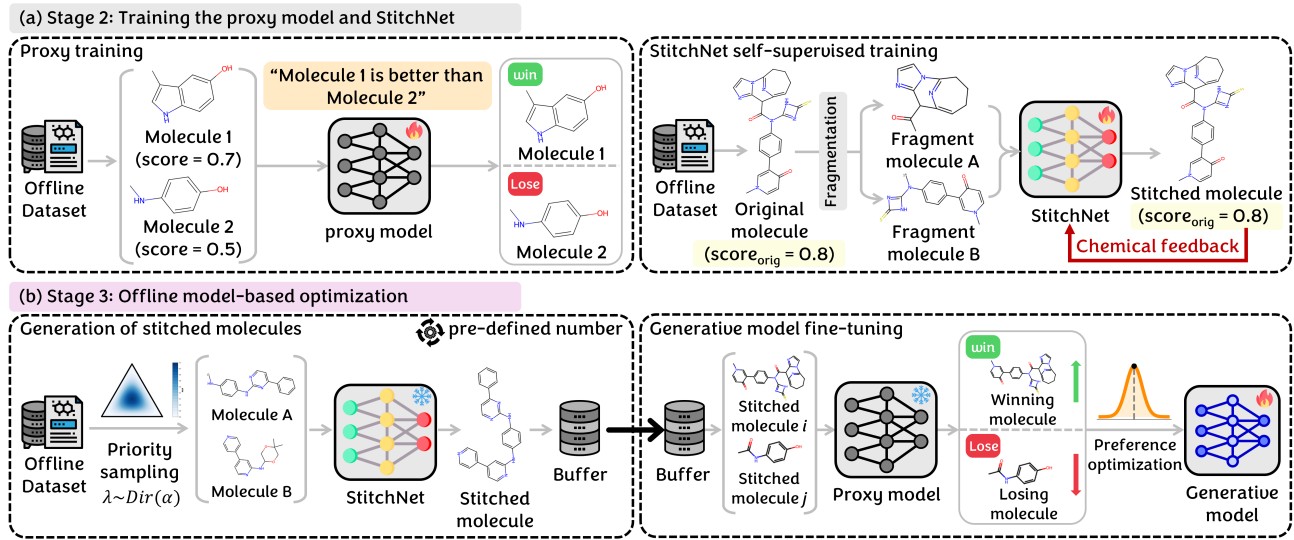

*Figure 4.* Main pipeline of the MolStitch framework. (a) Stage 2: The rank-based proxy is trained to classify which molecule in a given pair has desirable properties, while StitchNet undergoes self-supervised training with chemical feedback. (b) Stage 3: StitchNet generates stitched molecules, which are stored in a buffer. Once the buffer is full, the proxy evaluates the pairs and selects the superior molecule. Finally, the generative model is fine-tuned using preference optimization techniques to favor winning molecules and disfavor losing ones.

## 3.2. Stage 2: Training the Proxy model and StitchNet

**Rank-based proxy.** In the second stage, we obtain the offline dataset that consists of pre-collected molecules along with their true molecular objective scores. To facilitate the process of offline MBO, we require a proxy model capable of evaluating each molecule effectively. In this work, rather than training the proxy to approximate the exact objective scores, we train it to classify which molecule in a given pair has better objective scores. Specifically, as shown in Figure 4, we sample pairs of molecules from the offline dataset. Since we have access to the ground truth objective scores for each molecule in this dataset, we can establish a ranking between the molecules in each pair. We then train our rank-based proxy $\hat{f}_\theta$ using a pairwise ranking loss:

$$\mathcal{L}_{\text{proxy}}(\theta) = \sum_{(m_w, m_l) \in \mathcal{P}} \left[ -\log \sigma \left( \hat{f}_\theta(m_w) - \hat{f}_\theta(m_l) \right) \right], \quad (7)$$

where $\sigma(x) = \frac{1}{1+e^{-x}}$ is the sigmoid function, and $\mathcal{P}$ is the set of all valid molecule pairs $(m_w, m_l)$ within the offline dataset $\mathcal{D}$, defined as:

$$\mathcal{P} = \{(m_w, m_l) \mid m_w, m_l \in \mathcal{D},\ \mathbf{F}(m_w) > \mathbf{F}(m_l)\}. \quad (8)$$

**StitchNet.** While the pre-training stage focused on training StitchNet to learn chemical grammar and crossover operation, the focus in this stage is to integrate chemical feedback into StitchNet. To achieve this, we first sample the original molecule $m_{\text{orig}}$ from the offline dataset $\mathcal{D}$. Subsequently, we use the fragmentation function within the rule-based crossover operator to decompose this original molecule into two smaller fragment molecules. StitchNet is then employed to recombine these fragment molecules into a new stitched

molecule $\bar{m}_{\text{stit}}$. If the molecular similarity (Bender & Glen, 2004) between the original molecule and stitched molecules is above a certain threshold $\delta$, $\texttt{sim}(m_{\text{orig}}, \bar{m}_{\text{stit}}) \geq \delta$, we then train StitchNet $\mathcal{S}_\psi$ using the following loss function:

$$\mathcal{L}_{\text{stitch}}(\psi) = \sum_{m_{\text{orig}} \in \mathcal{D}} \mathbb{E}_{\bar{m}_{\text{stit}} \sim \mathcal{S}_\psi} \Bigg[ \Big( -\log \mathcal{S}_\psi(\bar{m}_{\text{stit}}) + \log \mathcal{S}_{\text{ref}}(\bar{m}_{\text{stit}}) + \mathcal{R}(m_{\text{orig}}) \Big)^2 \Bigg], \quad (9)$$

where $\mathcal{S}_{\text{ref}}$ refers to the pre-trained StitchNet that acts as a reference model for maintaining chemical validity. The $\mathcal{R}(m_{\text{orig}})$ represents the reward score, serving as chemical feedback derived from the given objective scores of $m_{\text{orig}}$. This loss $\mathcal{L}_{\text{stitch}}(\psi)$ guides $\mathcal{S}_\psi$ to generate $\bar{m}_{\text{stit}}$ with desirable objective scores, while not deviating too far from $\mathcal{S}_{\text{ref}}$.

Since we are addressing the offline MOMO problem, we cannot query the oracle to directly measure the objective scores of $\bar{m}_{\text{stit}}$ for computing $\mathcal{R}(\bar{m}_{\text{stit}})$. Instead, we utilize the given objective scores of $m_{\text{orig}}$ as a form of chemical feedback to approximate the objective scores of $\bar{m}_{\text{stit}}$. This approximation is reasonable because StitchNet generates $\bar{m}_{\text{stit}}$ by recombining fragment molecules that are derived directly from $m_{\text{orig}}$. Moreover, we ensure that $\bar{m}_{\text{stit}}$ is sufficiently similar to $m_{\text{orig}}$ through the similarity threshold $\delta$. This allows us to assume that the objective scores of $\bar{m}_{\text{stit}}$ are also similar to the objective scores of $m_{\text{orig}}$, as it is widely acknowledged that structurally similar molecules often exhibit similar properties and biological activities (Barbosa & Horvath, 2004; Alvesalo et al., 2006). Detailed visualization of this self-supervised training is presented in Appendix I.

## 3.3. Stage 3: Offline Model-based Optimization

In the third stage of our framework, we address the offline MOMO problem by utilizing the trained proxy model and StitchNet. The main goal of this stage is to train the generative model to generate novel molecules that potentially surpass the best-known molecule in $\mathcal{D}$. In the context of the MOMO problem, the scalarization approach is widely adopted, where a weighted sum of multiple objectives is combined into a single scalar objective, expressed as $F(m) = \sum_{i=1}^{k} \lambda_i f_i(m)$. Here, $k$ denotes the number of objectives, and $\lambda_i$ represents the weight assigned to each objective, reflecting its relative importance or priority. However, in an offline setting, the exact importance is often unknown, making it challenging to select proper weights. In addition, the goal of StitchNet is to combine molecules with different characteristics to generate novel stitched molecules that integrate desirable properties from both inputs. Hence, it is vital to provide StitchNet with diverse molecule pairs.

**Priority sampling.** To address these challenges, we introduce priority sampling using the Dirichlet distribution. This sampling approach generates a diverse set of weight configurations, allowing StitchNet to work with a wide variety of molecule pairs, each focusing on a different balance among multiple objectives. Our choice of the Dirichlet distribution is due to its capability to sample directly from the simplex, naturally providing valid weight combinations that are non-negative and sum to 1. The probability density function of the Dirichlet distribution can be expressed by:

$$p(\lambda_1, \lambda_2, \ldots, \lambda_k \mid \lambda \sim \text{Dir}(\alpha_1, \alpha_2, \ldots, \alpha_k)), \quad (10)$$

where $\text{Dir}(\cdot)$ refers to the Dirichlet distribution, and $\alpha$ denotes the concentration parameters. As illustrated in Figure 4, we use priority sampling $\lambda \sim \text{Dir}(\alpha_1, \alpha_2, \ldots, \alpha_k)$ to sample molecule pairs from the offline dataset. These sampled molecules are then fed into StitchNet, which outputs a novel stitched molecule $\bar{m}$. This newly generated stitched molecule is subsequently stored in a buffer $\mathcal{B}$ and utilized as a training sample for the fine-tuning training process of the generative model. Please refer to Appendix J for a detailed visualization and the rationale behind priority sampling.

**Preference optimization (fine-tuning).** Once the buffer $\mathcal{B}$ is populated with a pre-defined number of stitched molecules, we can proceed to fine-tune the generative model. Specifically, we sample pairs of stitched molecules $(\bar{m}_i, \bar{m}_j)$ from $\mathcal{B}$ and use our trained rank-based proxy to determine which molecule in each pair is more favorable such as:

$$(\bar{m}_w, \bar{m}_l) = \begin{cases} (\bar{m}_i, \bar{m}_j), & \text{if } \hat{f}_\theta(\bar{m}_i) > \hat{f}_\theta(\bar{m}_j) \\ (\bar{m}_j, \bar{m}_i), & \text{otherwise} \end{cases}, \quad (11)$$

where the winning and losing molecules are denoted as $\bar{m}_w$ and $\bar{m}_l$, respectively. Then, we can update the generative

model $G_\phi$ by increasing the log-likelihood of generating the winning molecule and decreasing the log-likelihood of the losing molecule. The loss for the generative model is:

$$\begin{aligned} \mathcal{L}_{\text{gen}}(\phi) = -\mathbb{E}\big[ &\log G_\phi(\bar{m}_w) - \log G_\phi(\bar{m}_l) \big] \\ &+ \beta \cdot \mathbb{D}_{\text{KL}}(G_\phi \| G_{\text{ref}}), \end{aligned} \quad (12)$$

where $G_{\text{ref}}$ represents the pre-trained generative model serving as a reference model. The KL divergence $\mathbb{D}_{\text{KL}}$ encourages $G_\phi$ not to deviate significantly from $G_{\text{ref}}$, ensuring that it maintains adherence to chemical validity. After formulating the initial loss function for the generative model, we can draw an intriguing parallel to preference optimization for language models (Rafailov et al., 2023; Tang et al., 2024). In this analogy, our generative model $G_\phi$ can be thought of as the language model and the favorable molecule $\bar{m}_w$ as the preferred response. This conceptual alignment allows us to incorporate various preference optimization techniques into our optimization process. Inspired by Direct Preference Optimization (DPO) (Rafailov et al., 2023), we can reformulate the Equation 12 into a DPO-like loss by employing the Bradley-Terry model (Bradley & Terry, 1952) such as:

$$\mathcal{L}_{\text{gen}}(\phi) = -\mathbb{E}\left[ \log \sigma \left( \beta \log \frac{G_\phi(\bar{m}_w)}{G_{\text{ref}}(\bar{m}_w)} - \beta \log \frac{G_\phi(\bar{m}_l)}{G_{\text{ref}}(\bar{m}_l)} \right) \right]. \quad (13)$$

This DPO-like loss integrates the separate KL divergence into a single term by utilizing the sigmoid of log odds ratios, simplifying the optimization process. However, despite its effectiveness, DPO is known to be prone to overfitting the preference dataset (Hu et al., 2024a). To address this, Identity Preference Optimization (IPO) (Azar et al., 2024) introduces a regularization term that penalizes the model when its confidence in the preference margin becomes excessively high. Building upon the concepts of IPO, we can modify the Equation 13 to adopt an IPO-like loss such as:

$$\mathcal{L}_{\text{gen}}(\phi) = -\mathbb{E}\left[ \left( \log \left( \frac{G_\phi(\bar{m}_w)}{G_\phi(\bar{m}_l)} \cdot \frac{G_{\text{ref}}(\bar{m}_l)}{G_{\text{ref}}(\bar{m}_w)} \right) - \frac{1}{2\beta} \right)^2 \right]. \quad (14)$$

Using the Equation 14, we fine-tune the generative model, REINVENT (Olivecrona et al., 2017), which is an RL-based model widely used for its robust performance across various molecular optimization tasks. More details of the generative model's loss function are in Appendix H, and the pseudocode for our MolStitch framework is in Appendix K.

## 4. Experiments

### 4.1. Experimental Design and Results

**Setup.** We conducted two main offline MOMO experiments to evaluate the effectiveness of our MolStitch framework. The first experiment focused on the Molecular Property Optimization (MPO) task (Gao et al., 2022), while the second

*Table 1.* Experimental results on the molecular property optimization task under the full-offline setting, with the best values in bold.

| Molecular objectives | GSK3$\beta$+JNK3 | | GSK3$\beta$+JNK3+QED | | GSK3$\beta$+JNK3+QED+SA | |
|---|---|---|---|---|---|---|
| Method | HV($\uparrow$) | R2($\downarrow$) | HV($\uparrow$) | R2($\downarrow$) | HV($\uparrow$) | R2($\downarrow$) |
| REINVENT | 0.462±0.133 | 0.921±0.259 | 0.196±0.083 | 2.646±0.327 | 0.168±0.046 | 3.969±0.664 |
| AugMem | 0.489±0.077 | 0.845±0.148 | 0.272±0.083 | 2.118±0.280 | 0.185±0.043 | 4.101±0.346 |
| GraphGA | 0.367±0.090 | 1.116±0.189 | 0.212±0.063 | 2.482±0.240 | 0.200±0.070 | 3.973±0.504 |
| Saturn | 0.531±0.087 | 0.785±0.159 | 0.293±0.058 | 1.977±0.280 | 0.281±0.058 | 3.339±0.280 |
| GeneticGFN | 0.482±0.073 | 0.869±0.117 | 0.309±0.087 | 1.990±0.365 | 0.237±0.066 | 3.630±0.453 |
| Grad | 0.494±0.058 | 0.857±0.126 | 0.205±0.045 | 2.502±0.231 | 0.171±0.026 | 4.176±0.319 |
| COMs | 0.479±0.063 | 0.877±0.109 | 0.205±0.072 | 2.496±0.288 | 0.171±0.062 | 4.219±0.628 |
| IOM | 0.506±0.070 | 0.807±0.138 | 0.215±0.060 | 2.380±0.336 | 0.195±0.065 | 4.042±0.529 |
| RoMA | 0.492±0.091 | 0.843±0.177 | 0.198±0.052 | 2.537±0.269 | 0.169±0.071 | 4.207±0.617 |
| Ensemble Proxy | 0.500±0.033 | 0.835±0.055 | 0.218±0.039 | 2.462±0.160 | 0.213±0.057 | 3.888±0.529 |
| BIB | 0.486±0.070 | 0.874±0.120 | 0.203±0.049 | 2.503±0.245 | 0.172±0.027 | 4.080±0.387 |
| BootGen | 0.540±0.113 | 0.741±0.167 | 0.225±0.067 | 2.452±0.319 | 0.201±0.074 | 4.092±0.560 |
| ICT | 0.514±0.049 | 0.827±0.104 | 0.213±0.080 | 2.429±0.385 | 0.180±0.060 | 4.197±0.593 |
| Tri-Mentoring | 0.510±0.042 | 0.824±0.079 | 0.216±0.071 | 2.458±0.363 | 0.195±0.057 | 4.067±0.467 |
| RaM | 0.492±0.062 | 0.851±0.117 | 0.282±0.057 | 2.268±0.281 | 0.249±0.049 | 3.541±0.412 |
| MolStitch (Ours) | **0.579±0.070** | **0.698±0.128** | **0.403±0.065** | **1.649±0.259** | **0.352±0.080** | **2.953±0.571** |

addressed the docking score optimization task (Lee et al., 2023). In the first experiment, we closely followed prior studies (Xie et al., 2021b; Shin et al., 2024b) and adopted four widely used molecular objectives. The objectives include JNK3 and GSK3$\beta$, which evaluate inhibition against target proteins associated with Alzheimer's disease, along with QED and SA, which are drug-likeness and synthesizability. For the second experiment, we also closely followed recent work (Guo & Schwaller, 2024b) and targeted the docking score optimization of five proteins—parp1, fa7, jak2, braf, and 5ht1b—alongside QED and SA. All experiments were conducted under offline setting and repeated with 10 different seeds. Further details are in Appendix L.

**Competing methods.** We compared our framework against two main categories of methods: molecular optimization and offline optimization. For molecular optimization, we included REINVENT (Olivecrona et al., 2017), AugMem (Guo & Schwaller, 2024a), GraphGA (Jensen, 2019), Saturn (Guo & Schwaller, 2024b), and GeneticGFN (Kim et al., 2024a). For offline optimization, we incorporated various offline MBO methods, including Gradient ascent (Grad) (Zinkevich, 2003), COMs (Trabucco et al., 2021), IOM (Qi et al., 2022), RoMA (Yu et al., 2021), Ensemble Proxy (Trabucco et al., 2022), ICT (Yuan et al., 2023), and Tri-Mentoring (Chen et al., 2023a). We also included BIB (Chen et al., 2023b), BootGen (Kim et al., 2023), and RaM (Tan et al., 2025), which are state-of-the-art models for offline optimization in biological sequence design. Note that we used REINVENT as the backbone generative model for all offline optimization methods, not only because it is one of the most robust models for diverse molecular optimization tasks, but also to ensure fairness, as REINVENT also serves as the main backbone model in our framework. For instance, the Grad refers to REINVENT with a vanilla proxy, followed by a fine-tuning via gradient ascent. Detailed descriptions of each competing method are presented in Appendix N.

*Table 2.* Experimental results on the docking score optimization task under the full-offline setting, with the best values in bold.

| Target protein | parp1 | jak2 | braf | fa7 | 5ht1b |
|---|---|---|---|---|---|
| Method | HV($\uparrow$) | HV($\uparrow$) | HV($\uparrow$) | HV($\uparrow$) | HV($\uparrow$) |
| REINVENT | 0.515 | 0.477 | 0.500 | 0.414 | 0.509 |
| Saturn | 0.528 | 0.498 | 0.523 | 0.431 | 0.537 |
| GeneticGFN | 0.539 | 0.476 | 0.508 | 0.441 | 0.523 |
| BootGen | 0.544 | 0.496 | 0.524 | 0.436 | 0.545 |
| RaM | 0.542 | 0.488 | 0.528 | 0.424 | 0.525 |
| MolStitch (Ours) | **0.560** | **0.515** | **0.554** | **0.451** | **0.575** |

**Evaluation metrics.** The performance of each method was evaluated using two evaluation metrics: the hypervolume indicator (HV) (Zitzler et al., 2003) and the R2 indicator (Brockhoff et al., 2012). The HV quantifies the volume of the space dominated by a set of solutions on the Pareto front, where higher values reflect better performance. In contrast, the R2 assesses the quality of a solution set by measuring the projection onto pre-defined reference points, with lower values indicating better performance. More detailed explanations of evaluation metrics are in Appendix O.

**Main results.** As shown in Table 1, we present the mean HV and R2 performance along with their standard deviations for the MPO task under the full-offline setting. We observed that our MolStitch framework consistently demonstrated superior performance across all scenarios with varying numbers of molecular objectives. This underscores the efficacy of our StitchNet in addressing the offline MOMO problem, as it leverages existing molecules to create novel stitched molecules, which serve as valuable training samples for fine-tuning the generative model. Among the competing methods, Saturn and GeneticGFN exhibited strong performance, both of which are recent methods that employ genetic algorithms. BootGen and RaM also demonstrated effectiveness as they are recognized as state-of-the-art offline MBO methods for biological sequence design. Furthermore,

*Table 3.* An ablation study for each component in MolStitch: Rank-based Proxy (RP), StitchNet (SN), and Priority Sampling (PS).

| Ablation study | | | GSK3$\beta$+JNK3+QED+SA | |
|---|---|---|---|---|
| RP | SN | PS | HV($\uparrow$) | R2($\downarrow$) |
| - | - | - | $0.171_{\pm 0.026}$ | $4.176_{\pm 0.319}$ |
| - | ✔ | - | $0.193_{\pm 0.053}$ | $4.134_{\pm 0.502}$ |
| - | ✔ | ✔ | $0.220_{\pm 0.054}$ | $3.835_{\pm 0.483}$ |
| ✔ | - | - | $0.251_{\pm 0.084}$ | $3.504_{\pm 0.634}$ |
| ✔ | ✔ | - | $0.289_{\pm 0.096}$ | $3.317_{\pm 0.713}$ |
| ✔ | ✔ | ✔ | $\mathbf{0.352}_{\pm \mathbf{0.080}}$ | $\mathbf{2.953}_{\pm \mathbf{0.571}}$ |

*Table 4.* Performance comparison of different data augmentation techniques in offline multi-objective molecular optimization.

| Molecular objectives | GSK3$\beta$+JNK3+QED+SA | |
|---|---|---|
| Data augmentation | HV($\uparrow$) | R2($\downarrow$) |
| Baseline (REINVENT) | $0.168_{\pm 0.046}$ | $3.969_{\pm 0.664}$ |
| + Stochastic sampling | $0.251_{\pm 0.084}$ | $3.504_{\pm 0.634}$ |
| + Crossover operator | $0.302_{\pm 0.072}$ | $3.110_{\pm 0.479}$ |
| + StitchNet (Ours) | $\mathbf{0.352}_{\pm \mathbf{0.080}}$ | $\mathbf{2.953}_{\pm \mathbf{0.571}}$ |

we evaluated the performance of our framework on an additional protein docking score optimization task. As presented in Table 2, MolStitch consistently outperformed all competing methods across all five proteins in terms of the HV performance, highlighting the robustness and effectiveness of our framework. Full HV and R2 results for docking score optimization tasks are provided in Appendix C.1 and C.2.

**Additional results.** Recently, semi-offline optimization, also known as batch hybrid learning, has gained significant attention in the field of large language models (Xiong et al., 2024). Specifically, this semi-offline setting allows for a limited number of online human feedback cycles and enables the model to be fine-tuned on new data through large batches. Inspired by this approach, we conducted additional experiments for the semi-offline setting, starting with an offline dataset and periodically querying oracle functions to evaluate molecules in large batches. Due to page limit, the results are deferred to Appendix C.4, where our MolStitch framework maintained its superior performance even under the semi-offline setting. We further investigated the impact of using different backbone generative models in MolStitch beyond REINVENT and confirmed its robustness across various backbone models, as detailed in Appendix C.5.

### 4.2. Ablation Study

To investigate the impact of each key component in our framework—Rank-based Proxy (RP), StitchNet (SN), and Priority Sampling (PS)—we conducted an ablation study.

**Effects of rank-based proxy.** When RP was ablated and replaced with a score-based proxy, which is analogous to the vanilla proxy that directly approximates objective scores, we observed a noticeable drop in performance. This observation highlights the efficacy of reformulating the proxy's task from score regression to pairwise classification, making the task easier and provide reliable feedback. Detailed investigations of score- and rank-based proxies are provided in Appendix D. In addition, we extended RP by employing multiple proxies, with the results presented in Appendix E.

**Benefits of StitchNet.** The ablation study highlighted the significant impact of SN in offline optimization process. By generating novel stitched molecules, SN provides valuable

training samples for fine-tuning the generative model. Importantly, SN incorporates a crossover mechanism similar to that in genetic algorithms but with the added capability of receiving chemical feedback. The efficacy of this crossover operation was validated in our main results, where genetic algorithm-based methods like GeneticGFN and Saturn also demonstrated strong performance. These findings suggest that incorporating the crossover operation, as SN does, is beneficial because it naturally promotes diversity by exploring novel combinations derived from existing molecules. Further investigations of SN can be found in Appendix P.

**Benefits of priority sampling.** PS played a crucial role in generating diverse weight configurations, which enabled SN to operate with a wide variety of molecule pairs. In the ablation study, PS significantly improved performance by enabling our framework to effectively navigate complex Pareto fronts and explore a broader range of trade-offs through diverse weight configurations. Additional insights into the impact of PS, especially in addressing the *reward hacking problem* in offline MOMO, are in Appendix Q.

### 4.3. Experimental Analysis and Discussion

**Data augmentation.** In our main results, we observed that employing StitchNet as a data augmentation approach significantly enhanced performance in offline MOMO. To investigate its effectiveness, we compared StitchNet with other data augmentation techniques. One technique is stochastic sampling, where new molecules are stochastically drawn from the generative model's learned distribution. To put it simply, this process can be represented in code-level terms as `model.sample()`. Another technique is the crossover operator, used in GeneticGFN and Saturn, which generates new offspring molecules by combining features from parent molecules in a rule-based manner. As shown in Table 4, all data augmentation techniques outperformed the baseline, underscoring their effectiveness in offline MOMO. The crossover operator demonstrated better performance than stochastic sampling due to its ability to combine existing high-quality molecules to create diverse offspring molecules. Most importantly, StitchNet achieved the best performance, demonstrating its effectiveness by leveraging the capability of a neural network to integrate valuable chemical feedback.

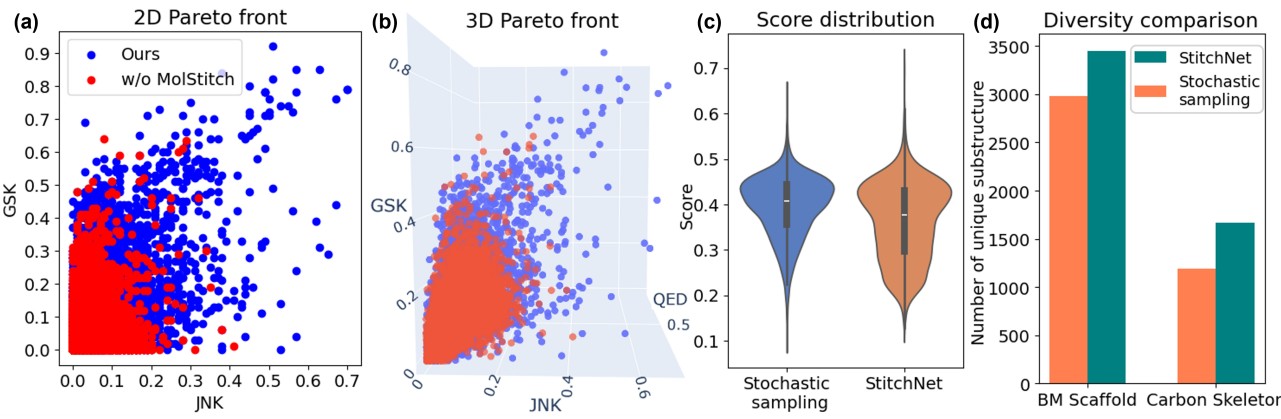

*Figure 5.* Visualizations of the 2D and 3D Pareto fronts (a-b), along with diversity analysis for StitchNet presented in (c-d).

*Table 5.* Performance comparison of various preference optimization techniques in offline multi-objective molecular optimization.

| Molecular objectives | GSK3$\beta$+JNK3+QED+SA | |
| --- | --- | --- |
| Preference optimization | HV($\uparrow$) | R2($\downarrow$) |
| Baseline (REINVENT) | 0.168$\pm$0.046 | 3.969$\pm$0.664 |
| + StitchNet & RLHF | 0.232$\pm$0.071 | 3.715$\pm$0.611 |
| + StitchNet & DPO | 0.327$\pm$0.081 | 3.015$\pm$0.493 |
| + StitchNet & IPO | 0.344$\pm$0.082 | 2.955$\pm$0.533 |
| + StitchNet & IPO & RP (Ours) | **0.352**$\pm$**0.080** | **2.953**$\pm$**0.571** |

**Preference optimization.** In our MolStitch framework, we fine-tuned the generative model using a process analogous to the preference optimization techniques employed in large language models. To evaluate different preference optimization techniques in offline MOMO, we explored alternatives such as RLHF (Ouyang et al., 2022), where the proxy model serves as a reward model to generate reward scores that are directly optimized. Other approaches involved removing the proxy by allowing the generative model to act as a judge (model-as-a-judge) to directly classify winning and losing molecules and update itself using DPO or IPO loss. As illustrated in Table 5, our MolStitch consistently outperformed other techniques by constructing the separate rank-based proxy (RP) for molecule evaluation and updating the generative model separately based on this proxy feedback. This separation has shown to be effective, as supported by recent studies (Singhal et al., 2024; Liu et al., 2024b), where maintaining a separate reward-ranking model helps to mitigate distributional shifts and enhance performance.

**Pareto front visualization.** To evaluate the impact of MolStitch on solution quality, we visualized the Pareto front in both 2D and 3D objective spaces. As depicted in Figure 5 (a-b), the Pareto front obtained from MolStitch dominated the baseline without MolStitch, indicating superior performance across all objectives. Notably, the solutions generated by MolStitch were concentrated in the upper right region of the Pareto front, signifying the effectiveness of molecular stitching process derived from StitchNet in offline MOMO.

**Diversity analysis.** In offline MOMO, promoting molecular diversity is crucial for identifying candidates with desirable properties while avoiding over-exploration of similar structures. To assess the diversity of augmented molecules generated by StitchNet in comparison to stochastic sampling, we visualized their objective score distributions using violin plots. As shown in Figure 5 (c), StitchNet exhibited a broader and more varied score distribution, demonstrating its capacity to provide a diverse range of augmented molecules for the generative model. We also evaluated the final molecules produced by the generative model fine-tuned with StitchNet against those from stochastic sampling, using diversity metrics that measure the number of unique substructures, specifically Bemis-Murcko (BM) scaffolds and carbon skeletons (Bemis & Murcko, 1996). As depicted in Figure 5 (d), the generative model fine-tuned with StitchNet exhibited greater diversity compared to stochastic sampling across both BM scaffolds and carbon skeletons. Additional diversity analysis for StitchNet is available in Appendix F.

**Scalarization analysis.** In our MolStitch framework, we initially employed linear scalarization due to its simplicity and foundational role in multi-objective optimization. However, we further investigated advanced scalarization techniques such as Chebyshev scalarization. Unlike linear scalarization, which combines objectives into a weighted sum, Chebyshev scalarization aims to minimize the maximum deviation from an ideal objective values. This characteristic enables it to better navigate non-convex Pareto fronts (Deb et al., 2016). To assess its compatibility with MolStitch, we conducted experiments integrating Chebyshev scalarization. As shown in tables provided in Appendix R, MolStitch with Chebyshev achieved comparable performance to the linear approach in the two-objective setting. However, it outperformed the linear approach in more complex three- and four-objective settings. We attribute this improvement to the growing complexity and non-convexity of the Pareto front as the number of objectives increases, where Chebyshev's emphasis on balancing extreme trade-offs provides greater robustness.

# 5. Related work

## 5.1. Generative Models for Molecular Discovery

In the past few years, a wide range of generative models has been proposed for molecular discovery, showing significant success in exploring chemical space and optimizing molecular properties. These models are commonly grouped into four main categories: genetic algorithms, sampling-based methods, reinforcement learning, and probabilistic methods.

**Genetic algorithms** operate by evolving a population of candidate molecules through mutation and crossover operations, guided by a fitness function. Then, high-performing candidates are selected and propagated to subsequent generations, with the aim of progressively improving population quality. A prominent example is GraphGA (Jensen, 2019), which demonstrates strong performance in navigating chemical space to generate molecules with desirable properties.

**Sampling-based methods** generate molecules by sampling from distributions that are biased toward regions of chemical space likely to yield desirable molecular properties. A representative example is MARS (Xie et al., 2021a), which employs Markov Chain Monte Carlo (MCMC) sampling to explore chemical space and identify high-quality molecules.

**Reinforcement learning** formulate molecular generation as a sequential decision-making process, where an agent interacts with a chemical environment to construct molecular structures through a series of actions. A prominent example is REINVENT (Olivecrona et al., 2017), which employs an agent to generate molecules in SMILES format in an autoregressive manner. The agent learns to improve its policy over time by receiving reward feedback based on the desirability of generated molecules with respect to target properties.

**Probabilistic methods** also frame molecular generation as a sequential process, similar to reinforcement learning. However, they differ in that actions are sampled from a learned probability distribution rather than a deterministic policy. A representative example is GFlowNets (Jain et al., 2022), which aim to generate diverse and high-reward molecules by sampling from a distribution proportional to their reward.

## 5.2. Multi-Objective Molecular Optimization (MOMO)

The MOMO problem involves simultaneously optimizing multiple, often conflicting, molecular objectives. Since a single solution cannot usually satisfy all objectives, the goal is to identify a diverse set of Pareto-optimal molecules that represent trade-offs among objectives. A common approach is to use scalarization techniques—such as weighted sums or Chebyshev—to effectively aggregate multiple objectives into a single objective function. For example, GeneticGFN (Kim et al., 2024a) employs linear scalarization to manage multiple objectives and demonstrates strong performance.

## 5.3. Offline Model-based Optimization (MBO)

The most straightforward implementation of offline MBO is to use a vanilla proxy to estimate objective scores and applies gradient ascent to update the generative model directly. However, this approach often suffers from inaccurate proxy predictions, which can misguide the optimization process. To address this issue, various methods have been proposed.

**Improving the proxy model.** One line of research focuses on improving the accuracy and robustness of the proxy itself. For instance, COMs (Trabucco et al., 2021) employ adversarial learning to encourage the proxy to produce conservative estimates. Similarly, IOM (Qi et al., 2022) introduces invariant representation learning through domain adaptation techniques to mitigate the impact of distributional shifts. Additionally, methods such as ICT (Yuan et al., 2023) and Tri-Mentoring (Chen et al., 2023a) leverage multiple proxy models to take advantage of ensemble learning.

**Improving optimization algorithms.** Another line of research focuses on enhancing the optimization strategies. For instance, BIB (Chen et al., 2023b) incorporates bidirectional learning by combining forward and backward mappings to generate input samples that are likely to yield high-quality outputs. BootGen (Kim et al., 2023) enhances optimization through a bootstrapping strategy that iteratively augments the offline dataset with top-performing synthetic samples. RaM (Tan et al., 2025) employs learning-to-rank techniques to guide optimization based on relative rankings, which is conceptually similar to our rank-based proxy. However, the novelty of our framework is characterized by its combination of molecular stitching, rank-based proxy, priority sampling, and preference optimization to tackle offline MOMO.

Extended related work section is provided in Appendix M.

# 6. Conclusion

In this study, we address two key challenges in real-world molecular discovery: *offline molecular optimization* and *multi-objective molecular optimization (MOMO)*. To tackle these challenges, we propose the MolStitch framework. For offline molecular optimization, MolStitch uses StitchNet to generate novel stitched molecules by combining the desirable properties from two parent molecules in an offline dataset. These stitched molecules are then evaluated using a rank-based proxy, which determines the more favorable molecule in a given pair. Using feedback from this proxy, the generative model is fine-tuned through preference optimization to enhance its capability to produce high-quality molecules. For MOMO, we further introduce priority sampling to explore trade-offs among multiple objectives more effectively. Through extensive experiments, we validate the efficacy of MolStitch in addressing offline MOMO problem. Future work and limitations can be found in Appendix T.

## Acknowledgment

This work was supported by the Institute of Information & Communications Technology Planning & Evaluation (IITP), funded by the Korea government (MSIT), through the Artificial Intelligence Graduate School Program at Korea University (No. RS-2019-II190079); and by the National Research Foundation of Korea (NRF) grants funded by the Korea government (MSIT) (No. RS-2023-00212498) and (No. RS-2024-00415812). It was also supported by the Ministry of Science and ICT, Korea government (MSIT), through the Information Technology Research Center (ITRC) support program (IITP-2025-RS-2024-00436857).

## Impact Statement

In this study, we address the offline multi-objective molecular optimization problem, which has potential applications in drug discovery. We emphasize the responsible application of our methodologies, with a strong focus on safety considerations. Although our framework enhances the efficiency of molecular optimization, it is crucial that all identified molecules must undergo experimental validation, safety assessments, and regulatory approval before being considered for real-world deployment. We caution against relying solely on computationally generated molecules without proper testing, as it could lead to unintended health or environmental risks. Furthermore, all datasets used in this study are publicly accessible and comply with ethical standards.

## Reproducibility Statement

We provide comprehensive information on the experimental settings, workflow, hyperparameters, and implementation details in Appendix L. Additionally, the source code for our proposed framework is available online at `https://github.com/MolecularTeam/MolStitch`.

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

# Appendix

## A. Online and Offline settings for Molecular Discovery

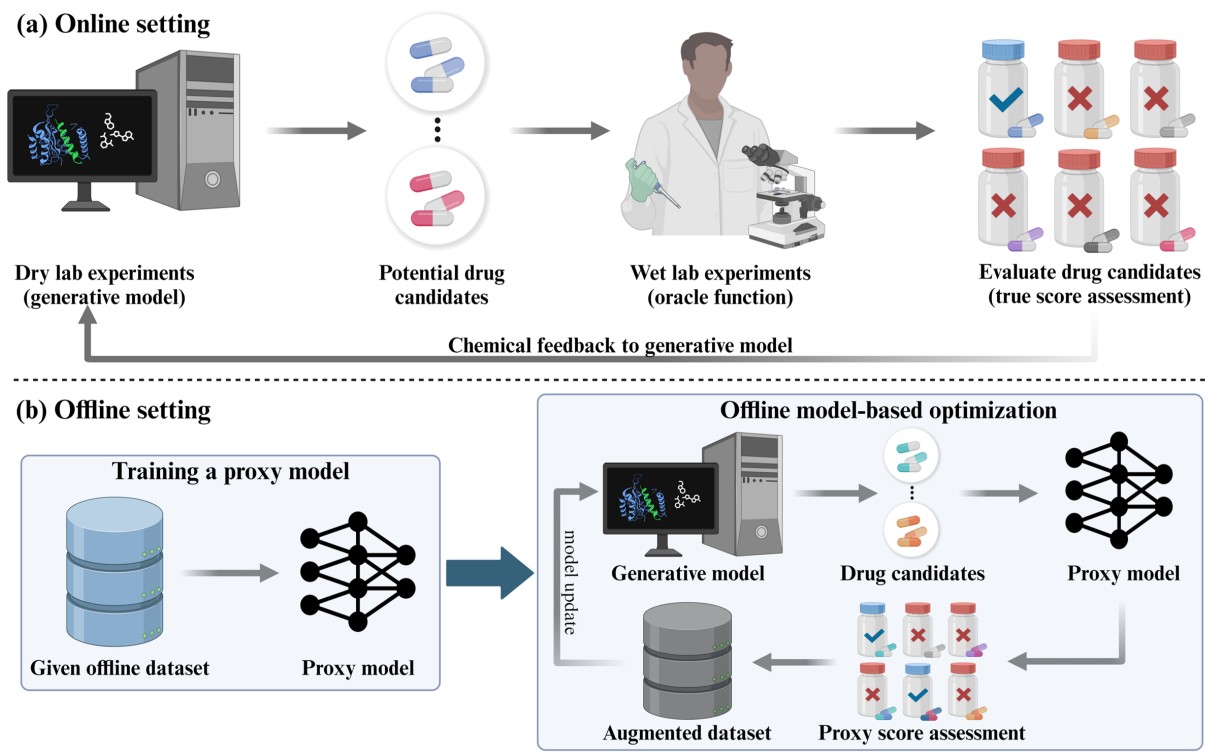

*Figure 6.* An illustration of online and offline settings for molecular discovery.

In this section, we delve into the detailed pipeline for online and offline settings in molecular discovery, using the specific case of *in silico* drug discovery combined with real-world wet lab experiments as an illustrative example.

**Online setting.** Traditionally, many *in silico* drug discovery methods have been based on the assumptions of the online setting (Jiménez-Luna et al., 2021). As depicted in Figure 6 (a), the online setting begins in the computational or 'dry' lab, where a generative model produces potential drug candidates that are predicted to be potent. Researchers then select the top $K$ drug candidates or apply specific filters to choose which drug candidates to advance. These selected drugs are sent to the wet lab, where they undergo physical biological experiments to validate their efficacy. Note that these wet lab experiments serve as a true oracle function, which provides accurate assessments of drug potency and properties based on real-world testing. Once the wet lab experiments are complete, the results—true score assessments—are sent back to the dry lab as chemical feedback. This feedback is then used to update the generative model, enabling it to produce more desirable and potential drug candidates in the next iteration. This iterative process continues until successful drug candidates are identified or predefined criteria, such as reaching a certain optimization score, are met.

**Why offline setting?** The main advantage of the online setting is its ability to continuously refine the generative model using feedback from the true oracle function. However, this feedback relies on real-world wet lab experiments, which are typically time-consuming and costly. Therefore, querying the true oracle function for every drug candidate is often impractical, and safety concerns can further limit its use (Loiodice et al., 2019; Yusuf, 2023). Even if we assume these challenges are mitigated and resources are available to query the true oracle function as needed, there still remains the challenge of a significant time mismatch between the dry lab and the wet lab. The dry lab can generate new drug candidates within hours, but the wet lab evaluation—including chemical synthesis, purification, and biological testing—can take weeks or even months (Payton et al., 2023). This significant lag means that while the wet lab is engaged in lengthy experiments, *in silico* generative models in dry lab remain idle, leading to inefficiencies and underutilization of computational resources. To address these limitations, the offline setting has gained considerable attention in recent years (Xue et al., 2024). In the offline setting, the generative model can be trained using existing offline datasets without relying on continuous feedback from wet lab experiments. This would allow generative models to be continually improved while awaiting wet lab results,

enabling the generation of higher-quality candidate molecules for subsequent experimental rounds.

**Offline setting.** One of the most prevalent and widely adopted approaches for handling the offline setting is offline model-based optimization (MBO). As illustrated in Figure 6 (b), the process begins by training a proxy model on the given offline dataset. This proxy model serves as a surrogate for evaluating drug candidates, as access to the true oracle function is not available in the offline setting. Once the proxy model is trained, the offline MBO process is initiated to enable the training of the generative model without relying on real-world wet lab feedback. Specifically, the generative model produces new drug candidates, which are evaluated by the proxy model instead of being sent to the wet lab. The proxy model provides estimated proxy scores for these candidates, and these pairs of drug candidates and their proxy scores are stored in a buffer to create an augmented dataset. This augmented dataset is then utilized to update the generative model via gradient ascent, leveraging the proxy model's predictions. This iterative cycle continues until predefined criteria are met.

**Our Framework: MolStitch.** After introducing the online and offline settings in molecular discovery, we now highlight how our proposed framework, MolStitch, differentiates itself from conventional offline MBO methods for tackling the offline multi-objective molecular optimization (MOMO) problem. First, instead of using a generative model to produce augmented molecules, MolStitch employs *StitchNet* to generate novel stitched molecules. In molecular discovery, promoting molecular diversity is crucial because structurally diverse molecules capable of producing similar biological effects help mitigate the development of resistance and improve the overall efficacy of treatments. As discussed in the main manuscript and Appendix F, we validated that StitchNet exhibits better molecular diversity compared to conventional augmentation derived from generative models. This is because StitchNet naturally promotes diversity by exploring novel combinations derived from existing parent molecules. Second, conventional offline MBO methods utilize a score-based proxy to approximate the true oracle function by regressing objective scores of molecules. However, accurately approximating these scores is challenging due to scarce data and the inherent difficulty of molecular property prediction tasks—a challenge that is even more pronounced in the MOMO problem. In MolStitch, we develop a *rank-based proxy* that learns the ranking relationship between pairs of molecules based on desired properties, classifying which molecule is more favorable. This transformation simplifies the task for the proxy, enabling it to provide more reliable feedback by focusing on comparative assessments rather than exact score predictions. Lastly, because we use a rank-based proxy, we cannot fine-tune the generative model, REINVENT, using conventional ways that require reward scores for updates (e.g., a score-based proxy could directly regress the objective score and use it as a pseudo-reward). To address this, we introduce *preference optimization techniques* to fine-tune and update the generative model. Specifically, instead of using pseudo-reward scores, the preference optimization technique enables the model to increase the log-likelihood of generating the preferred (winning) molecule while decreasing the log-likelihood of generating the less preferred (losing) molecule. As shown in Table 5 of our manuscript, experimental results demonstrate that this preference optimization is effective and significantly improves performance. In summary, MolStitch differentiates itself from conventional offline MBO methods by promoting molecular diversity through the use of StitchNet, simplifying proxy modeling via a rank-based approach, and incorporating preference optimization techniques to fine-tune the generative model without relying on approximating objective scores. These novel components collectively contribute to a more efficient and effective approach for tackling the offline MOMO problem.

## B. Experimental Workflow for Offline Multi-Objective Molecular Optimization (MOMO)

**Overall workflow.** In this subsection, we aim to conduct an in-depth exploration and comparison of key components in offline MOMO. Specifically, our goal is to outline the critical components that should be considered for solving the offline MOMO problem, discuss the available options for each component, and explain the rationale behind our choices. Figure 7 provides a visual representation of the overall workflow for addressing the offline MOMO problem. The primary objective of offline MOMO is to enhance the generative model's capability to generate molecules that surpass the best-known molecules in the offline dataset. To achieve this, the predominant approach is offline MBO, which involves training a proxy model, performing data augmentation, generating synthetic data, and subsequently training the generative model with this synthetic data under the guidance of the proxy model. Consequently, data augmentation is a pivotal aspect of the offline MOMO problem, and we begin our discussion with this component.

**Data augmentation.** As highlighted in the main manuscript, we propose StitchNet as a neural network model designed for data augmentation, and demonstrate its effectiveness. However, we acknowledge that StitchNet is not the only viable option. Alternative approaches include stochastic sampling, where new molecules are randomly drawn from the generative model's learned distribution. Additionally, rule-based crossover operators from genetic algorithms can be employed to generate new offspring molecules by combining features from parent molecules.

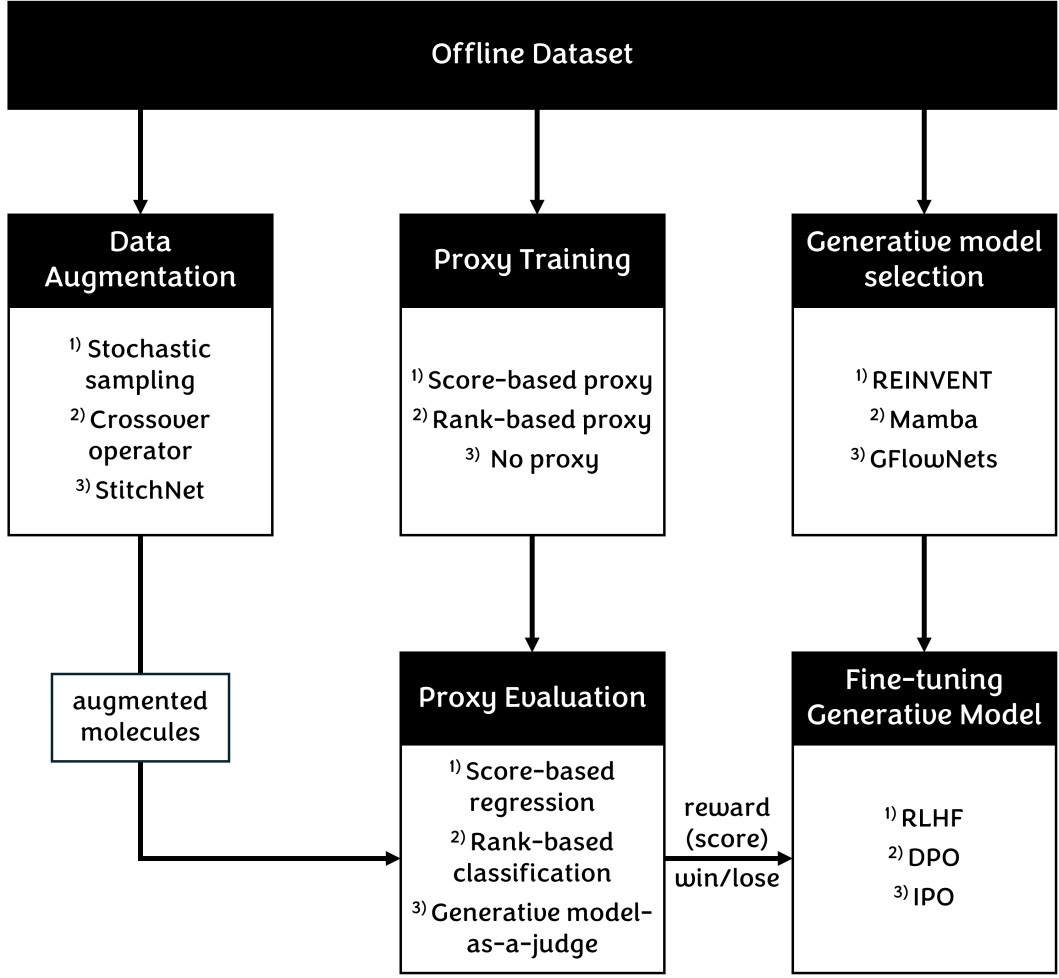

*Figure 7.* An illustration of the overall workflow for the offline molecular optimization process.

**Proxy training and evaluation.** After augmenting the synthetic data, the next step involves training a proxy model to evaluate this augmented dataset. The most straightforward approach is the score-based proxy (vanilla proxy), which directly approximates the scores of the true objective function. However, we anticipate that as the problem complexity increases, the vanilla proxy may encounter challenges and yield unreliable predictions. To mitigate this, we propose a rank-based proxy that learns the ranking relationships between pairs of molecules based on desired properties, thereby classifying which molecule is more favorable. This transformation from a regression task to a classification task simplifies the proxy's role, enhancing its reliability in providing feedback to the generative model. It is worth noting that a proxy model is not always necessary; in some cases, the generative model itself can evaluate new synthetic data, a mechanism referred to as the "model-as-a-judge".

**Generative model selection.** Several generative models are available for molecular optimization. In this work, we employ REINVENT as our main generative model due to its widespread use and recognition in various molecular optimization tasks. Nonetheless, recent advancements have introduced new generative models such as Mamba and GFlowNets. To ensure the robustness and versatility of our MolStitch, we also evaluate various backbone generative models within this framework.

**Fine-tuning the generative model.** With synthetic data, a trained proxy model, and a trained generative model in place, the final step involves fine-tuning the generative model using the synthetic data guided by the proxy model. This fine-tuning process can be considered analogous to the preference optimization process used in large language models. Therefore, we explore various preference optimization techniques within the context of offline MOMO. The first option is RLHF, where

the proxy model serves as a reward model to generate rewards that are directly optimized. Another option is DPO, which bypasses reward modeling and focuses on optimizing preferences directly. Lastly, IPO can be applied as an extension of DPO, providing a more theoretically sound and principled approach to preference optimization.

**Overview of MolStitch components.** Table 6 presents a detailed summary of the components constituting the MolStitch framework, including its variants and the methods examined in our ablation studies. We hope that this table helps to understand the function of each component in our framework and facilitates a clearer understanding of the structure of MolStitch and its variants.

*Table 6.* Summary of our MolStitch framework components, its variants, and the methods utilized in our ablation studies.

| Experiment | Data Augmentation | Proxy Training | Generative Model | Proxy Evaluation | Fine-tuning |
|---|---|---|---|---|---|
| MolStitch (Table 1) | StitchNet | Rank-based proxy | REINVENT | Rank-based classification | IPO |
| Score-based proxy (Table 3) | Stochastic sampling | Score-based proxy | REINVENT | Score-based regression | RLHF |
| Stochastic sampling (Table 4) | Stochastic sampling | Rank-based proxy | REINVENT | Rank-based classification | IPO |
| Crossover operator (Table 4) | Crossover operator | Rank-based proxy | REINVENT | Rank-based classification | IPO |
| StitchNet & RLHF (Table 5) | StitchNet | Rank-based proxy | REINVENT | Score-based regression | RLHF |
| StitchNet & DPO (Table 5) | StitchNet | No proxy | REINVENT | Generative model-as-a-judge | DPO |
| StitchNet & IPO (Table 5) | StitchNet | No proxy | REINVENT | Generative model-as-a-judge | IPO |
| Mamba + MolStitch (Table 12) | StitchNet | Rank-based proxy | Mamba | Rank-based classification | IPO |
| GFlowNets + MolStitch (Table 12) | StitchNet | Rank-based proxy | GFlowNets | Rank-based classification | IPO |

# C. Additional Results

## C.1. Full HV Performance Results for the Docking Score Optimization Task

**Results for HV performance.** In our main results, we only presented performance evaluations using the Hypervolume (HV) indicator for a select group of strong competing methods. Specifically, we highlighted Saturn and GeneticGFN, both of which demonstrated strong performance and represent recent advancements employing genetic algorithms, as well as BootGen and RaM, recognized as state-of-the-art offline MBO methods for biological sequence design. Due to space constraints, our primary results focused on these prominent methods. However, in this subsection, we provide the full results for the docking score optimization task, offering a comprehensive comparison across various competing methods.

*Table 7.* Experimental results on docking score optimization tasks under the full-offline setting using the HV indicator.

| Target protein | parp1 | jak2 | braf | fa7 | 5ht1b |
|---|---|---|---|---|---|
| Method | HV($\uparrow$) | HV($\uparrow$) | HV($\uparrow$) | HV($\uparrow$) | HV($\uparrow$) |
| REINVENT | $0.515\pm0.016$ | $0.477\pm0.009$ | $0.500\pm0.008$ | $0.414\pm0.006$ | $0.509\pm0.011$ |
| AugMem | $0.532\pm0.039$ | $0.499\pm0.053$ | $0.511\pm0.008$ | $0.430\pm0.038$ | $0.521\pm0.014$ |
| Saturn | $0.528\pm0.009$ | $0.498\pm0.030$ | $0.523\pm0.046$ | $0.431\pm0.034$ | $0.537\pm0.033$ |
| GeneticGFN | $0.539\pm0.033$ | $0.476\pm0.008$ | $0.508\pm0.005$ | $0.441\pm0.054$ | $0.523\pm0.011$ |
| Grad | $0.513\pm0.007$ | $0.481\pm0.014$ | $0.510\pm0.007$ | $0.445\pm0.053$ | $0.525\pm0.033$ |
| COMs | $0.510\pm0.010$ | $0.478\pm0.014$ | $0.505\pm0.022$ | $0.411\pm0.007$ | $0.509\pm0.008$ |
| IOM | $0.520\pm0.009$ | $0.474\pm0.008$ | $0.500\pm0.013$ | $0.411\pm0.005$ | $0.519\pm0.042$ |
| RoMA | $0.512\pm0.010$ | $0.470\pm0.009$ | $0.512\pm0.032$ | $0.429\pm0.053$ | $0.512\pm0.013$ |
| Ensemble Proxy | $0.517\pm0.008$ | $0.479\pm0.010$ | $0.501\pm0.010$ | $0.414\pm0.006$ | $0.507\pm0.008$ |
| BIB | $0.514\pm0.010$ | $0.476\pm0.007$ | $0.497\pm0.006$ | $0.414\pm0.006$ | $0.505\pm0.009$ |
| BootGen | $0.544\pm0.032$ | $0.496\pm0.007$ | $0.524\pm0.007$ | $0.436\pm0.030$ | $0.545\pm0.063$ |
| ICT | $0.516\pm0.005$ | $0.476\pm0.006$ | $0.504\pm0.021$ | $0.410\pm0.005$ | $0.506\pm0.010$ |
| Tri-Mentoring | $0.529\pm0.038$ | $0.482\pm0.017$ | $0.511\pm0.019$ | $0.416\pm0.008$ | $0.513\pm0.009$ |
| RaM | $0.542\pm0.028$ | $0.488\pm0.007$ | $0.528\pm0.034$ | $0.424\pm0.008$ | $0.525\pm0.013$ |
| MolStitch (Ours) | $\mathbf{0.560\pm0.037}$ | $\mathbf{0.515\pm0.041}$ | $\mathbf{0.554\pm0.042}$ | $\mathbf{0.451\pm0.061}$ | $\mathbf{0.575\pm0.051}$ |

## C.2. R2 Performance Results for the Docking Score Optimization Task

**Results for R2 performance.** We present additional R2 performance results for the docking score optimization task in Table 8. Consistent with the findings in Table 7, our MolStitch framework demonstrated superior performance by achieving the lowest R2 indicator score compared to all competing methods.

*Table 8.* Experimental results on docking score optimization tasks under the full-offline setting using the R2 indicator.

| Target protein | parp1 | jak2 | braf | fa7 | 5ht1b |
|---|---|---|---|---|---|
| Method | R2($\downarrow$) | R2($\downarrow$) | R2($\downarrow$) | R2($\downarrow$) | R2($\downarrow$) |
| REINVENT | $1.426\pm0.090$ | $1.589\pm0.042$ | $1.497\pm0.044$ | $1.791\pm0.033$ | $1.454\pm0.054$ |
| AugMem | $1.374\pm0.163$ | $1.523\pm0.159$ | $1.471\pm0.044$ | $1.729\pm0.220$ | $1.421\pm0.064$ |
| Saturn | $1.376\pm0.053$ | $1.501\pm0.155$ | $1.420\pm0.176$ | $1.726\pm0.201$ | $1.350\pm0.139$ |
| GeneticGFN | $1.326\pm0.148$ | $1.589\pm0.039$ | $1.484\pm0.025$ | $1.701\pm0.228$ | $1.410\pm0.057$ |
| Grad | $1.422\pm0.032$ | $1.555\pm0.079$ | $1.461\pm0.036$ | $1.750\pm0.184$ | $1.401\pm0.134$ |
| COMs | $1.448\pm0.041$ | $1.568\pm0.089$ | $1.467\pm0.109$ | $1.816\pm0.031$ | $1.459\pm0.045$ |
| IOM | $1.402\pm0.041$ | $1.597\pm0.045$ | $1.488\pm0.070$ | $1.806\pm0.034$ | $1.421\pm0.160$ |
| RoMA | $1.431\pm0.053$ | $1.604\pm0.044$ | $1.434\pm0.153$ | $1.738\pm0.241$ | $1.449\pm0.058$ |
| Ensemble Proxy | $1.415\pm0.035$ | $1.568\pm0.062$ | $1.491\pm0.036$ | $1.800\pm0.028$ | $1.470\pm0.038$ |
| BIB | $1.425\pm0.045$ | $1.573\pm0.029$ | $1.500\pm0.034$ | $1.801\pm0.028$ | $1.478\pm0.043$ |
| BootGen | $1.320\pm0.136$ | $1.521\pm0.037$ | $1.420\pm0.030$ | $1.712\pm0.142$ | $1.336\pm0.184$ |
| ICT | $1.428\pm0.024$ | $1.591\pm0.029$ | $1.473\pm0.100$ | $1.810\pm0.028$ | $1.472\pm0.045$ |
| Tri-Mentoring | $1.373\pm0.155$ | $1.553\pm0.083$ | $1.428\pm0.098$ | $1.793\pm0.033$ | $1.443\pm0.045$ |
| RaM | $1.323\pm0.112$ | $1.546\pm0.036$ | $1.397\pm0.158$ | $1.778\pm0.037$ | $1.401\pm0.047$ |
| MolStitch (Ours) | $\mathbf{1.276\pm0.153}$ | $\mathbf{1.445\pm0.177}$ | $\mathbf{1.312\pm0.174}$ | $\mathbf{1.674\pm0.261}$ | $\mathbf{1.231\pm0.165}$ |

## C.3. Evaluating Molecular Optimization Methods Using Average Property Score of Top 10 and Top 100 Molecules

**Results for APS performance.** In our main results, we presented performance using the Hypervolume (HV) and R2 indicator metrics, which are widely regarded as the most appropriate evaluation metrics for multi-objective optimization tasks. However, within the molecular discovery community, the average property score (APS) is another commonly used metric, specifically tailored for assessing molecular optimization methods. To provide a more comprehensive assessment, we conducted additional experiments to report APS for various molecular optimization methods. The methods we evaluated include GraphGA (Jensen, 2019), which generates molecules by using rule-based crossover operations to combine features from parent molecules; MolGPT (Bagal et al., 2021), which is suitable for offline settings as it does not require oracle calls during molecule generation; DST (Fu et al., 2022), which leverages a proxy model to facilitate precise functional group editing; and REINVENT (Olivecrona et al., 2017), our backbone generative model, known for its robust performance in molecular optimization tasks. Additionally, we also considered AugMem (Guo & Schwaller, 2024a), a leading model in the PMO benchmark, Saturn (Guo & Schwaller, 2024b), which enhances sample efficiency in molecular design, and GeneticGFN (Kim et al., 2024a), which integrates GFlowNets with genetic algorithms to achieve state-of-the-art performance across various molecular optimization tasks. In this experiment, we calculated the APS of the top 10 and top 100 molecules generated by each method and reported the mean APS. As shown in Table 9, our MolStitch framework consistently outperformed all competing methods, even when evaluated with the molecule-specific metric. This result demonstrates the robustness and superiority of MolStitch across diverse evaluation criteria.

*Table 9.* Experimental results on molecular property optimization tasks under the full-offline setting using the average property score.

| Molecular objectives | GSK3$\beta$+JNK3 | | GSK3$\beta$+JNK3+QED | | GSK3$\beta$+JNK3+QED+SA | |
|---|---|---|---|---|---|---|
| Method | top10 ($\uparrow$) | top100 ($\uparrow$) | top10 ($\uparrow$) | top100 ($\uparrow$) | top10 ($\uparrow$) | top100 ($\uparrow$) |
| REINVENT | $0.515\pm0.076$ | $0.312\pm0.036$ | $0.464\pm0.018$ | $0.383\pm0.005$ | $0.564\pm0.018$ | $0.491\pm0.003$ |
| AugMem | $0.558\pm0.066$ | $0.374\pm0.036$ | $0.515\pm0.041$ | $0.407\pm0.010$ | $0.579\pm0.015$ | $0.505\pm0.005$ |
| MolGPT | $0.335\pm0.027$ | $0.199\pm0.005$ | $0.461\pm0.027$ | $0.380\pm0.005$ | $0.548\pm0.014$ | $0.485\pm0.002$ |
| GraphGA | $0.466\pm0.079$ | $0.313\pm0.058$ | $0.512\pm0.048$ | $0.415\pm0.012$ | $0.593\pm0.038$ | $0.507\pm0.010$ |
| DST | $0.456\pm0.058$ | $0.315\pm0.037$ | $0.531\pm0.059$ | $0.451\pm0.039$ | $0.601\pm0.027$ | $0.539\pm0.029$ |
| Saturn | $0.559\pm0.074$ | $0.358\pm0.037$ | $0.546\pm0.032$ | $0.443\pm0.041$ | $0.608\pm0.043$ | $0.513\pm0.041$ |
| GeneticGFN | $0.540\pm0.077$ | $0.379\pm0.078$ | $0.548\pm0.058$ | $0.451\pm0.051$ | $0.599\pm0.027$ | $0.524\pm0.029$ |
| MolStitch (Ours) | $\mathbf{0.627\pm0.056}$ | $\mathbf{0.432\pm0.039}$ | $\mathbf{0.591\pm0.040}$ | $\mathbf{0.468\pm0.016}$ | $\mathbf{0.671\pm0.041}$ | $\mathbf{0.564\pm0.024}$ |

## C.4. Semi-Offline Optimization

**Definition of semi-offline optimization.** Semi-offline optimization, also referred to as batch hybrid learning (Xiong et al., 2024), is an optimization approach that bridges the gap between offline and online optimization. In this semi-offline setting, models are trained on a combination of pre-existing offline datasets and periodically collected new data, enabling periodic updates without the need for continuous or real-time oracle queries. Unlike the full-offline setting, where the model is trained exclusively on a static offline dataset, the semi-offline setting allows for the periodic incorporation of new data in large batches, facilitating a more dynamic learning process. This semi-offline optimization is particularly useful in scenarios where obtaining new data in real-time is either too costly or logistically challenging, yet some level of interaction or adaptation to new data is beneficial.

**Semi-offline optimization in LLMs.** Semi-offline optimization has gained considerable attention in the field of large language models (LLMs). Several studies (Bai et al., 2022; Touvron et al., 2023) have implemented a strategy of iteratively applying the RLHF process on a weekly cadence. This involves periodically deploying updated RLHF models to interact with users or crowdworkers to collect new preference data. The models are then fine-tuned with this feedback on a regular schedule. Recently, (Xiong et al., 2024) further extended this approach by formulating it as a batch hybrid framework, establishing a more general setting for the hybrid learning process.

**Experimental setup for semi-offline optimization.** Motivated by these practical applications, we conducted additional experiments on MPO tasks under the semi-offline setting. We began by constructing an initial offline dataset using 5,000 oracle calls. In contrast to the full-offline setting, where all remaining 5,000 oracle calls were used for evaluation, the semi-offline setting employed a different allocation strategy. Specifically, we allocated 2,500 oracle calls for the periodic integration of new molecular data in large batches. This allocation enabled the generative model to iteratively update and adapt based on the newly acquired data. The remaining 2,500 oracle calls were reserved for the final evaluation.

*Table 10.* Experimental results on molecular property optimization tasks for the **semi-offline** setting, with the best values in bold.

| Molecular objectives | GSK3$\beta$+JNK3 | | GSK3$\beta$+JNK3+QED | | GSK3$\beta$+JNK3+QED+SA | |
|---|---|---|---|---|---|---|
| Method | HV($\uparrow$) | R2($\downarrow$) | HV($\uparrow$) | R2($\downarrow$) | HV($\uparrow$) | R2($\downarrow$) |
| REINVENT | 0.581$\pm$0.057 | 0.694$\pm$0.109 | 0.208$\pm$0.065 | 2.372$\pm$0.300 | 0.175$\pm$0.064 | 4.053$\pm$0.747 |
| AugMem | 0.636$\pm$0.063 | 0.602$\pm$0.113 | 0.348$\pm$0.075 | 1.888$\pm$0.237 | 0.292$\pm$0.087 | 3.225$\pm$0.650 |
| GraphGA | 0.521$\pm$0.084 | 0.819$\pm$0.136 | 0.392$\pm$0.102 | 1.623$\pm$0.277 | 0.265$\pm$0.080 | 3.493$\pm$0.537 |
| Saturn | 0.623$\pm$0.049 | 0.621$\pm$0.086 | 0.428$\pm$0.040 | 1.581$\pm$0.160 | 0.382$\pm$0.088 | 2.686$\pm$0.510 |
| GeneticGFN | 0.642$\pm$0.065 | 0.592$\pm$0.107 | 0.414$\pm$0.123 | 1.660$\pm$0.425 | 0.361$\pm$0.086 | 2.879$\pm$0.569 |
| Grad | 0.584$\pm$0.075 | 0.708$\pm$0.136 | 0.216$\pm$0.086 | 2.458$\pm$0.371 | 0.180$\pm$0.037 | 4.109$\pm$0.455 |
| COMs | 0.571$\pm$0.058 | 0.717$\pm$0.105 | 0.219$\pm$0.073 | 2.505$\pm$0.351 | 0.186$\pm$0.046 | 3.956$\pm$0.505 |
| IOM | 0.603$\pm$0.061 | 0.647$\pm$0.081 | 0.221$\pm$0.077 | 2.349$\pm$0.395 | 0.205$\pm$0.065 | 3.899$\pm$0.621 |
| RoMA | 0.588$\pm$0.067 | 0.680$\pm$0.109 | 0.215$\pm$0.070 | 2.414$\pm$0.258 | 0.180$\pm$0.036 | 4.105$\pm$0.414 |
| Ensemble Proxy | 0.602$\pm$0.084 | 0.648$\pm$0.146 | 0.227$\pm$0.071 | 2.435$\pm$0.332 | 0.216$\pm$0.069 | 3.730$\pm$0.573 |
| BIB | 0.563$\pm$0.066 | 0.713$\pm$0.122 | 0.215$\pm$0.078 | 2.440$\pm$0.388 | 0.189$\pm$0.070 | 4.062$\pm$0.735 |
| BootGen | 0.608$\pm$0.057 | 0.646$\pm$0.098 | 0.233$\pm$0.093 | 2.399$\pm$0.462 | 0.219$\pm$0.090 | 3.924$\pm$0.651 |
| ICT | 0.601$\pm$0.078 | 0.662$\pm$0.143 | 0.216$\pm$0.089 | 2.455$\pm$0.389 | 0.185$\pm$0.048 | 4.094$\pm$0.454 |
| Tri-Mentoring | 0.592$\pm$0.078 | 0.678$\pm$0.144 | 0.219$\pm$0.054 | 2.467$\pm$0.241 | 0.206$\pm$0.073 | 3.966$\pm$0.603 |
| MolStitch (Ours) | **0.689$\pm$0.041** | **0.514$\pm$0.073** | **0.539$\pm$0.045** | **1.238$\pm$0.157** | **0.493$\pm$0.050** | **2.014$\pm$0.202** |

*Table 11.* Performance of various preference optimization techniques for the **semi-offline** setting.

| Molecular objectives | GSK3$\beta$+JNK3 | | GSK3$\beta$+JNK3+QED | | GSK3$\beta$+JNK3+QED+SA | |
|---|---|---|---|---|---|---|
| Method | HV($\uparrow$) | R2($\downarrow$) | HV($\uparrow$) | R2($\downarrow$) | HV($\uparrow$) | R2($\downarrow$) |
| Baseline (REINVENT) | 0.462$\pm$0.133 | 0.921$\pm$0.259 | 0.196$\pm$0.083 | 2.646$\pm$0.327 | 0.168$\pm$0.046 | 3.969$\pm$0.664 |
| + StitchNet & RLHF | 0.675$\pm$0.059 | 0.526$\pm$0.091 | 0.448$\pm$0.066 | 1.540$\pm$0.221 | 0.383$\pm$0.082 | 2.647$\pm$0.463 |
| + StitchNet & DPO | 0.685$\pm$0.047 | 0.520$\pm$0.083 | 0.507$\pm$0.078 | 1.342$\pm$0.221 | 0.447$\pm$0.060 | 2.320$\pm$0.331 |
| + StitchNet & IPO | 0.681$\pm$0.042 | 0.521$\pm$0.069 | 0.527$\pm$0.055 | 1.256$\pm$0.133 | 0.462$\pm$0.055 | 2.187$\pm$0.299 |
| + StitchNet & IPO & RP (Ours) | **0.689$\pm$0.041** | **0.514$\pm$0.073** | **0.539$\pm$0.045** | **1.238$\pm$0.157** | **0.493$\pm$0.050** | **2.014$\pm$0.202** |

**Results for semi-offline optimization.** As illustrated in Table 10, our MolStitch framework consistently outperformed all competing methods under the semi-offline setting. Notably, we observed a general improvement in performance compared to the full-offline setting, as shown in Table 1 of the main manuscript. This finding highlights the benefits of incorporating periodic new data, as it enables the generative model to be fine-tuned and trained on newly acquired samples, thereby further enhancing its optimization capabilities. Consistent with the trends observed in the full-offline setting, Saturn and GeneticGFN maintained strong performance among competing methods, highlighting the effectiveness of genetic algorithms in offline MOMO. Their success could be attributed to the inherent strengths of genetic algorithms in maintaining population diversity and effectively exploring the Pareto front through crossover operations. This finding aligns with our framework, which employs a mechanism analogous to crossover, but with the added advantage of incorporating chemical feedback. Additionally, we conducted experiments for preference optimization techniques under the semi-offline setting, as depicted in Table 11. The trends observed were similar to those in the full-offline setting, our MolStitch consistently achieved the highest performance among all competing methods. While RLHF performed well on the two-objective scenario, its performance declined significantly as the number of objectives increased. Both DPO and IPO demonstrated strong performance, with IPO showing a slight edge over DPO.

## C.5. Evaluating Mamba and GFlowNets as Additional Backbone Models

**Various backbone models.** In this work, we chose REINVENT as our backbone generative model due to its widespread use and reputation as one of the top-performing models for various molecular optimization tasks. However, as previously mentioned, Saturn and GeneticGFN demonstrated strong performance in numerous offline MOMO experiments. Since these methods utilized Mamba and GFlowNets as their respective backbone models, we conducted additional experiments using Mamba and GFlowNets as the backbone generative model for our MolStitch framework.

**Results for backbone models.** As illustrated in Table 12, we report the performance of each backbone generative model—REINVENT, Mamba, and GFlowNets—on MPO tasks under the full-offline setting, alongside the performance of integrating either our rank-based proxy or MolStitch framework with each backbone model (e.g., REINVENT + MolStitch). Similarly, Table 13 presents the performance of the backbone generative models and their respective integrations with MolStitch under the semi-offline setting. As shown in Table 12, both the rank-based proxy and the MolStitch framework provide performance improvements across various generative models. However, the integration with the rank-based proxy still falls short compared to the full MolStitch framework, emphasizing the additional benefits brought by StitchNet and priority sampling. Notably, Mamba + MolStitch and GFlowNets + MolStitch outperformed REINVENT + MolStitch in both three-objective and four-objective scenarios. This superior performance could be attributed to the greater capacity of Mamba and GFlowNets to manage the increased complexity associated with optimizing multiple objectives beyond two. Overall, the consistent performance improvements across different backbone generative models under both full-offline and semi-offline settings demonstrate the robustness and versatility of our MolStitch. Moreover, these additional results highlight the MolStitch's ability to seamlessly integrate with a range of backbone models, demonstrating its adaptability and robustness beyond a single model architecture.

*Table 12.* Performance comparison of different generative models on molecular property optimization tasks under the **full-offline** setting.

| Molecular objectives | GSK3$\beta$+JNK3 | | GSK3$\beta$+JNK3+QED | | GSK3$\beta$+JNK3+QED+SA | |
|---|---|---|---|---|---|---|
| Method | HV($\uparrow$) | R2($\downarrow$) | HV($\uparrow$) | R2($\downarrow$) | HV($\uparrow$) | R2($\downarrow$) |
| REINVENT | 0.462$\pm$0.133 | 0.921$\pm$0.259 | 0.196$\pm$0.083 | 2.646$\pm$0.327 | 0.168$\pm$0.046 | 3.969$\pm$0.664 |
| + Rank-based Proxy | 0.545$\pm$0.063 | 0.773$\pm$0.120 | 0.319$\pm$0.059 | 1.928$\pm$0.314 | 0.251$\pm$0.084 | 3.504$\pm$0.634 |
| + MolStitch (Ours) | 0.579$\pm$0.070 | 0.698$\pm$0.128 | 0.403$\pm$0.065 | 1.649$\pm$0.259 | 0.352$\pm$0.080 | 2.953$\pm$0.571 |
| Mamba | 0.531$\pm$0.087 | 0.785$\pm$0.159 | 0.293$\pm$0.058 | 1.977$\pm$0.280 | 0.281$\pm$0.058 | 3.339$\pm$0.280 |
| + Rank-based Proxy | 0.538$\pm$0.068 | 0.758$\pm$0.105 | 0.327$\pm$0.100 | 1.946$\pm$0.404 | 0.281$\pm$0.072 | 3.317$\pm$0.486 |
| + MolStitch (Ours) | 0.544$\pm$0.071 | 0.761$\pm$0.128 | 0.407$\pm$0.077 | 1.617$\pm$0.199 | 0.361$\pm$0.063 | 2.893$\pm$0.424 |
| GFlowNets | 0.482$\pm$0.073 | 0.869$\pm$0.117 | 0.309$\pm$0.087 | 1.990$\pm$0.365 | 0.237$\pm$0.066 | 3.630$\pm$0.453 |
| + Rank-based Proxy | 0.522$\pm$0.040 | 0.805$\pm$0.085 | 0.364$\pm$0.070 | 1.809$\pm$0.305 | 0.323$\pm$0.054 | 2.953$\pm$0.304 |
| + MolStitch (Ours) | 0.525$\pm$0.063 | 0.770$\pm$0.111 | 0.415$\pm$0.087 | 1.685$\pm$0.343 | 0.366$\pm$0.088 | 2.708$\pm$0.652 |

*Table 13.* Performance comparison of different generative models on molecular property optimization tasks under the **semi-offline** setting.

| Molecular objectives | GSK3$\beta$+JNK3 | | GSK3$\beta$+JNK3+QED | | GSK3$\beta$+JNK3+QED+SA | |
|---|---|---|---|---|---|---|
| Method | HV($\uparrow$) | R2($\downarrow$) | HV($\uparrow$) | R2($\downarrow$) | HV($\uparrow$) | R2($\downarrow$) |
| REINVENT | 0.581$\pm$0.057 | 0.694$\pm$0.109 | 0.208$\pm$0.065 | 2.372$\pm$0.300 | 0.175$\pm$0.064 | 4.053$\pm$0.747 |
| + MolStitch (Ours) | 0.689$\pm$0.041 | 0.514$\pm$0.073 | 0.539$\pm$0.045 | 1.238$\pm$0.157 | 0.493$\pm$0.050 | 2.014$\pm$0.202 |
| Mamba | 0.623$\pm$0.049 | 0.621$\pm$0.086 | 0.428$\pm$0.040 | 1.581$\pm$0.160 | 0.382$\pm$0.088 | 2.686$\pm$0.510 |
| + MolStitch (Ours) | 0.653$\pm$0.046 | 0.580$\pm$0.090 | 0.485$\pm$0.054 | 1.430$\pm$0.196 | 0.434$\pm$0.044 | 2.385$\pm$0.176 |
| GFlowNets | 0.642$\pm$0.065 | 0.592$\pm$0.107 | 0.414$\pm$0.123 | 1.660$\pm$0.425 | 0.361$\pm$0.086 | 2.879$\pm$0.569 |
| + MolStitch (Ours) | 0.658$\pm$0.068 | 0.563$\pm$0.108 | 0.579$\pm$0.041 | 1.137$\pm$0.130 | 0.482$\pm$0.076 | 2.181$\pm$0.438 |

## C.6. Comprehensive Ablation Study for the Molecular Property Optimization (MPO) Task

**Comprehensive Ablation Study Results.** In the main manuscript, we presented ablation study results focusing only on the four-objective (GSK3$\beta$+JNK3+QED+SA) scenario due to space constraints. Here, we provide a complete ablation study, including two- and three-objective scenarios, for a thorough evaluation. As shown in Table 14, the results clearly illustrate the effectiveness of our framework's key components: rank-based proxy (RP), StitchNet (SN), and priority sampling (PS). The comprehensive results consistently show improvements across all objective scenarios, underscoring the critical role of each component in driving superior performance for the offline MOMO problem.

*Table 14.* An ablation study for key components in our framework: Rank-based Proxy (RP), StitchNet (SN), and Priority Sampling (PS).

| Ablation | | | GSK3$\beta$+JNK3 | | GSK3$\beta$+JNK3+QED | | GSK3$\beta$+JNK3+QED+SA | |
|---|---|---|---|---|---|---|---|---|
| RP | SN | PS | HV($\uparrow$) | R2($\downarrow$) | HV($\uparrow$) | R2($\downarrow$) | HV($\uparrow$) | R2($\downarrow$) |
| - | - | - | $0.494 \pm 0.058$ | $0.857 \pm 0.126$ | $0.205 \pm 0.045$ | $2.502 \pm 0.231$ | $0.171 \pm 0.026$ | $4.176 \pm 0.319$ |
| - | ✔ | - | $0.513 \pm 0.073$ | $0.780 \pm 0.106$ | $0.269 \pm 0.081$ | $2.183 \pm 0.318$ | $0.193 \pm 0.053$ | $4.134 \pm 0.502$ |
| - | ✔ | ✔ | $0.505 \pm 0.049$ | $0.824 \pm 0.084$ | $0.277 \pm 0.083$ | $2.195 \pm 0.357$ | $0.220 \pm 0.054$ | $3.835 \pm 0.483$ |
| ✔ | - | - | $0.545 \pm 0.063$ | $0.773 \pm 0.120$ | $0.319 \pm 0.059$ | $1.928 \pm 0.314$ | $0.251 \pm 0.084$ | $3.504 \pm 0.634$ |
| ✔ | ✔ | - | $0.573 \pm 0.078$ | $\mathbf{0.688 \pm 0.138}$ | $0.337 \pm 0.068$ | $1.967 \pm 0.311$ | $0.289 \pm 0.096$ | $3.317 \pm 0.713$ |
| ✔ | ✔ | ✔ | $\mathbf{0.579 \pm 0.070}$ | $0.698 \pm 0.128$ | $\mathbf{0.403 \pm 0.065}$ | $\mathbf{1.649 \pm 0.259}$ | $\mathbf{0.352 \pm 0.080}$ | $\mathbf{2.953 \pm 0.571}$ |

## C.7. Comprehensive Results for Performance Comparison of Data Augmentation Techniques in Offline MOMO

**Comprehensive Evaluation of Data Augmentation Techniques.** In the main manuscript, we focused exclusively on the four-objective scenario (GSK3$\beta$+JNK3+QED+SA) due to space limitations. Here, we extend the analysis to include two- and three-objective scenarios, offering a broader perspective on the performance of various data augmentation techniques. As summarized in Table 15, the findings clearly establish StitchNet as the most effective technique, consistently outperforming alternative approaches and demonstrating its ability to enhance performance across all objective scenarios in the offline MOMO problem.

*Table 15.* Performance comparison of different data augmentation techniques in offline MOMO.

| Molecular objectives | GSK3$\beta$+JNK3 | | GSK3$\beta$+JNK3+QED | | GSK3$\beta$+JNK3+QED+SA | |
|---|---|---|---|---|---|---|
| Augmentation | HV($\uparrow$) | R2($\downarrow$) | HV($\uparrow$) | R2($\downarrow$) | HV($\uparrow$) | R2($\downarrow$) |
| Baseline (REINVENT) | $0.462 \pm 0.133$ | $0.921 \pm 0.259$ | $0.196 \pm 0.083$ | $2.646 \pm 0.327$ | $0.168 \pm 0.046$ | $3.969 \pm 0.664$ |
| + Stochastic sampling | $0.545 \pm 0.063$ | $0.773 \pm 0.120$ | $0.319 \pm 0.059$ | $1.928 \pm 0.314$ | $0.251 \pm 0.084$ | $3.504 \pm 0.634$ |
| + Crossover operator | $0.540 \pm 0.088$ | $0.790 \pm 0.181$ | $0.367 \pm 0.062$ | $1.793 \pm 0.245$ | $0.302 \pm 0.072$ | $3.110 \pm 0.479$ |
| + StitchNet (Ours) | $\mathbf{0.579 \pm 0.070}$ | $\mathbf{0.698 \pm 0.128}$ | $\mathbf{0.403 \pm 0.065}$ | $\mathbf{1.649 \pm 0.259}$ | $\mathbf{0.352 \pm 0.080}$ | $\mathbf{2.953 \pm 0.571}$ |

## C.8. Comprehensive Results for Various Preference Optimization Techniques in Offline MOMO

**Expanded Evaluation of Preference Optimization Techniques.** The main manuscript presented results for the four-objective scenario (GSK3$\beta$+JNK3+QED+SA) to maintain a concise focus. In this section, we broaden the scope to include evaluations for two- and three-objective scenarios, offering a more comprehensive assessment of preference optimization methods. As depicted in Table 16, our proposed approach (constructing the separate rank-based proxy (RP) and updating the model separately based on the proxy feedback) demonstrates a clear advantage, consistently outperforming other preference optimization techniques. These results underline its effectiveness and reliability across varying objective scenarios within the offline MOMO problem.

*Table 16.* Performance comparison of various preference optimization techniques in offline MOMO.

| Molecular objectives | GSK3$\beta$+JNK3 | | GSK3$\beta$+JNK3+QED | | GSK3$\beta$+JNK3+QED+SA | |
|---|---|---|---|---|---|---|
| Method | HV($\uparrow$) | R2($\downarrow$) | HV($\uparrow$) | R2($\downarrow$) | HV($\uparrow$) | R2($\downarrow$) |
| Baseline (REINVENT) | $0.462 \pm 0.133$ | $0.921 \pm 0.259$ | $0.196 \pm 0.083$ | $2.646 \pm 0.327$ | $0.168 \pm 0.046$ | $3.969 \pm 0.664$ |
| + StitchNet & RLHF | $0.561 \pm 0.055$ | $0.742 \pm 0.098$ | $0.303 \pm 0.087$ | $2.012 \pm 0.318$ | $0.232 \pm 0.071$ | $3.715 \pm 0.611$ |
| + StitchNet & DPO | $0.557 \pm 0.094$ | $0.747 \pm 0.174$ | $0.363 \pm 0.069$ | $1.843 \pm 0.271$ | $0.327 \pm 0.081$ | $3.015 \pm 0.493$ |
| + StitchNet & IPO | $0.552 \pm 0.056$ | $0.746 \pm 0.106$ | $0.385 \pm 0.062$ | $1.755 \pm 0.232$ | $0.344 \pm 0.082$ | $2.955 \pm 0.533$ |
| + StitchNet & IPO & RP (Ours) | $\mathbf{0.579 \pm 0.070}$ | $\mathbf{0.698 \pm 0.128}$ | $\mathbf{0.403 \pm 0.065}$ | $\mathbf{1.649 \pm 0.259}$ | $\mathbf{0.352 \pm 0.080}$ | $\mathbf{2.953 \pm 0.571}$ |

# D. Detailed Analysis of Rank-based Proxy

In this section, we provide an in-depth analysis of both rank-based and score-based proxies. Our study suggests that the formulation of rank-based proxy simplifies the proxy's task, thereby enabling it to deliver more reliable feedback to the generative model. To further explore this, we delve deeper into the performance of each proxy type, examining whether the rank-based proxy truly surpasses the score-based proxy in handling complex multi-objective molecular optimization tasks.

**Proxy models.** In the context of utilizing proxy models, they offer distinct advantages, but they also present notable challenges. Specifically, Grad is built upon REINVENT and incorporates a vanilla score-based proxy that directly approximates objective scores. As shown in Table 1 of our main manuscript, while Grad outperforms the baseline REINVENT, its performance gains gradually diminish as the number of objectives increases from two to four. This suggests that with the rise in the number of objectives, the problem complexity increases, causing the vanilla proxy to struggle to accurately approximate the objective scores. In contrast, our framework demonstrates particularly strong performance in the three and four objective scenarios, which highlights the effectiveness of reformulating the proxy model's task from direct property score regression to pairwise classification.

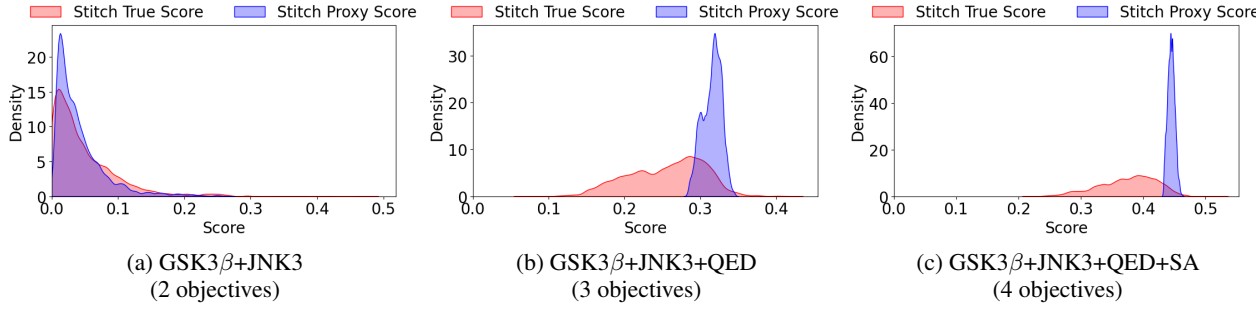

(a) GSK3$\beta$+JNK3
(2 objectives)

(b) GSK3$\beta$+JNK3+QED
(3 objectives)

(c) GSK3$\beta$+JNK3+QED+SA
(4 objectives)

*Figure 8.* Distribution comparison of true objective scores (red) and score-based proxy model predictions (blue) for stitched molecules across varying numbers of objectives. As the number of objectives increases, the score-based proxy model's predictions show less variability and exhibit a sharper central peak, failing to accurately represent the true score distribution.

**Score-based proxy.** As shown in Figure 8, we visualize the distribution of the true scores for the stitched molecules alongside the predicted scores from the score-based proxy model. Compared to the distribution of true objective scores, the predictions made by the score-based proxy model are significantly more confined to a narrow range. This issue becomes more pronounced as the number of objectives increases, with the score-based proxy model's predictions showing even less variability and a stronger central peak, failing to represent the true score distribution accurately. Therefore, this result indicates that the score-based proxy model fails to provide meaningful feedback to the generative model, potentially leading to suboptimal optimization. To address these limitations, we propose a rank-based proxy model that learns the relative ranking between pairs of molecules based on desired properties, determining which molecule is more favorable. This approach bypasses the direct approximation of true objective scores and instead focuses on ranking relationships, providing more reliable feedback signals for the generative model.

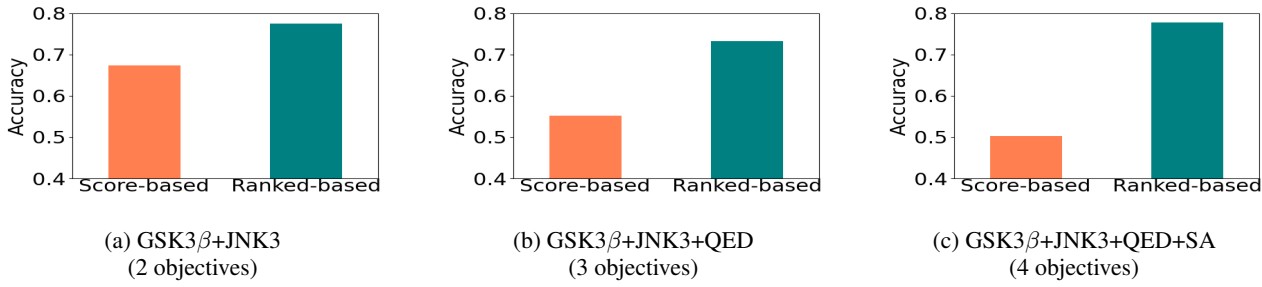

(a) GSK3$\beta$+JNK3
(2 objectives)

(b) GSK3$\beta$+JNK3+QED
(3 objectives)

(c) GSK3$\beta$+JNK3+QED+SA
(4 objectives)

*Figure 9.* Accuracy comparison of score-based and rank-based proxy models in predicting the ranking of randomly selected molecule pairs across varying numbers of objectives: (a) 2 objectives, (b) 3 objectives, and (c) 4 objectives.

**Rank-based proxy.** To demonstrate the effectiveness of the rank-based proxy, we compare the performance of score-based

and rank-based proxy models in predicting the rank of randomly selected pairs of molecules. As illustrated in Figure 9, the rank-based model consistently outperforms the score-based model across all scenarios with varying objectives. This performance gap widens as the number of objectives increases, with the rank-based model maintaining relatively high accuracy even with four objectives, while the score-based model's accuracy drops significantly. These findings validate the superiority of the rank-based proxy over the score-based proxy in effectively addressing the complexities of offline MOMO.

## E. Additional Experiments on Multiple Proxies

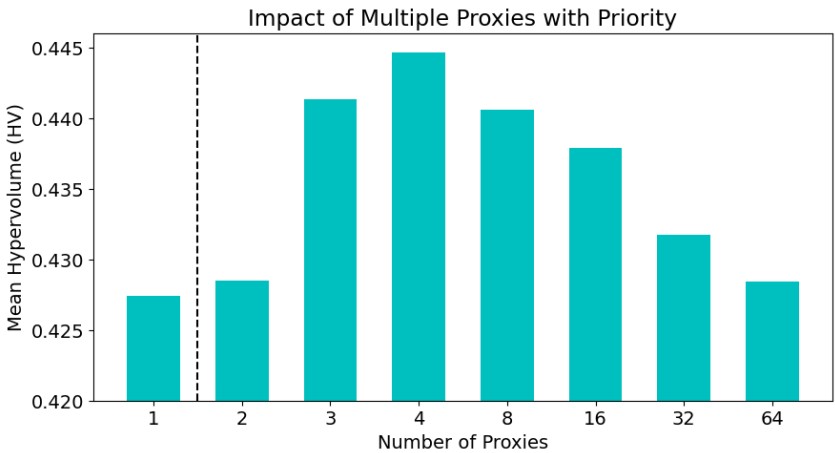

*Figure 10.* An illustration of the impact of employing multiple proxies with priority sampling in our framework. The evaluation metric is the mean hypervolume across all numbers of objectives for the MPO task under the full-offline setting. The results demonstrate that the optimal configuration for our framework is four proxies, achieving the best performance before a decline due to redundancy.

**Motivation for multiple proxies.** In this section, we provide a detailed process and analysis of employing multiple proxy models within our framework. The motivation for experimenting with multiple proxies arises from observations in both offline MBO and LLM research. In offline MBO, methods employing multiple proxies—such as Ensemble Proxy, ICT, and Tri-Mentoring—generally outperform single proxy methods like Grad. This finding aligns with a recent study in large language models (LLMs) (Chakraborty et al., 2024), which highlights the drawbacks of using a single reward model to represent human preferences. Researchers note that human preferences are inherently diverse, and a single model often fails to reflect this variability, leading to biased or suboptimal outcomes. To address this, they propose using multiple reward models to capture a broader spectrum of preference distributions, thereby enhancing alignment with diverse human judgments.

**Setup for multiple proxies.** Inspired by these insights, we enhance our proxy model by incorporating ensemble learning through the use of multiple proxies. In the context of LLMs, preferences reflect human sentiments, opinions, or judgments about desirable outputs. In molecular optimization, however, preference represents the relative importance or priority of each objective within the optimization process. To effectively capture this diversity of priorities, we employ priority sampling for each proxy model, allowing them to prioritize objectives differently according to their assigned importance. Specifically, each proxy receives weight configurations sampled from a Dirichlet distribution, enabling it to focus more on certain objectives than others. As a result, each proxy can determine which molecule in a given pair is superior from its unique perspective. These individual assessments are then combined using a majority voting strategy, providing a comprehensive evaluation of molecules from multiple viewpoints to determine the overall superior molecule.

**Results for multiple proxies.** As demonstrated in Figure 10, the performance of our framework increases with the number of proxies, peaking at four before gradually declining thereafter. The observed decline in performance beyond four proxies can be attributed to the balance between ensemble diversity and redundancy. For an ensemble to be effective, the individual proxy models should be diverse, each providing unique insights into molecule evaluation. While adding proxy models up to a certain point enhances performance by capturing a wider range of priorities, adding too many proxies can introduce redundancy. Beyond the optimal number, additional proxies may become similar to existing ones, offering little new information and potentially amplifying common errors. In addition, with a large number of proxies, majority voting can

overlook minority opinions, reducing ensemble diversity and neglecting smaller yet significant priorities. Lastly, note that all configuration settings—whether employing a single proxy or multiple proxies—outperform all competing methods, underscoring the effectiveness of our framework.

**Analysis for multiple proxies.** One might question how majority voting works with an even number of proxies, as it could lead to a tie. In such cases where the proxies are evenly split in their assessments (e.g., two proxies favor a molecule while two do not), we interpret this as an indication of uncertainty or difficulty in evaluating the molecule. Rather than making a hasty decision that could misguide the optimization process, we choose to pass and skip these uncertain molecules. This approach ensures that only molecules with a higher degree of consensus among the proxies influence the optimization, enhancing the reliability of the feedback signals. The results also validate that employing four proxies surpasses the performance of using three proxies. In the four-proxy setup, a molecule must receive at least three favorable votes to be considered superior, raising the confidence threshold compared to the two-out-of-three votes required in the three-proxy setup. The stricter criterion in the four-proxy setup leads to more reliable and accurate feedback, contributing to improved optimization performance.

## F. Additional Analysis on Molecular Diversity of StitchNet

**Additional diversity metrics.** In the manuscript, we compared the diversity achieved by StitchNet with that of its data augmentation counterpart, stochastic sampling. We found that StitchNet exhibits greater diversity, which we attribute to its crossover-like mechanism that enables the generation of considerably more diverse molecules than stochastic sampling. To further investigate the diversity achieved by StitchNet, we propose the use of additional diversity metrics to provide a more comprehensive analysis from multiple perspectives. To quantify the diversity of the augmented molecules, we employed the inverse of the Tanimoto similarity (Bender & Glen, 2004). Specifically, we calculated the maximum Tanimoto similarity for each augmented molecule with respect to all other augmented molecules, then averaged these values and subtracted the result from 1, which we term the '`Within augmented`' diversity metric. In addition, we computed the maximum Tanimoto similarity between each augmented molecule and the molecules in the offline dataset, similarly subtracting this value from 1 to derive the '`Against offline dataset`' diversity metric.

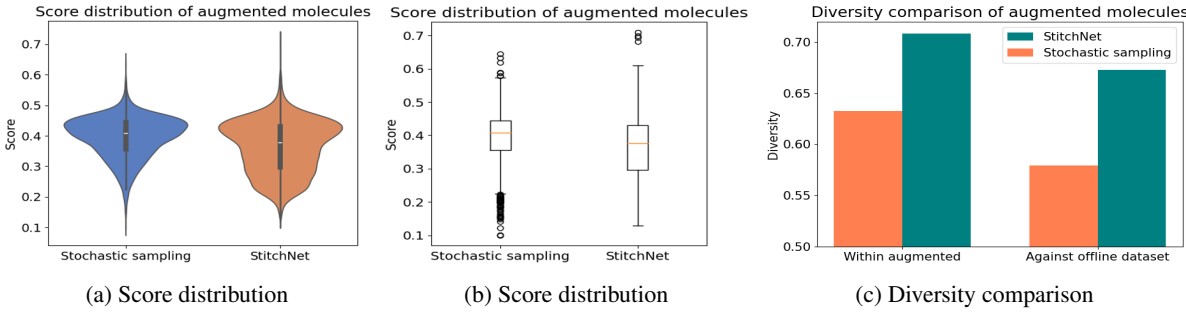

|  |  |  |
|:---:|:---:|:---:|
| (a) Score distribution | (b) Score distribution | (c) Diversity comparison |

*Figure 11.* Diversity analysis of augmented molecules generated by StitchNet and its data augmentation counterpart, stochastic sampling. The results demonstrate the superior capability of StitchNet in generating a diverse and novel set of augmented molecules.

**Additional diversity results.** The results in Figure 11 (a-b) demonstrate that StitchNet produces a much broader and more varied score distribution compared to stochastic sampling. This broader distribution highlights the StitchNet's capability to generate augmented molecules with higher diversity, thereby enriching the fine-tuning process for the generative model. Moreover, the additional diversity metrics further emphasize the advantages of StitchNet over stochastic sampling. As shown in Figure 11 (c), StitchNet consistently achieves higher values in both the `Within augmented` and `Against offline dataset` diversity metrics. This indicates that augmented molecules generated by StitchNet not only show greater diversity among themselves but also display more novelty in comparison to the molecules present in the offline dataset. Additionally, we evaluated the final molecules produced by the generative model fine-tuned with StitchNet against those fine-tuned with stochastic sampling, as shown in Figure 12. The generative model fine-tuned with StitchNet outperforms its counterpart in every aspect: (a) score distribution of final molecules, (b) diversity metrics for both `Within augmented` and `Against offline dataset`, and (c) diversity based on Bemis-Murcko (BM) scaffolds and Carbon Skeletons (CS) (Bemis & Murcko, 1996).

**BM scaffolds & Carbon Skeletons.** BM scaffolds are an essential tool for breaking down organic molecules to identify

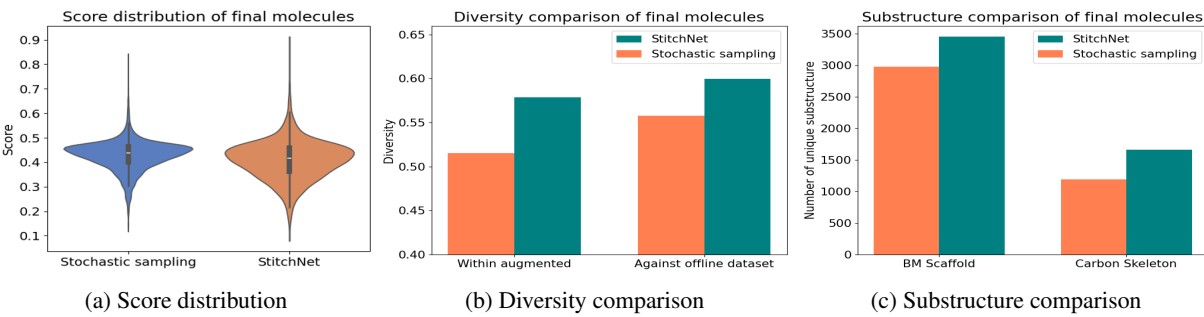

(a) Score distribution      (b) Diversity comparison      (c) Substructure comparison

*Figure 12.* Diversity analysis of final molecules produced by the generative model fine-tuned with StitchNet and with stochastic sampling. The results demonstrate that the generative model fine-tuned with StitchNet consistently achieves higher diversity and performance across all diversity metrics.

their core chemical substructures. As shown in Figure 13 (a), BM scaffolds simplify molecules by removing side chains while preserving the core substructures—such as ring systems and connecting linkers—representing the molecular backbone. This approach allows for a more effective quantitative assessment of structural diversity by comparing the backbones of different molecules. Another method for assessing structural diversity is through CS, which describe various configurations of carbon atoms, including straightline, branching, and ring, as depicted in 13 (b). In particular, straightline skeletons consist of carbon atoms connected in a linear arrangement, while branching skeletons contain side chains that extend from the main carbon chain that potentially affects the molecule's reactivity and interactions with biological targets. Ring skeletons are closed loops of carbon atoms, commonly found in biologically active compounds. Both BM scaffolds and CS serve as complementary methods for simplifying and categorizing molecular structures to better understand their properties and interactions. While BM scaffolds focus on the core substructures by removing side chains and functional groups, CS emphasizes the basic carbon framework of a given molecule. By incorporating both approaches in our analysis, we believe we can conduct a more comprehensive evaluation of structural diversity across the generated molecules.

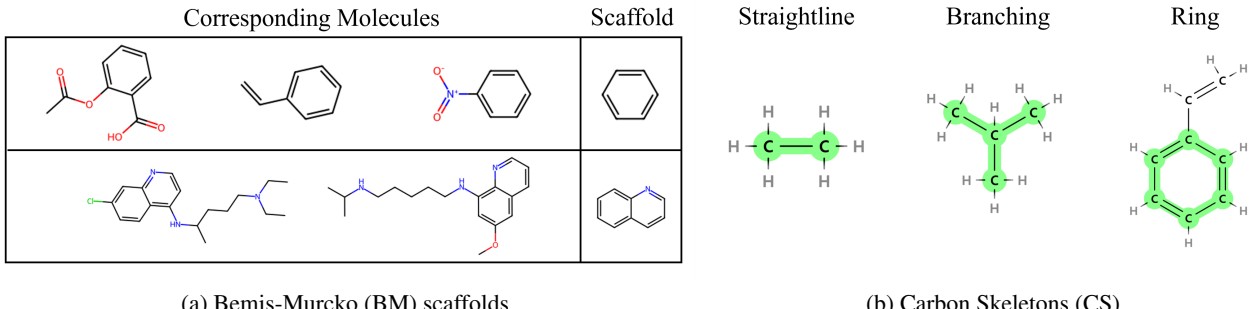

(a) Bemis-Murcko (BM) scaffolds      (b) Carbon Skeletons (CS)

*Figure 13.* Visual representations of (a) the Bemis-Murcko scaffolds and (b) the Carbon Skeletons.

# G. Pre-Training Process for the Generative Model (REINVENT)

**(a) Molecule generation process**

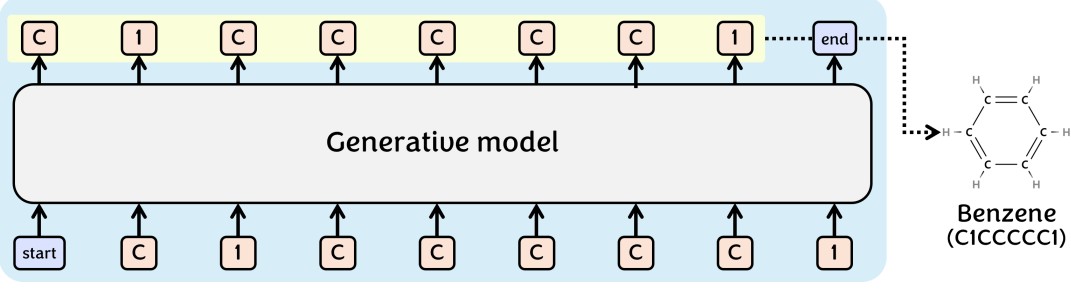

**(b) Pre-training process for the generative model**

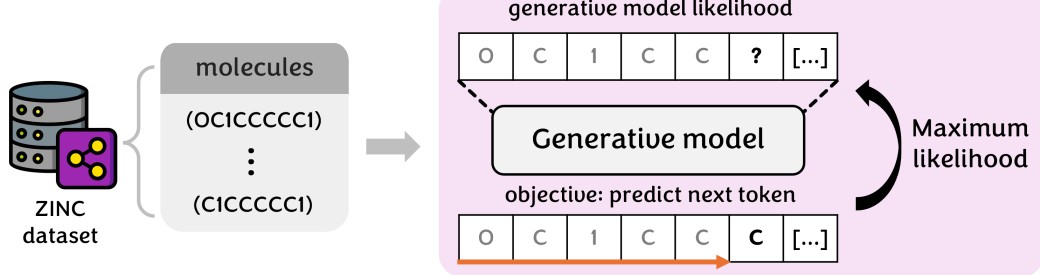

*Figure 14.* (a) The generative model produces molecules in SMILES format using an auto-regressive approach. (b) During the pre-training stage, molecules from the ZINC dataset are used as ground truth labels. The generative model is then updated through maximum likelihood approach to maximize the probability of the correct next molecular token (component) given the preceding sequence.

In this section, we present an illustration of the molecule generation process and describe the pre-training process for the generative model. As depicted in Figure 14 (a), which exemplifies the generation of a benzene molecule, the generative model produces molecules in an auto-regressive manner, similar to how language models generate sentences sequentially. Specifically, the generative model produces molecules in SMILES format, where each token corresponds to an atom or bond. The generation process begins with an initial token, and the model predicts the subsequent token based on the previously generated sequence, continuing this process until the complete molecule is formed.

Moving on to the pre-training process for the generative model, we employ an approach analogous to the next-token prediction loss used in language model training, as shown in Figure 14 (b). Specifically, the model is trained using the maximum likelihood approach, where molecules sampled from the ZINC dataset serve as ground truth labels. The objective of this pre-training process is to maximize the likelihood of accurately predicting the next molecular token (component) based on the preceding sequence. The cross-entropy loss is employed to measure the difference between the predicted probability distribution and the true distribution of the next token, guiding the model to learn the correct sequence of molecular components and generate chemically valid molecules.

Building upon the pre-training of our generative model using the ZINC dataset, we now detail the specific generative model employed in our framework. As mentioned in the main manuscript, REINVENT was selected as our main generative model due to its widespread adoption and proven effectiveness in various molecular optimization tasks. In REINVENT, the molecule optimization process is formulated as a Markov decision process, utilizing the RL algorithm to generate molecules based on a given scoring (reward) function. The training architecture of REINVENT comprises two distinct policy models: the prior model and the agent model. The prior model, denoted as $G_{ref}$, is a pre-trained reference model that encodes chemical grammar to ensure the chemical validity of the generated molecules, as depicted in Figure 14 (b). The agent model $G_\phi$ is initialized from the prior model and serves as the main policy that aims to maximize the reward score associated with the desired molecular properties, while not deviating too far from the prior model. The training objective for the agent model

can be defined as:

$$\mathcal{L}_{\text{agent}}(\phi) = \mathbb{E}_{m \sim D} \left[ \left( -\log G_\phi(m) + \log G_{\text{ref}}(m) + R(m) \right)^2 \right],$$

where $R(m)$ represents the reward score for molecule $m$ within the offline dataset $\mathcal{D}$. Note that this work addresses the offline MOMO problem, where the offline dataset comprises pairs of molecules and their corresponding property (objective) scores. Therefore, these property scores can be used as reward scores for training the agent model. To sum up, this loss function $\mathcal{L}_{\text{agent}}(\phi)$ guides $G_\phi(m)$ to maximize the reward $R(m)$ while aligning with $G_{\text{ref}}(m)$. For a detailed derivation and background of this REINVENT loss function, please refer to prior studies (Olivecrona et al., 2017; Guo & Schwaller, 2024a). After completing the initial training phase on the offline dataset, the agent model is fine-tuned to further enhance its performance beyond the constraints of the offline dataset. This fine-tuning process involves optimizing the agent model with stitched molecules using preference optimization techniques, as described in Equation 14 of the main manuscript.

## H. Preference Optimization Techniques for the Generative Model

### H.1. From Initial Loss Formulation to DPO-like Loss Formulation

As mentioned in Subsection 3.3, the initial loss formulation for the generative model is as follows:

$$\mathcal{L}_{\text{gen}}(\phi) = -\mathbb{E}_{(\bar{m}_w, \bar{m}_l) \sim \mathcal{B}} \left[ \log G_\phi(\bar{m}_w) - \log G_\phi(\bar{m}_l) \right] + \beta \cdot \mathbb{D}_{\text{KL}}(G_\phi \| G_{\text{ref}}).$$

This loss equation consists of two key components: the first term represents the difference in log-likelihoods between generating the winning molecule $G_\phi(\bar{m}_w)$ and the losing molecule $G_\phi(\bar{m}_l)$, while the second part introduces a KL divergence between the current generative model $G_\phi$ and the reference model $G_{\text{ref}}$. Following (Tang et al., 2024), the KL divergence term can be defined as:

$$\mathbb{D}_{\text{KL}}(G_\phi \| G_{\text{ref}}) := \mathbb{E}_{(\bar{m}) \sim \mathcal{B}} \left[ \log \frac{G_\phi(\bar{m})}{G_{\text{ref}}(\bar{m})} \right].$$

Since we are focusing on a pairwise comparison between winning and losing molecules, $(\bar{m}_w, \bar{m}_l)$, it is possible to apply the KL divergence to each component and simply the loss function as follows:

$$\mathcal{L}_{\text{gen}}(\phi) := -\mathbb{E}_{(\bar{m}_w, \bar{m}_l) \sim \mathcal{B}} \left[ \beta \left( \log \frac{G_\phi(\bar{m}_w)}{G_{\text{ref}}(\bar{m}_w)} - \log \frac{G_\phi(\bar{m}_l)}{G_{\text{ref}}(\bar{m}_l)} \right) \right].$$

At this point, we can leverage the notion of the Bradley-Terry model that the log odds of one item winning over another (in our case, $m_w$ over $m_l$) can also be written as:

$$\log \frac{G_\phi(\bar{m}_w)}{G_\phi(\bar{m}_l)},$$

and this log-odds can be converted into a probability using the sigmoid function $\sigma(\cdot)$, defined as:

$$\sigma(x) = \frac{1}{1 + e^{-x}}.$$

To incorporate the probabilistic nature of the comparison, we can now apply the sigmoid function to a combination of the two log-odds from $G_\phi$ and $G_{\text{ref}}$ as follows:

$$\sigma \left[ \beta \left( \log \frac{G_\phi(\bar{m}_w)}{G_{\text{ref}}(\bar{m}_w)} - \log \frac{G_\phi(\bar{m}_l)}{G_{\text{ref}}(\bar{m}_l)} \right) \right].$$

Finally, the initial formulation can be re-organized into the following compact DPO-like form:

$$\mathcal{L}_{\text{gen-dpo}}(\phi) = -\mathbb{E}_{(\bar{m}_w, \bar{m}_l) \sim \mathcal{B}} \left[ \log \sigma \left( \beta \log \frac{G_\phi(\bar{m}_w)}{G_{\text{ref}}(\bar{m}_w)} - \beta \log \frac{G_\phi(\bar{m}_l)}{G_{\text{ref}}(\bar{m}_l)} \right) \right].$$

## H.2. IPO-like Loss Formulation

Building on the methodology presented in (Tang et al., 2024), we can represent the DPO-like loss formulation in a more generalized form as follows:

$$\mathcal{L}_{\text{gen-dpo}}(\phi) = -\mathbb{E}_{(\bar{m}_w, \bar{m}_l) \sim \mathcal{B}} \left[ \mathcal{F} \left( \beta \left( \log \frac{G_\phi(\bar{m}_w)}{G_{\text{ref}}(\bar{m}_w)} - \log \frac{G_\phi(\bar{m}_l)}{G_{\text{ref}}(\bar{m}_l)} \right) \right) \right],$$

where $\mathcal{F}$ is a scalar function $\mathcal{F} : \mathbb{R} \to \mathbb{R}$ that map input values to scalar outputs. In the case of DPO, $\mathcal{F}$ is typically chosen to be the log-sigmoid function. However, DPO can encounter difficulties when preferences are deterministic. For example, if the probability of $m_w$ defeating $m_l$ is exactly 1, indicating deterministic preference, the difference between them becomes unbounded and approaches toward infinity such as follows:

$$\left( \frac{G_\phi(\bar{m}_w)}{G_{\text{ref}}(\bar{m}_w)} \gg \frac{G_\phi(\bar{m}_l)}{G_{\text{ref}}(\bar{m}_l)} \right) \implies \left( \log \frac{G_\phi(\bar{m}_w)}{G_{\text{ref}}(\bar{m}_w)} - \log \frac{G_\phi(\bar{m}_l)}{G_{\text{ref}}(\bar{m}_l)} \right) \to +\infty.$$

Assuming that $\beta$ is a positive real number, the term inside the log-sigmoid function becomes infinite, leading to:

$$\log \sigma \left( \beta \left( \log \frac{G_\phi(\bar{m}_w)}{G_{\text{ref}}(\bar{m}_w)} - \log \frac{G_\phi(\bar{m}_l)}{G_{\text{ref}}(\bar{m}_l)} \right) \right) \to \log \sigma(+\infty).$$

Since $\sigma(+\infty) = 1$, it follows that:

$$\log \sigma(+\infty) = \log(1) = 0.$$

Therefore, when preferences are deterministic, the loss function converges to 0 for any value of $\beta$. In other words, the regularization term $\beta$ becomes irrelevant and does not play any role in such cases.

To address these challenges, IPO introduces a stronger regularization term that penalizes models for exhibiting excessive confidence in preference margins. Specifically, IPO replaces the log-sigmoid function used in DPO with a squared loss function (Tang et al., 2024). The quadratic nature of the squared loss penalizes large deviations more heavily, discouraging the model from generating extreme outputs (Rosasco et al., 2004). In deterministic preference cases, the squared loss establishes the boundary to prevent the loss function from converging to 0 for any value of $\beta$ (Azar et al., 2024).

Recall that we can express the IPO-like loss formulation as follows:

$$\mathcal{L}_{\text{gen-ipo}}(\phi) = -\mathbb{E}_{(\bar{m}_w, \bar{m}_l) \sim \mathcal{B}} \left[ \left( \log \left( \frac{G_\phi(\bar{m}_w)}{G_\phi(\bar{m}_l)} \cdot \frac{G_{\text{ref}}(\bar{m}_l)}{G_{\text{ref}}(\bar{m}_w)} \right) - \frac{1}{2\beta} \right)^2 \right].$$

As shown, the squared loss function is implemented and $\beta$ is explicitly positioned outside the logarithm term. Let us examine the behavior of this IPO-like loss for different values of $\beta$. In the case of $\beta \to \infty$, the term $\frac{1}{2\beta} \to 0$, simplifying the loss function to:

$$\mathcal{L}_{\text{gen-ipo}}(\phi) = -\mathbb{E} \left[ \left( \log \left( \frac{G_\phi(\bar{m}_w)}{G_\phi(\bar{m}_l)} \cdot \frac{G_{\text{ref}}(\bar{m}_l)}{G_{\text{ref}}(\bar{m}_w)} \right) \right)^2 \right].$$

To minimize this loss, the following conditions should ideally be met:

$$\frac{G_\phi(\bar{m}_w)}{G_\phi(\bar{m}_l)} \approx \frac{G_{\text{ref}}(\bar{m}_w)}{G_{\text{ref}}(\bar{m}_l)}.$$

Thus, as $\beta \to \infty$, our current model $G_\phi$ converges to the reference model $G_{\text{ref}}$. In contrast, as $\beta \to 0$, the term $\frac{1}{2\beta} \to \infty$ begins to dominate the loss function, causing the IPO-like loss to converge toward the DPO-like formulation. This suggests that the IPO-like loss exhibits distinct behavior depending on the value of $\beta$, even in deterministic preference scenarios. In contrast, the DPO-like loss renders $\beta$ irrelevant in such scenarios, meaning the loss remains unaffected by changes in $\beta$.

# I. Self-Supervised Training Process for StitchNet

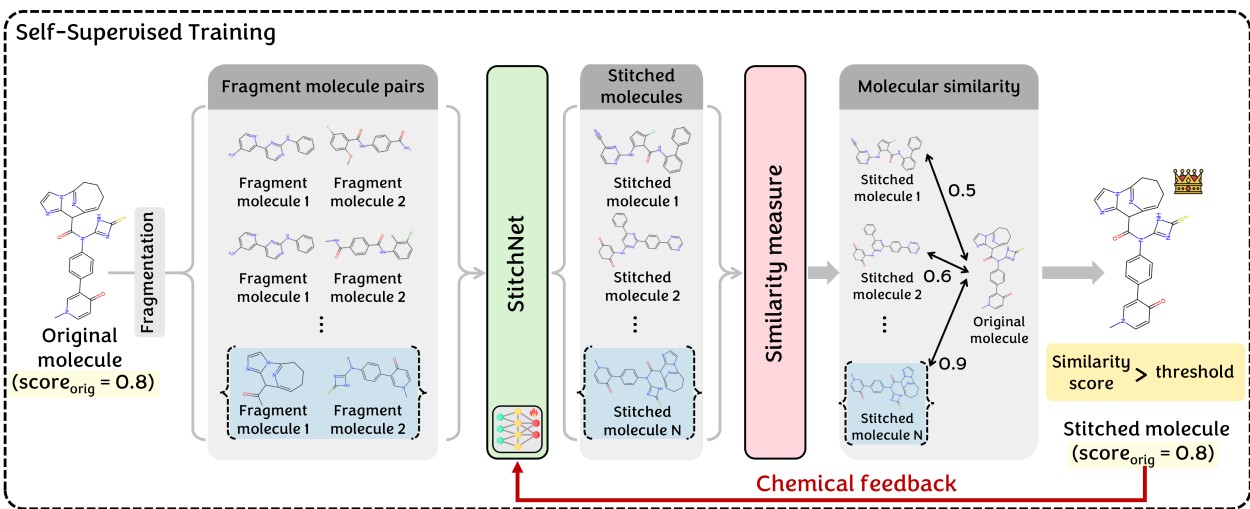

*Figure 15.* An illustration of the self-supervised training process for StitchNet. An original molecule is sampled from the offline dataset and decomposed into two fragment molecules using a fragmentation function. These pairs of fragment molecules are then fed into StitchNet to generate new stitched molecules. The molecular similarity between the stitched molecule and the original molecule is measured, and if it exceeds a pre-defined threshold, the true objective score of the original molecule is leveraged as a chemical feedback.

In this section, we provide a detailed explanation of the self-supervised training process for StitchNet, which is a key differentiating factor from traditional rule-based crossover operators. The importance of this process lies in its ability to leverage chemical feedback, allowing StitchNet to better understand how stitched molecules are likely to exhibit objective scores when two molecules are combined. Unlike rule-based crossover operators, StitchNet is built using a neural network architecture that enables it to learn from such chemical feedback.

As shown in Figure 15, we begin by sampling an original molecule from the offline dataset, each with corresponding known objective scores. We then apply a fragmentation function within the crossover operator (Jensen, 2019) to decompose the original molecule into two smaller fragment molecules. There are multiple possible pairings of these fragment molecules, and we consider all viable pairs as inputs to StitchNet. Subsequently, StitchNet takes these pairs of fragment molecules and generates corresponding offspring stitched molecules. We then measure the molecular similarity between each stitched molecule and the original molecule. If the similarity exceeds a certain threshold (e.g., 0.9), we consider the stitched molecule sufficiently similar to the original molecule. This high similarity allows us to leverage the known objective scores of the original molecule as an approximation for the stitched molecule's objective scores, effectively providing chemical feedback to StitchNet. We use this feedback to train StitchNet with the loss function specified in Equation 9. We think that this approach is reasonable based on two key assumptions. First, since the fragment molecules are derived from the original molecule, the stitched molecule is expected to share similar characteristics. Second, because structurally similar molecules often exhibit similar properties (Barbosa & Horvath, 2004; Alvesalo et al., 2006; Maggiora et al., 2014), we assume that the stitched molecule will likely exhibit objective scores comparable to the original molecule. By ensuring that the stitched molecule is sufficiently similar to the original, we can reasonably use the original molecule's objective scores as an approximation for the stitched molecule's scores.

The rationale for this self-supervised training process arises from the inherent nature of the offline MOMO problem. In an online setting, it would be possible to sample two molecules from the offline dataset, input them into StitchNet, generate a stitched molecule, and then query an oracle to obtain its true objective scores for chemical feedback. However, in an offline setting, additional oracle queries are not possible. Therefore, rather than simply using two random molecules from the offline dataset, we decompose a single molecule into two fragment molecules, which are then input into StitchNet. Since the true objective scores of the stitched molecules cannot be obtained due to the unavailability of additional oracle queries, we instead leverage the objective scores of the original molecule as a form of chemical feedback. This allows us to approximate the likely performance of the stitched molecule, which in turn can guide StitchNet in learning how to generate stitched molecules with desirable objective scores during the molecular stitching process.

## J. Priority Sampling Process for StitchNet

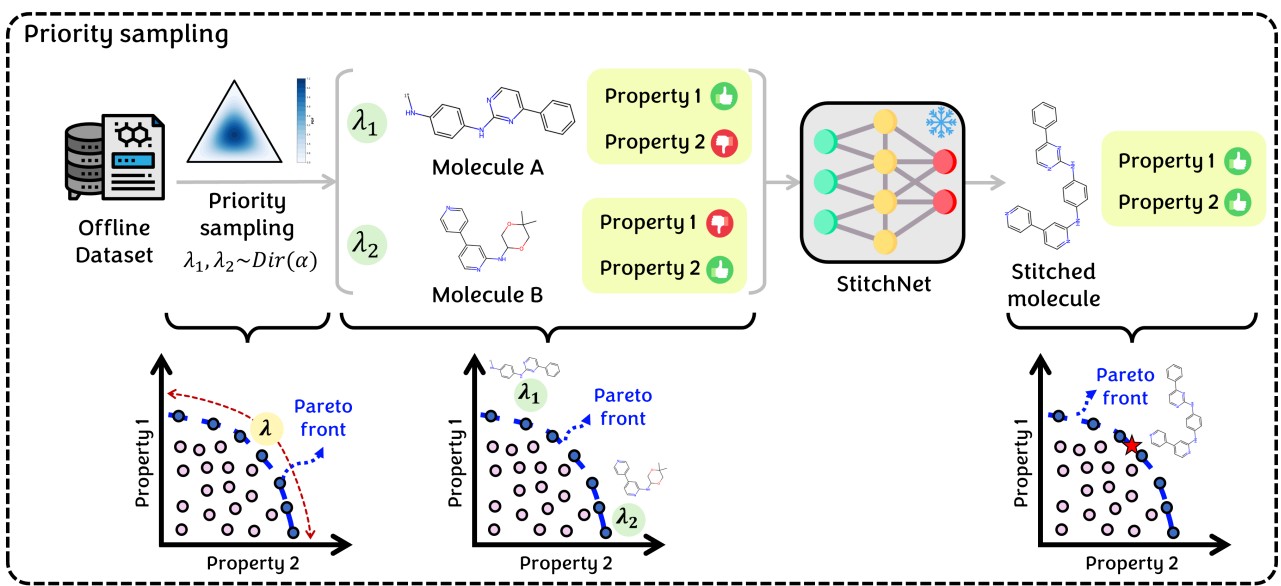

*Figure 16.* An illustration of the priority sampling process for StitchNet. The figure demonstrates how different weight configurations $\lambda$ are sampled from the Dirichlet distribution $\text{Dir}(\alpha)$, guiding the selection of molecular pairs from the offline dataset. For instance, $\lambda_1$ focuses more on property 1, while $\lambda_2$ emphasizes property 2, resulting in the selection of molecules A and B, respectively. These molecules are then fed into StitchNet, which generates a novel stitched molecule with the aim of combining the desirable properties of both parent molecules. This priority sampling promotes diversity and balance in the stitched molecules, enhancing the convergence towards the Pareto front.

In this section, we visualize the priority sampling process and explain why it is beneficial for the molecular stitching process in StitchNet. Consider a scenario where we are optimizing two molecular properties: property 1 and property 2, as shown in Figure 16. Our goal is to sample a diverse set of molecular pairs from the offline dataset; for instance, one molecule (Molecule A) exhibits characteristics more aligned with property 1, and the other molecule (Molecule B) emphasizes property 2. This diversity is crucial because StitchNet seeks to combine these molecules to create a novel stitched molecule that inherits the desirable properties of each parent molecule. If we sample pairs of molecules that have similar characteristics or properties, the benefit of the molecular stitching process diminishes due to the lack of diversity. To address this, we propose using priority sampling with a Dirichlet distribution to automatically generate diverse weight configurations, denoted as $\lambda$. Different weight configurations indicate varying levels of importance or priority for each property, allowing us to sample molecules from the offline dataset using different perspectives or priorities.

As depicted in Figure 16, $\lambda_1$ represents a weight configuration that focuses more on property 1, while $\lambda_2$ emphasizes property 2. It is important to note that these weight configurations are sampled from the Dirichlet distribution $\text{Dir}(\alpha)$. Based on these configurations, corresponding molecules are sampled from the offline dataset and fed into StitchNet as molecule A and molecule B. StitchNet then generates a novel stitched molecule with the aim of possessing both favorable property 1 and property 2. In terms of the Pareto front, sampling molecules based on $\lambda_1$ corresponds to selecting molecules near the y-axis (emphasizing property 1), whereas $\lambda_2$ corresponds to molecules near the x-axis (emphasizing property 2). By performing molecular stitching via StitchNet, we aim to generate molecules that balance both properties, thereby improving convergence towards the Pareto front. Please note that weight configurations for focusing on property 1 and property 2 is merely an example for better understanding. In practice, we can sample diverse molecular pairs using priority sampling, as the Dirichlet distribution allows us to automatically generate diverse combinations of weight configurations.

## K. Pseudo-Code

---

**Algorithm 1** StitchNet

---

**Input:** StitchNet $\mathcal{S}_\psi$, Unlabelled dataset $\mathcal{D}_u$, Offline dataset $\mathcal{D}$, Crossover operation *Crossover*, Dirichlet distribution *Dir*,
Concentration constant $\alpha$, Fragmentation function *Cut*, Similarity threshold $\delta$, Similarity function $\texttt{sim}$

**Output:** Generated stitched molecules $\bar{m}$

---

$\triangleright$ *Pre-training for StitchNet*

Sample parent molecules from unlabelled chemical dataset; $m_i \sim \mathcal{D}_u$ and $m_j \sim \mathcal{D}_u$

Generate offspring molecule with crossover operation; $m_o \leftarrow Crossover(m_i, m_j)$

Train StitchNet $\mathcal{S}_\psi$ to resemble crossover operation; $\psi \leftarrow \arg\max_\psi \mathbb{P}(m_o \mid S_\psi(m_i, m_j))$

Set pretrained StitchNet as a reference model; $S_{\text{ref}} \leftarrow S_\psi$

---

$\triangleright$ *Self-supervised training for StitchNet*

Sample objective preference; $\lambda \sim Dir(\alpha)$

Sample molecule and its score from offline dataset with preference; $(m_s, r_s) \overset{\lambda}{\sim} \mathcal{D}$

Cut $m_s$ into all possible $Z$ fragment molecule sets; $\{(m_{ai}, m_{bi})\}_{i=1}^{Z} \leftarrow Cut(m_s)$

Find the most similar offspring and its fragment set with original molecule $m_s$;

$(m_a, m_b) \leftarrow \underset{(m_{ai}, m_{bi})}{\arg\max} \texttt{sim}(m_s, Crossover(m_{ai}, m_{bi})) \quad \text{subject to} \quad \texttt{sim}(\cdot) \geq \delta$

Provide chemical feedback to StitchNet while maintaining the chemical validity;

$\mathcal{L}_{\text{stitch}}(\psi) \leftarrow (-\log \mathcal{S}_\psi(\bar{m}_{\text{stit}}) + \log \mathcal{S}_{\text{ref}}(\bar{m}_{\text{stit}}) + \mathcal{R}(m_{\text{orig}}))^2 \cdots \text{eq.9};$

---

$\triangleright$ *Molecular Stitching*

Sample two objective priorities; $\lambda_1 \sim Dir(\alpha)$ and $\lambda_2 \sim Dir(\alpha)$

Sample parent molecules of different objective priorities; $m_1 \overset{\lambda_1}{\sim} \mathcal{D}$ and $m_2 \overset{\lambda_2}{\sim} \mathcal{D}$

Generate a novel stitched molecule using fine-tuned StitchNet; $\bar{m} \sim \mathcal{S}_\psi(m_1, m_2)$

**Return** $\bar{m}$

---

**Algorithm 2** MolStitch

---

**Input:** Pretrained Generator $G_{\text{ref}}$ Pretrained StitchNet $\mathcal{S}_{\text{ref}}$, Offline dataset $\mathcal{D}$, Proxy model $\hat{f}_\theta$, Dirichlet distribution *Dir*,
Concentration constant $\alpha$,

**Output:** Final molecules for evaluations $m_{final}$

---

Initialize Generative model; $G_\phi \leftarrow G_{\text{ref}}$

Initialize StitchNet; $\mathcal{S}_\psi \leftarrow \mathcal{S}_{\text{ref}}$

Update Generative model $G_\phi$ with offline dataset $\mathcal{D}$;

Train proxy model $\hat{f}_\theta$ with pairwise ranking loss in eq.7;

Sample objective preference; $\lambda \sim Dir(\alpha)$

Finetuning StitchNet $\mathcal{S}_\psi$ with preference $\lambda$ ;

Sample objective preferences; $\lambda_1, \lambda_2 \sim Dir(\alpha)$

Sample stitched molecule $\bar{m}$ by molecular stitching; $\bar{m} \sim \mathcal{S}_\psi(m_1, m_2)$

Determine winning and losing molecules using proxy model $\hat{f}_\theta$ by eq.11; $(m_w, m_l) \leftarrow \hat{f}_\theta(\bar{m})$

Fine-tuning Generative model with IPO-like loss in eq.14;

Sample final molecules for evaluations; $m_{final} \sim G_\phi$

**Return** $m_{final}$

---

# L. Experimental Details

In this section, we provide a comprehensive overview of our experimental setup, covering (1) experimental settings and configurations, (2) a detailed description of the molecular objectives used in our study, and (3) an in-depth discussion of hyperparameters and implementation details.

## L.1. Experimental Settings and Configurations

**Oracle calls.** In this work, we conducted two main experiments: 1) Molecular Property Optimization (MPO) task (Gao et al., 2022) and 2) docking score optimization task (Lee et al., 2023). Recall that both experiments were designed to simulate real-world constraints by restricting the number of oracle calls, which represent expensive evaluations of molecular properties. For the MPO task, the total number of oracle calls was limited to 10,000 (Gao et al., 2022). Following this guideline, we allocated 5,000 calls to construct the offline dataset and reserved the remaining 5,000 for evaluation. Specifically, we used the initial 5,000 oracle calls to build the offline dataset, which served as the training data for developing and fine-tuning the generative model during the offline optimization process. After completing offline optimization, the performance of the fine-tuned generative model was evaluated using the remaining 5,000 oracle calls on the molecules it newly generated. For the docking score optimization task, the total number of oracle calls was restricted to 3,000 (Lee et al., 2023). This lower allocation might be due to the longer time required for evaluating docking scores. Similar to the MPO task in concept, we allocated 1,500 oracle calls to construct the offline dataset and the remaining 1,500 to evaluate the performance of the fine-tuned generative model.

**Offline dataset collection.** To construct the offline datasets for both experiments, we utilized the ZINC dataset (Sterling & Irwin, 2015), which is a publicly available chemical database that provides a collection of commercially available compounds. The ZINC dataset offers a wide variety of molecular structures, providing a large chemical space to explore for potential drug candidates. Its compounds are also available in formats suitable for molecular docking, making it a good resource for identifying potential compounds that may bind to biological targets. Therefore, we considered the ZINC dataset to be well-suited for both the MPO task and the docking score optimization task. It is worth noting that we also used the ZINC dataset during the pre-training stage; however, at that stage, we only utilized the molecular structures without any associated objective scores or additional information. When aiming to optimize specific molecular objectives, we needed to query the oracle to obtain the objective scores of molecules within the ZINC dataset. For the MPO task, we randomly sampled 5,000 molecules from the ZINC dataset and executed 5,000 oracle calls to evaluate their corresponding molecular objective scores, such as JNK3, GSK3$\beta$, QED, and SA. We collected this data in the form of `(molecule, objective scores)` pairs. Similarly, for the docking score optimization task, we randomly sampled 1,500 molecules from the ZINC dataset and performed 1,500 oracle calls to evaluate their corresponding docking scores for five proteins alongside QED and SA. These constructed offline datasets were subsequently used for offline optimization in our proposed framework as well as across all competing methods to ensure a fair comparison.

## L.2. Descriptions of the Molecular Objectives

In this work, we adopted four commonly used molecular objectives—JNK3, GSK3$\beta$, QED, and SA—for the MPO task. For the docking score optimization task, we targeted the docking scores of five proteins—parp1, fa7, jak2, braf, and 5ht1b—alongside QED and SA. The docking scores were calculated following the experimental protocol of prior work (Guo & Schwaller, 2024b), using the normalized QuickVina2 docking score (Alhossary et al., 2015). Specifically, the normalized docking score (DS) is calculated using the given equation:

$$\text{Normalized DS} = -\frac{\text{DS}}{20},$$

where DS represents the original docking score. Detailed descriptions of each molecular objective and protein are provided below.

**JNK3.** JNK3 is a member of the c-Jun N-terminal kinases (JNKs) family, which belongs to the mitogen-activated protein kinase (MAPK) pathway and is primarily expressed in the central nervous system (Bogoyevitch & Kobe, 2006). It plays a crucial role in mediating cellular responses to stress, including apoptosis, inflammation, and neuronal damage (Bogoyevitch & Kobe, 2006). Targeting JNK3 inhibition is one of the key molecular objectives in drug discovery because it may prevent or reduce neuronal cell death and inflammation, making it a promising therapeutic target for neurodegenerative diseases such as Alzheimer's disease (Resnick & Fennell, 2004).

**GSK3$\beta$.** Glycogen synthase kinase 3 beta (GSK3$\beta$) is a serine/threonine protein kinase involved in various cellular processes, including glycogen metabolism, cell proliferation, differentiation, and apoptosis (Beurel et al., 2015). It has gained significant attention in neurodegenerative disease research due to its role in regulating tau protein phosphorylation, amyloid precursor protein processing, and neuronal survival (Jope et al., 2007). Inhibiting GSK3$\beta$ is considered a vital molecular objective in drug discovery, as it could modulate these pathological processes and potentially slow or prevent the progression of neurodegenerative diseases (Jope et al., 2007).

**QED.** Quantitative Estimate of Drug-likeness (QED) is a metric widely used in molecular optimization to evaluate the drug-likeness of a molecule (Bickerton et al., 2012). It consists of several physicochemical properties, including molecular weight, lipophilicity (logP), topological polar surface area (TPSA), the number of hydrogen bond donors and acceptors, and the count of aromatic rings and rotatable bonds (Guan et al., 2019). It provides a score ranging from 0 to 1, with higher scores indicating molecules that are more likely to have favorable drug-like properties.

**SA.** Synthetic Accessibility (SA) is a metric used in molecular optimization to assess the ease with which a molecule can be synthesized in a laboratory setting (Ertl & Schuffenhauer, 2009). It considers various structural features that influence synthesis complexity, such as the presence of complex ring systems, functional groups, stereocenters, and the overall size and branching of the molecule (Ertl & Schuffenhauer, 2009). The SA score ranges from 1 to 10, with lower scores indicating higher synthetic feasibility. In this work, we transform the SA score into the normalized SA score, following prior studies (Lee et al., 2023; Guo & Schwaller, 2024b), to formulate it as a maximization objective. Specifically, the normalized SA score is given by the following equation:

$$\text{Normalized SA} = \frac{10 - \text{SA}}{9}.$$

This adjustment ensures that higher normalized SA scores correspond to molecules that are easier to synthesize, within the score range of 0 to 1.

**parp1.** Poly (ADP-ribose) polymerase 1 (parp1) is a protein enzyme that plays a crucial role in DNA damage detection and repair (Rouleau et al., 2010). It is involved in various cellular processes, including chromatin remodeling, transcriptional regulation, and cell death signaling (Ray Chaudhuri & Nussenzweig, 2017). In recent years, dysregulation of parp1 activity has been linked to several neurodegenerative diseases, such as Parkinson's disease, where excessive activation of parp1 can lead to neuronal death through a process known as parthanatos (Liu et al., 2022). Consequently, targeting parp1 has become a key molecular objective in drug discovery, not only for cancer treatment but also for developing neurotherapeutics aimed at preventing neuronal loss (Zhang et al., 2023).

**fa7.** Coagulation factor VII (fa7), also known as proconvertin, is a vital protein in the blood coagulation pathway (Hall et al., 1964). It plays a crucial role in initiating the clotting process by activating factor X in the presence of tissue factor (TF), leading to the conversion of prothrombin to thrombin and ultimately forming a blood clot (Eigenbrot, 2002). Targeting fa7 represents another key molecular objective in drug discovery, particularly for managing thrombotic and cardiovascular diseases. Specifically, inhibitors of fa7 are being explored as potential anticoagulants to prevent and treat conditions such as deep vein thrombosis, embolism, and stroke (Robinson et al., 2010).

**jak2.** Janus kinase 2 (jak2) is a non-receptor tyrosine kinase that plays a critical role in the signaling pathways of various cytokines (Yamaoka et al., 2004). It is involved in various cellular processes, including cell growth, differentiation, and immune function (Seavey & Dobrzanski, 2012). In drug discovery, jak2 has gained attention due to its association with myeloproliferative neoplasms and other hematological malignancies (Senkevitch & Durum, 2017). Inhibiting jak2 is considered a key molecular objective, as it can potentially provide therapeutic benefits in inflammatory and autoimmune disorders (Seavey & Dobrzanski, 2012).

**braf.** B-Raf proto-oncogene (braf) encodes a serine/threonine kinase that is part of the MAPK/ERK signaling pathway, which plays a crucial role in regulating cell growth and migration during various cellular processes (González-González et al., 2020). Mutations in the braf gene are commonly found in various cancers, including melanoma, colorectal cancer, and thyroid cancer (Atiqur Rahman et al., 2014). Therefore, targeting the braf can be a critical therapeutic objective in oncology to target these cancer-specific mutations and halt the progression of the disease (Sanz-Garcia et al., 2017).

**5ht1b.** 5-Hydroxytryptamine receptor 1B (5ht1b) is a G protein-coupled receptor that binds serotonin (Launay et al., 2002). It is widely expressed in the central nervous system and plays important roles in regulating neurotransmitter release, neuronal firing, mood, and appetite (Fink & Göthert, 2007). 5ht1b has emerged as an important molecular target in drug discovery for neurological and psychiatric disorders, particularly in the treatment of migraine and depression (Giniatullin, 2022).

### L.3. Hyperparameters and Implementation Details

**Implementation of the generative model.** We closely followed the architecture settings for REINVENT as described in the PMO benchmark (Gao et al., 2022), while the settings for GFlowNets were based on GeneticGFN (Kim et al., 2024a), and those for Mamba were taken from Saturn (Guo & Schwaller, 2024b). Since all of these generative models were originally designed for an online setting, we made necessary adjustments to the number of molecule updates and the experience replay to adapt them for our offline settings. The final hyperparameters for the generative models were primarily determined based on the performance of REINVENT, which served as our backbone generative model, and are detailed in Table 17.

**Stabilizing GFlowNets.** During the training of GFlowNets, we encountered instability with the original setting of the `logZ` parameter, which plays a crucial role in trajectory balancing and needs to be adjusted according to specific settings (Malkin et al., 2022). To be more specific, it was initially set to a high value (`logZ` = 5.0) with a learning rate of 0.1, as specified in GeneticGFN. To stabilize the training process, we reduced the `logZ` value to 0.001 and aligned the learning rate with that of the generative model (from 0.1 to 0.0005). This adjustment resulted in more stable training and significantly improved performance. Additionally, during preference optimization, while both REINVENT and Mamba require only the generative model's likelihood as input, we recommend using the sum of likelihood and `logZ` for GFlowNets in order to further improve performance.

**Hyperparameters for StitchNet.** Recall that StitchNet combines two parent molecules as input and generates stitched molecules in an auto-regressive manner. Therefore, it operates by computing the hidden dimensions $h_1$ and $h_2$ of two parent molecules $m_1$ and $m_2$, respectively, and then averaging these hidden dimensions as $\frac{h_1+h_2}{2}$. StitchNet is built upon the REINVENT architecture. During the self-supervised training process for StitchNet, we applied a similarity threshold $\delta = 0.8$ between the original molecules and the stitched molecules. During the molecular stitching process, StitchNet combines two parent molecules, each sampled with different weight configurations through priority sampling. The resulting stitched molecules are stored in a buffer. Once the buffer is full, two molecules are randomly sampled to create non-overlapping pairs. These pairs are then evaluated by the proxy model to identify the winning and losing molecules. Subsequently, the IPO-like loss is applied to increase the likelihood of generating winning molecules while reducing the likelihood of generating losing molecules. The hyperparameter settings for StitchNet are summarized in Table 18.

*Table 17.* The hyperparameter settings for generative models in MolStitch framework.

| REINVENT | | GFlowNets | | Mamba | |
|---|---|---|---|---|---|
| Batch size | 200 | Batch size | 200 | Batch size | 200 |
| Embedding dimension | 128 | Embedding dimension | 128 | Embedding dimension | 256 |
| Hidden dimension | 512 | Hidden dimension | 512 | Hidden dimension | 256 |
| Number of layers | 3 | Number of layers | 3 | Number of layers | 12 |
| Sigma | 500 | Sigma | 500 | Sigma | 500 |
| Experience replay size | 300 | Experience replay size | 300 | Experience replay size | 300 |
| Augmentation round | 8 | Augmentation round | 8 | Augmentation round | 8 |
| Batch update | 2 | Batch update | 2 | Batch update | 2 |
| Learning rate | 5e-04 | Learning rate | 5e-04 | Learning rate | 5e-04 |
| | | `logZ` | 0.001 | | |

*Table 18.* The hyperparameter settings for StitchNet.

| Molecular stitching | |
|---|---|
| $\alpha$ for priority sampling | 1.0 |
| Number of stitch rounds | 16 |
| Stitched molecules per stitch round | 250 |
| Population pool | 1000 |
| Temperature $\beta$ for IPO | 0.2 |

# M. Extended Related Work

## M.1. Generative Models for Molecular Discovery

The rapid advancement of generative models has profoundly impacted various fields, including computer vision (Croitoru et al., 2023), natural language processing (Chang et al., 2024), and audio signal processing (Deshmukh et al., 2023). This progress has extended to molecular discovery (Anstine & Isayev, 2023; Son et al., 2024) and drug discovery (Jiménez-Luna et al., 2021; Shin et al., 2025b), where generative models have proven effective in designing and optimizing molecules towards promising regions of the chemical space. Various types of generative models have been proposed, such as follows:

**Genetic algorithms.** Inspired by the principles of natural evolution, genetic algorithms (GAs)—also known as evolutionary search methods (Shin et al., 2025a)—maintain a population of candidate solutions that are iteratively refined based on a predefined fitness function. These algorithms enhance population quality over successive generations by applying key operations such as selection, crossover, mutation, and replacement. In the domain of molecular discovery, GraphGA (Jensen, 2019) has shown significant promise in generating high-quality molecular structures using GA-based techniques.

**Sampling-based methods.** These methods leverage advanced sampling techniques to draw samples from distributions that are likely to yield desirable molecular properties. MARS (Xie et al., 2021a) is a notable example that employs Markov Chain Monte Carlo (MCMC) sampling to efficiently search for high-quality molecules. By focusing on probabilistic sampling, these methods can explore the chemical space more efficiently than deterministic approaches.

**Reinforcement learning (RL).** RL-based methods formulate the molecule generation process as a Markov decision process, allowing an RL agent to interact with a chemical environment to construct molecular structures in an autoregressive manner. A prominent example is REINVENT (Olivecrona et al., 2017), which utilizes a GRU model (Chung et al., 2014) as its RL agent to generate molecules in SMILES format. REINVENT has been acknowledged as one of the best models for various molecular property optimization tasks, showcasing the effectiveness of RL-based methods. More specifically, the strength of RL-based methods lies in its effective use of trial-and-error to navigate the environment (Angermueller et al., 2019; Shin et al., 2022; 2024a), allowing it to discover promising solutions and achieve robust performance across diverse scenarios. Following its success, several variants have been proposed to enhance its capabilities. One line of research focuses on improving the underlying neural architecture by replacing the GRU with either a transformer (He et al., 2022) or Mamba (Gu & Dao, 2023; Lee et al., 2025). Another approach incorporates data augmentation techniques to boost sample efficiency, leading to methods like Augmented Memory (Guo & Schwaller, 2024a), which achieved new state-of-the-art performance.

**GFlowNets.** While RL-based methods have shown effectiveness, they often struggle with maintaining diversity in the generated molecules due to a tendency to exploit a single promising direction. GFlowNets (Jain et al., 2022; M Ghari et al., 2024; Son et al., 2025) aims to address this limitation by emphasizing probabilistic sampling over reward maximization, inherently promoting diversity in the generated molecules. As a result, GFlowNets have gained popularity in multi-objective molecular optimization tasks, where generating a diverse set of high-quality molecules across multiple objectives is crucial.

## M.2. Multi-Objective Molecular Optimization

The multi-objective molecular optimization (MOMO) problem differs from single-objective optimization by requiring the simultaneous optimization of multiple molecular properties, which often conflict with one another. In the context of the MOMO problem, identifying a single solution that optimally satisfies all objectives is generally infeasible. Instead, the goal shifts to discovering a diverse set of Pareto optimal molecules, where improving one objective may lead to trade-offs in others. To tackle multiple objectives, several studies have integrated Bayesian optimization (BO) within their molecular optimization frameworks. For instance, GPBO (Tripp et al., 2021) incorporate BO into GraphGA , resulting in enhanced sample efficiency. In a similar approach, LamBO (Stanton et al., 2022) applies BO alongside denoising autoencoders to address the multi-objective biological sequence design problem. Other studies have employed scalarization, which simplifies the multi-objective problem by converting multiple objectives into a single scalar objective function (Gunantara, 2018). This scalarization is typically achieved by combining the objectives using a weighted sum or other aggregation techniques (Marler & Arora, 2010; Deb et al., 2016). Scalarization offers simplicity and ease of implementation, making it a popular choice for its scalability and computational efficiency (Cho et al., 2017). In the context of the MOMO problem, MIMOSA (Fu et al., 2021) utilizes linear scalarization to efficiently manage the complexity of multiple objectives, while demonstrating strong performance and scalability. Similarly, MARS (Xie et al., 2021a) applies scalarization to effectively handle up to four molecular objectives, further showcasing the potential of scalarization in the MOMO problem. However, scalarization presents challenges in selecting appropriate weights. Users must assign weights to each objective to reflect its relative

importance, a process that is often sensitive and subjective (Royer et al., 2023). Incorrect or biased weight selection may fail to accurately represent true preferences, potentially resulting in suboptimal solutions (Zhang & Golovin, 2020). In our study, we also employ the scalarization approach due to its widespread adoption and practical advantages (Fromer & Coley, 2023). However, to mitigate the limitations associated with subjective weight selection, we introduce priority sampling using the Dirichlet distribution to generate a diverse set of weight configurations. This enables our StitchNet to operate on a wide variety of molecular pairs, each representing a different balance of multiple objectives.

### M.3. Offline Model-based Optimization

As mentioned earlier in the main manuscript, one of the most promising approaches for addressing the offline MOMO problem is offline model-based optimization (MBO) (Trabucco et al., 2022). The goal of offline MBO is to optimize the objective function using a pre-collected offline dataset, without the ability to acquire new data during the optimization process. In this approach, the proxy (surrogate) model—such as Gaussian processes, random forests, or neural networks—is trained on the offline dataset to approximate the objective function (Dao et al., 2024). This proxy model is then used to predict objective scores for new inputs, guiding the optimization algorithm in finding inputs that maximize the predicted objective scores. The most straightforward approach in offline MBO is to use a differentiable vanilla proxy model and apply gradient ascent to find optimal inputs. However, this approach may face limitations such as increased inaccuracies and a higher risk of overfitting. To address these limitations, various recent studies have been proposed.

**Improving the proxy model.** One line of research focuses on enhancing the accuracy and robustness of the proxy model to better handle high-dimensional and complex objective functions. Some studies (Trabucco et al., 2021; Qi et al., 2022) enforce constraints to mitigate overfitting and address distributional shifts caused by out-of-distribution (OOD) inputs, while another study (Yu et al., 2021) enhances the generalization capabilities of the proxy model by employing a local smoothness prior. Additionally, more sophisticated methods such as gradient matching (Hoang et al., 2024) and policy-guided gradient search (Chemingui et al., 2024) have been proposed to improve the proxy model and overall performance.

**Ensemble learning.** To leverage the benefits of ensemble learning, several studies (Trabucco et al., 2022; Yuan et al., 2023; Chen et al., 2023a) have proposed to utilize multiple proxy models to combine their predictions, thereby enhancing the robustness and reliability of the optimization process. Building on the effectiveness of ensemble learning, we conducted additional experiments utilizing multiple proxy models, as described in Appendix E. The results reveal that increasing the number of proxies improves the performance of our framework, underscoring the robustness and reliability that ensemble methods bring to the optimization process.

**Improving optimization algorithms.** Another line of research concentrates on improving the optimization algorithms used within the offline MBO framework. For example, the bidirectional learning technique (Chen et al., 2022; 2023b) has been introduced to utilize both forward and backward mappings to generate input configurations that are likely to produce optimal outputs while adhering to the data distribution of the offline dataset. Furthermore, the synthetic pretraining technique (Nguyen et al., 2023) has been proposed to enhance sample efficiency and overall performance by integrating unsupervised learning with in-context pretraining. Additionally, the bootstrapping technique (Kim et al., 2023) has been developed to enhance the optimization process by iteratively augmenting the offline dataset with self-generated data.

### M.4. Preference Optimization in Generative Models

In recent years, preference optimization has gained significant attention, particularly with the rise of large language models and generative models (Tang et al., 2024). As these models grow more powerful and are deployed into real-world applications, the need to align their outputs with human expectations becomes increasingly important. Preference optimization enables models to better align with human standards in subjective areas such as sentiment, creativity, and ethical considerations.

**Reinforcement Learning from Human Feedback (RLHF).** A leading and widely adopted method for incorporating human preferences into model training is RLHF (Ouyang et al., 2022). By embedding human feedback within an RL framework, RLHF allows models to generate higher-quality content that aligns more closely with human judgments. Notable implementations like OpenAI's ChatGPT (Achiam et al., 2023) have demonstrated significant performance improvements through RLHF, highlighting its potential in fine-tuning models. This success has driven further research into more streamlined approaches that aim to simplify the incorporation of human preferences.

**Direct Preference Optimization (DPO).** DPO (Rafailov et al., 2023) is a recent method that moves away from RL and focuses directly on optimizing for human preferences without the need for reward modeling. It operates by directly

training on human preference pairs, enabling the model to generate outputs that are consistently favored over less preferred alternatives. This approach is considered more straightforward and potentially more stable than RLHF, as it bypasses the complexities associated with RL training. However, DPO has exhibited limitations, particularly in scenarios involving deterministic preferences, due to its relatively weak regularization mechanisms.

**Identity Preference Optimization (IPO).** IPO (Azar et al., 2024) is a more recent method that builds on DPO by introducing enhancements to address its limitations and offering a more theoretically sound framework. Specifically, IPO incorporates a stronger regularization term that penalizes models for excessive confidence in preference margins. This is achieved by replacing the log-sigmoid function used in DPO with a squared loss function. The stronger regularization term in IPO aims to balance adaptation to the preference dataset while maintaining generalization capabilities, which is crucial for model performance on out-of-distribution (OOD) data. While IPO offers theoretical improvements over DPO, empirical results have been mixed. Some studies report IPO performing on par with or slightly better than DPO (Pal et al., 2024; Calandriello et al., 2024), while others observe diminished performance in certain settings (Hu et al., 2024b).

**Preference optimization in molecular discovery.** In large language models, preference typically reflects human sentiments, opinions, or judgments about what constitutes a desirable output. On the other hand, in the field of molecular discovery, preference represents the relative importance of each objective within the optimization process. When the generative model is tasked with optimizing several conflicting objectives, preference guides the optimization process by specifying how much weight or priority each objective should be given. For example, if a researcher wants to prioritize potency over safety, their preferences would assign more importance to optimizing potency. Conversely, if safety is more critical, the preference would shift toward that objective. Recently, preference optimization has been widely adopted in structure-based drug design to align the pre-trained generative model with preferred functional properties (Liu et al., 2024a; Gu et al., 2024). Our work also focuses on optimizing molecules with desired properties. However, unlike recent studies (Gu et al., 2024) that primarily use DPO and rely on existing preference datasets, our approach differs in several key ways. We explore a variety of preference optimization techniques—including RLHF, DPO, and IPO—and apply them to the offline multi-objective molecular optimization problem. More importantly, we generate a new preference dataset using our StitchNet model, which creates novel stitched molecules with desirable properties from pairs of existing molecules. In other words, rather than depending solely on existing datasets for preferences, we construct a separate proxy model and use StitchNet to build a tailored preference dataset, leveraging existing molecules to further enhance the optimization process. Additionally, we extend our approach to a semi-offline setting—a direction that recent studies have not explored yet. In this setting, we utilize a limited number of online evaluations by periodically querying an oracle function to assess molecules in large batches. This extension allows us to explore ways of further enhancing the optimization process by integrating new evaluation data.

## N. Competing Methods Details

In this section, we present a comprehensive review of the competing methods, highlighting their core principles, methodologies, and their comparative position relative to our proposed framework. Before delving into the details, we first aim to explain how molecular optimization methods, such as REINVENT (Olivecrona et al., 2017), are adapted to offline settings. While we use REINVENT as an example, this approach applies to all competing molecular optimization methods. In online settings, REINVENT actively generates molecules, queries the oracle to obtain objective scores as rewards, and updates the log-likelihood of generating those molecules based on the feedback. In contrast, in offline settings, it relies on a pre-existing offline dataset containing pairs of molecules and their corresponding objective scores, rather than actively generating and evaluating new molecules through oracle queries. This offline dataset becomes the sole source of information for training and optimizing the generative model. In this context, REINVENT computes the log-likelihood of a molecule and utilizes the corresponding objective scores from the offline dataset as rewards, updating itself in a supervised manner. This adaptation enables REINVENT to operate in offline settings, leveraging the available offline data to refine its generative capabilities.

- **REINVENT** (Olivecrona et al., 2017) is a reinforcement learning (RL) approach designed for molecular generation, where an agent interacts with its environment to create molecules. This approach autoregressively generates molecules as SMILES strings, with each new element (token) in the sequence building upon the previously generated elements. Note that this generation process is guided by a pre-trained model that enforces chemical grammar rules, ensuring the validity of the generated molecules. REINVENT has demonstrated superior performance in molecular optimization tasks, as highlighted by the PMO benchmark. This remarkable performance has led numerous follow-up studies to adopt REINVENT as their backbone generative model. Following this established trend, we have also integrated REINVENT as our backbone model to take advantage of its proven effectiveness in various molecular optimization

tasks. As a competing method in our study, REINVENT serves as a baseline, as it is trained exclusively on the offline dataset without applying further offline MBO techniques. This straightforward approach positions it as a reference point for evaluating the effectiveness of various MBO techniques, which leverage proxy models to fine-tune the generative model beyond the constraints of the offline dataset.

- **Augmented Memory (AugMem)** (Guo & Schwaller, 2024a) builds upon the REINVENT method by incorporating molecular data augmentation techniques and experience replay to enhance performance. The authors report that AugMem has achieved state-of-the-art results on the PMO benchmark, showcasing its effectiveness in molecular optimization tasks. In the context of offline optimization, offline MBO techniques typically use proxy models to guide the generation of synthetic data. This process involves the generative model producing new data points, which are then evaluated by the proxy model. The resulting augmented dataset allows the generative model to explore beyond the initial offline dataset. AugMem, in contrast, introduces a different approach to data augmentation specifically designed for molecular generation. By implementing AugMem in our study, we establish a valuable reference point for comparing specialized molecular data augmentation techniques against the proxy model-guided approaches used in conventional offline MBO.

- **GraphGA** (Jensen, 2019) is a method based on genetic algorithms that generates molecules by evolving a population through repeated cycles of selection, crossover, and mutation, all driven by a fitness function. GraphGA utilizes domain knowledge from the chemical experts to develop effective mutation and crossover strategies that facilitate an efficient exploration of molecular space. In our study, GraphGA serves as a key reference for implementing rule-based crossover operations, which we have also incorporated into our framework.

- **GeneticGFN** (Kim et al., 2024a) integrates genetic algorithms into the GFlowNets model for molecular generation. Specifically, this method leverages domain-specific genetic operators to efficiently explore the chemical space, enabling the generative model to implicitly acquire relevant domain knowledge. Consequently, the generative model's performance is enhanced through the strategic guidance provided by the genetic algorithm. The authors also highlight a complementary relationship between the two components: the genetic algorithm enhances GFlowNets' capacity for effective exploitation, while GFlowNets, in turn, increases the population diversity for the genetic algorithm. In our study, GeneticGFN serves as a crucial reference point for evaluating the effectiveness of genetic algorithms in the offline MOMO problem. Specifically, it allows us to assess the advantages gained from incorporating domain-specific knowledge through genetic operators in this context.

- **Saturn** (Guo & Schwaller, 2024b) builds upon the core mechanism of REINVENT while introducing significant architectural improvements. While REINVENT employs a GRU architecture, Saturn replaces it with the more powerful Mamba architecture. This substitution is motivated by Mamba's potentially greater capacity for modeling complex molecular structures more effectively. Furthermore, Saturn incorporates genetic algorithms into its Mamba-based model, drawing parallels to GeneticGFN's approach. This integration allows Saturn to leverage domain-specific genetic operators, potentially enhancing its ability to navigate the chemical space effectively. In our study, Saturn serves as a valuable reference point for two key aspects: first, it demonstrates the application of the Mamba architecture in molecular optimization tasks, and second, it provides insights into the benefits of incorporating domain-specific genetic operators in the context of offline MOMO.

- **Grad** (Zinkevich, 2003) represents the most straightforward offline MBO approach for tackling the offline MOMO problem. In particular, it employs a vanilla proxy model that directly approximates the true objective scores, training this proxy on the offline dataset. To address the generative aspect of the offline MOMO problem, Grad utilizes REINVENT as its backbone generative model, the same approach used in our proposed framework. This choice is consistently applied across all offline MBO-based competing methods to ensure a fair comparison. After training the vanilla proxy model, Grad fine-tunes the generative model using gradient ascent with respect to the trained vanilla proxy model's predictions. In our study, Grad serves as a crucial reference point as it demonstrates the basic application of offline MBO in the context of offline MOMO. Specifically, Grad enables us to investigate whether a vanilla proxy model is sufficient for this task, or if more sophisticated approaches are necessary for meaningful improvements in the offline MOMO problem.

- **COMs** (Trabucco et al., 2021) represents a more sophisticated offline MBO approach. Unlike Grad's vanilla proxy model, COMs employs adversarial learning to encourage the proxy model to provide conservative estimates of the true objective functions. This method establishes lower bounds on the objective estimates, which are then used during the

offline optimization process. By doing so, COMs aims to prevent erroneous overestimation caused by distributional shift, a common challenge in various offline optimization scenarios. In our study, COMs enables us to investigate whether these sophisticated methods offer significant improvements in the context of offline MOMO.

- **IOM** (Qi et al., 2022) considers offline MBO from a domain adaptation perspective. This method aims to train a proxy model that can accurately predict true objective scores ('target domain') when trained solely on the given offline dataset ('source domain'). To achieve this, IOM introduces invariant representation learning, which enforces alignment between the learned distribution of the offline dataset and the distribution of optimized decisions. In our study, IOM serves as a reference point similar to COMs, enabling us to evaluate the effectiveness of invariant representation learning in addressing distributional shifts and enhancing performance in the offline MOMO problem.

- **RoMA** (Yu et al., 2021) also addresses the challenge of overestimation issues when approximating true objective scores. To mitigate this issue, RoMA proposes robust model adaptation by incorporating a local smoothness prior as a regularizer. This regularizer aims to enforce a flat loss landscape, thereby enhancing the proxy model's generalization capabilities and ensuring stable training. In our study, RoMA serves as a reference point, similar to COMs and IOM, allowing us to assess the effectiveness of using regularization techniques to improve robustness and performance in the offline MOMO problem.

- **Ensemble proxy** (Trabucco et al., 2022) takes a different offline MBO approach by leveraging multiple proxy models through ensemble learning. This approach addresses the limitations of a single proxy model, which can be prone to overfitting issues. Ensemble proxy uses multiple proxies with different initializations and averages their predictions to approximate true objective scores. In our study, Ensemble proxy serves as a reference point, enabling us to evaluate the effectiveness of ensemble learning in the offline MOMO problem and assess whether the potential performance gains justify the increased computational cost associated with using multiple proxy models.

- **ICT** (Yuan et al., 2023) utilizes multiple proxies, similar to Ensemble proxy, but enhances the approach through a co-teaching process. This process facilitates information exchange between proxies and encourages knowledge transfer. Additionally, ICT incorporates a meta-learning-based sample reweighting mechanism that iteratively updates the importance weights of samples to mitigate potential inaccuracies in pseudo-labels. In our study, ICT serves as a reference point, enabling us to evaluate the effectiveness of advanced ensemble techniques, such as co-teaching and meta-learning, in the offline MOMO problem.

- **Tri-Mentoring** (Chen et al., 2023a) is closely related to ICT, utilizing multiple proxies and facilitating learning between them through a mentoring process. However, Tri-Mentoring shifts its focus to generating pairwise comparison labels rather than directly approximating objective scores. Instead of averaging predictions, it employs majority voting to combine decisions from each proxy model. In our study, Tri-Mentoring serves as a crucial reference point, enabling us to evaluate the effectiveness of using the rank-based proxy over the score-based proxy, aligning closely with the approach of our proxy model.

- **BIB** (Chen et al., 2023b) employs a bidirectional learning approach that utilizes both forward and backward mappings to generate input configurations likely to produce optimal outputs, while conforming to the data distribution of the offline dataset. BIB constructs its proxy model using a pre-trained language model and applies a deep linearization scheme to derive a closed-form loss function. It is recognized as one of the best models for tackling the offline biological sequence design problem. In our study, BIB serves as a reference point to evaluate how well a high-performing method designed for offline biological sequence design performs in the offline MOMO problem.

- **BootGen** (Kim et al., 2023) employs a bootstrapping technique to enhance the optimization process by iteratively augmenting the offline dataset with self-generated data, using the proxy model as a pseudo-labeler. The goal is to align and refine the generative model through iterative training, where high-quality samples are added to the augmented dataset based on the proxy model's guidance. BootGen is also recognized as one of the best models for offline biological sequence design. In our study, BootGen serves as a reference point to evaluate the effectiveness of the bootstrapping technique in offline optimization, and, similar to BIB, to assess how well a high-performing method designed for offline biological sequence design can be adapted to tackle the offline MOMO problem.

- **RaM** (Tan et al., 2025) identifies the limitations of mean squared error loss for offline MBO and instead advocates for a ranking-based model. By employing learning-to-rank (LTR) techniques, RaM prioritizes promising candidates based on their relative ordering rather than absolute scores, achieving state-of-the-art performance across various offline

MBO tasks. While RaM's ranking-based model shares conceptual similarities with the rank-based proxy mechanism in our framework, our work extends beyond RaM's focus by integrating the rank-based proxy as one component within a more comprehensive framework. Specifically, our MolStitch framework combines molecular stitching, rank-based proxy, priority sampling, and preference optimization to tackle the offline MOMO problem in real-world molecular discovery. The novelty of our work lies not in the rank-based proxy itself but in developing a unified framework that leverages this component alongside others. Additionally, we enhance the rank-based proxy through experiments that incorporate multiple proxies to capture diverse priorities. Using priority sampling, each proxy model emphasizes different objectives based on their assigned importance. Our findings indicate that increasing the number of rank-based proxies improves performance, but only up to a certain threshold, revealing a nuanced interplay between model diversity and performance. As a result, our study offers a more comprehensive exploration of rank-based proxy. In our study, RaM serves as a critical reference point for evaluating the effectiveness of ranking-based models, highlighting the advantages of these methods in offline MBO, and offering valuable insights into their broader applicability.

## O. Details on Evaluation Metrics

This section provides an overview of the evaluation metrics used in this study: the hypervolume (HV) indicator (Zitzler et al., 2003) and the R2 indicator (Brockhoff et al., 2012). Both metrics are widely employed in multi-objective optimization due to their effectiveness in evaluating solution quality across conflicting objectives. The HV indicator quantifies the volume of the objective space dominated by the Pareto front relative to a reference point, reflecting convergence and diversity. In contrast, the R2 indicator measures how well the Pareto front aligns with a set of reference directions, assessing solution distribution. Using both metrics together provides complementary insights into the performance of optimization algorithms and the exploration of trade-offs among objectives.

### O.1. Hypervolume Indicator

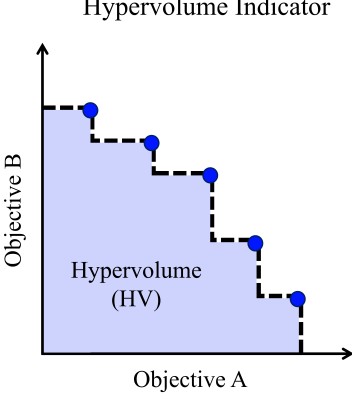

*Figure 17.* Visualization of the hypervolume (HV) indicator in a 2D space, where the HV corresponds to the volume of the shaded region.

The HV indicator denoted as $I_H$, measures the volume in the objective space that is dominated by the Pareto front derived from the optimization algorithm. To be more specific, the HV indicator is defined as the volume in the objective space that is dominated by a set of solutions $\mathcal{X}$ relative to a reference point $z^r$. Of note, the reference point $z^r$ is chosen such that it is dominated by all solutions in $\mathcal{X}$, representing the worst acceptable value for each objective. Mathematically, the HV can be expressed using the Lebesgue integral as follows:

$$I_H(\mathcal{X}, z^r) = \int_{\mathbb{R}^n} \mathbb{I}_{\{z^r | z^r \leq x \text{ for some } x \in \mathcal{X}\}}(z^r) \, dz^r,$$

where $\mathbb{I}$ is the indicator function that equals to 1 if the reference point $z^r \in \mathbb{R}^n$ is dominated by at least one solution $x \in \mathcal{X}$, i.e., $z^r \leq x$ for some $x \in \mathcal{X}$, and 0 otherwise. This formulation essentially measures the volume of the region in the objective space that is dominated by the solutions in $\mathcal{X}$ and bounded above by the reference point $z^r$. Alternatively, the HV

can be calculated more practically as follows:

$$I_H(\mathcal{X}, z^r) = \text{Vol}\left(\bigcup_{x \in \mathcal{X}} [x, z^r]\right),$$

where $[x, z^r]$ denotes the hyperrectangle with lower corner $x$ and upper corner $z^r$. This representation provides a more intuitive understanding of the HV indicator as it directly corresponds to the union of hyperrectangles formed by each solution in $\mathcal{X}$ with respect to $z^r$. In a nutshell, the HV indicator quantifies the size of the objective space that is simultaneously dominated by all solutions in $\mathcal{X}$ and is within the bounds defined by $z^r$. A larger HV value indicates a more preferable set of solutions, as it implies that a greater portion of the objective space is covered by the set $\mathcal{X}$.

To provide a clear understanding, we visualized HV as shown in Figure 17, where the blue points represent a Pareto front composed of non-dominated solutions. Then the HV is defined as a measure of the region in the objective space that is dominated by the Pareto front and bounded by a reference point. In this study, as we have normalized all objective values between $0$ and $1$, we set the reference point as the origin (e.g., $(0,0)$ for two-dimensional space, $(0,0,0)$ for three-dimensional space, and so on) in each respective dimensional space.

## O.2. R2 Indicator

The R2 indicator (Brockhoff et al., 2012) is a set-based performance metric used in multi-objective optimization to evaluate the quality of a set of solutions $\mathcal{X}$ in approximating the true Pareto front. Unlike the HV indicator, which measures the volume of the dominated region, the R2 indicator uses a set of predefined weight vectors to assess how well the solutions in $\mathcal{X}$ represent various trade-offs among objectives. It is defined as the maximum of the worst-case weighted distances between the solutions in $\mathcal{X}$ and an ideal or utopian point. A lower R2 value indicates better performance, as it signifies that the solutions in $\mathcal{X}$ are closer to the ideal point for all considered weight vectors.

Mathematically, let $\mathcal{W}$ be a set of weight vectors $\mathbf{w} = (w_1, w_2, \ldots, w_m)$, where $w_i \geq 0$ and $\sum_{i=1}^{m} w_i = 1$, representing different priorities for the objectives. The R2 indicator, denoted as $R2(\mathcal{X}, \mathcal{W})$, can be defined as:

$$R2(\mathcal{X}, \mathcal{W}) = \max_{\mathbf{w} \in \mathcal{W}} \min_{\mathbf{x} \in \mathcal{X}} \left\{ \sum_{i=1}^{m} w_i \cdot [f_i^*(\mathbf{x}) - f_i(\mathbf{x})] \right\},$$

where $f_i(\mathbf{x})$ is the value of the $i$-th objective for the solution $\mathbf{x}$, and $f_i^*(\mathbf{x})$ is the value of the $i$-th objective for the ideal or utopian point (typically the maximum achievable value for maximization problems). This formulation calculates the deviation of the solution set $\mathcal{X}$ from the ideal point for each weight vector $\mathbf{w}$ and then takes the maximum of these deviations across all weight vectors in $\mathcal{W}$. The use of the maximum operator ensures that the R2 indicator focuses on the worst-case scenario for any given weight vector, reflecting the least favorable trade-off among objectives that the solution set $\mathcal{X}$ can achieve. A lower R2 value means that $\mathcal{X}$ is closer to the ideal point across all weight vectors, indicating a better approximation of the Pareto front.

In summary, the R2 indicator quantifies the worst-case performance of a set of solutions $\mathcal{X}$ in terms of their proximity to an ideal point for a given set of weight vectors $\mathcal{W}$. A lower R2 value is better as it indicates a closer approximation to the ideal performance across all weight vectors.

# P. Further Investigations of StitchNet within Our MolStitch Framework

## P.1. Quantitative Assessment of StitchNet's Ability to Learn Crossover Operations

*Table 19.* Assigned Scores and overall similarity between StitchNet and Crossover operator

|                | High Scoring | Middle Scoring | Low Scoring | Similarity |
|----------------|:------------:|:--------------:|:-----------:|:----------:|
| Assigned Score | 43%          | 31%            | 26%         | 0.644      |

In this section, we present the quantitative results evaluating how effectively StitchNet learns the crossover operation. To assess this, we generated 300 offspring molecules using rule-based crossover operations, and 100 molecules using StitchNet with the same parent molecule pairs. Then, the 300 molecules from rule-based crossover were categorized into three groups based on their mean target objective scores (GSK3$\beta$+JNK3+QED+SA): high-scoring, middle-scoring, and low-scoring. For each group, we calculated the mean Tanimoto similarity score with the 100 molecules generated by StitchNet. Each StitchNet-generated molecule was then assigned to the group with which it exhibited the highest similarity score.

The results, presented in Table 19, demonstrate that the overall similarity scores are reasonable, suggesting that StitchNet effectively learns crossover operations through its unsupervised pre-training process. Notably, molecules generated by StitchNet were most frequently assigned to the top-scoring group, with the lowest assignment to the low-scoring group. This outcome highlights the advantages of StitchNet's self-supervised training, which integrates chemical feedback to guide the generation of stitched molecules with desirable objective scores. As a result, StitchNet can perform crossover operations in a way that preferentially generates offspring molecules with higher objective scores.

## P.2. Effectiveness and Contribution of StitchNet in Comparison to Existing Offline Dataset

*Table 20.* Overall improvement in objective scores when comparing stitched molecules against existing molecules in the offline dataset.

|             | QED    | SA     | JNK3    | GSK3$\beta$ |
|-------------|:------:|:------:|:-------:|:-----------:|
| Improvement | -5.79% | -3.15% | +16.10% | +42.18%     |

To assess the quality of the newly generated molecules from StitchNet, we measured the improvement and non-improvement in objective scores (GSK3$\beta$, JNK3, QED, SA) between the stitched molecules and the existing molecules in the offline dataset. Table 20 presents the results, showing the percentage of improvement and non-improvement. Compared to the existing molecules in the offline dataset, the newly generated molecules from StitchNet exhibited significant increases in challenging objectives such as GSK3$\beta$ and JNK3, while showing slight decreases in easier-to-optimize objectives like QED and SA. This suggests that StitchNet effectively provides diversity beyond the offline dataset and enhances performance in challenging objectives with only a minor reduction in easier objectives. Consequently, the generative model can learn from this enriched set of high-quality molecules generated by StitchNet, leading to an overall improvement in performance.

# Q. Reward Hacking Problem in Offline Multi-Objective Molecular Optimization

In this study, we tackle the offline multi-objective molecular optimization problem, which requires optimizing multiple molecular objectives simultaneously. Throughout this optimization process, we observed conflicts between certain molecular objectives. To investigate these conflicts further, we conducted a detailed analysis of each property score within a four-objective scenario (GSK3$\beta$, JNK3, QED, and SA).

We found that models often prioritized easier objectives, such as QED and SA, over more challenging objectives like GSK3$\beta$ and JNK3. As reported in previous study (Gao et al., 2022), QED is often considered too trivial, allowing most models to achieve high scores on this objective with minimal effort. This suggests that increasing and optimizing the QED score is much simpler compared to tackling more challenging objectives. For instance, models like REINVENT, which receive rewards based on the average property score, may focus on easily attainable objectives to maximize the overall reward. Consequently, this creates the ***reward hacking problem***, where the model overfit to easier objectives while neglecting the more challenging ones. This behavior highlights the inherent difficulty in multi-objective optimization, particularly when some objectives are easier to optimize than others.

One possible approach to address this issue could be adjusting the weights assigned to each objective to balance their influence—placing more emphasis on the challenging objectives and less on the easier ones. However, this approach relies on having prior domain knowledge about the difficulty of each objective, which is not always available. Moreover, in offline optimization settings, immediate feedback to refine weights is limited, making this approach impractical.

To overcome these challenges, we introduced priority sampling using a Dirichlet distribution within our MolStitch framework for Pareto optimization. This approach efficiently generates diverse weight configurations, ensuring a balanced exploration of all objectives. By using priority sampling within our framework, we promote the generation of a diverse set of stitched molecules that do not disproportionately favor easier objectives, thereby mitigating the risk of reward hacking.

*Table 21.* Property scores for each objective in a four-objective scenario (GSK3$\beta$, JNK3, QED, SA).

|  | QED | SA | JNK3 | GSK3$\beta$ |
|---|---|---|---|---|
| w/o MolStitch | **0.843** | **0.889** | 0.128 | 0.397 |
| MolStitch (Ours) | 0.709 | 0.802 | **0.485** | **0.688** |

To validate the effectiveness of our MolStitch framework, we compared the property scores for each objective in a four-objective scenario (GSK3$\beta$, JNK3, QED, and SA) before and after applying our MolStitch framework that incorporates priority sampling. The results, presented in Table 21, clearly indicate that without MolStitch, the models suffer from the reward hacking problem, achieving disproportionately high scores on easier objectives like QED and SA while exhibiting extremely low scores on more challenging objectives such as JNK3 and GSK3$\beta$. In contrast, applying our MolStitch framework results in a more balanced optimization, with relatively improved and well-distributed scores across all objectives.

## R. Chebyshev Scalarization

While our initial implementation of MolStitch employed linear scalarization due to its simplicity and foundational role in multi-objective optimization, we also explored Chebyshev scalarization to assess its potential advantages. Specifically, Chebyshev scalarization minimizes the maximum weighted deviation from a reference point, which is typically defined as the ideal vector in objective space. Unlike linear scalarization, which aggregates all objectives into a single weighted sum, Chebyshev scalarization evaluates each objective independently and focuses on minimizing the worst-performing objective. This mechanism encourages more balanced solutions and enables the algorithm to explore non-convex and concave regions of the Pareto front, which are often inaccessible to linear scalarization techniques (Deb et al., 2016).

To empirically assess the influence of Chebyshev scalarization within our MolStitch, we conducted experiments across two-objective (GSK3$\beta$+JNK3), three-objective (GSK3$\beta$+JNK3+QED), and four-objective (GSK3$\beta$+JNK3+QED+SA) optimization tasks. As shown in Table 22, MolStitch with Chebyshev scalarization achieved comparable performance to the linear approach in the two-objective setting. This is consistent with our expectation that the Pareto front remains relatively simple and convex in low-dimensional spaces, limiting the potential benefits of more sophisticated scalarization techniques. In contrast, Chebyshev scalarization consistently outperformed linear scalarization in both three- and four-objective settings. This improvement is likely due to the increased complexity and non-convexity of the Pareto front in higher dimensions, where Chebyshev's emphasis on extreme deviations facilitates more effective balancing of trade-offs across objectives.

*Table 22.* Additional experimental results on the molecular property optimization task using the Chebyshev scalarization technique.

| Molecular objectives | GSK3$\beta$+JNK3 | | GSK3$\beta$+JNK3+QED | | GSK3$\beta$+JNK3+QED+SA | |
|---|---|---|---|---|---|---|
| Method | HV($\uparrow$) | R2($\downarrow$) | HV($\uparrow$) | R2($\downarrow$) | HV($\uparrow$) | R2($\downarrow$) |
| MolStitch (w/ Linear) | 0.579$\pm$0.070 | **0.698**$\pm$**0.128** | 0.403$\pm$0.065 | 1.649$\pm$0.259 | 0.352$\pm$0.080 | 2.953$\pm$0.571 |
| MolStitch (w/ Chebyshev) | **0.580**$\pm$**0.068** | 0.707$\pm$0.125 | **0.440**$\pm$**0.085** | **1.568**$\pm$**0.297** | **0.397**$\pm$**0.078** | **2.619**$\pm$**0.453** |

We further analyzed the impact of Chebyshev scalarization on molecular diversity in the four-objective setting. Specifically, we measured the number of unique Bemis-Murcko (BM) scaffolds and carbon skeletons—two widely used evaluation metrics of structural diversity—among the molecules located on the Pareto front. As demonstrated in Table 23, Chebyshev scalarization resulted in a notably more diverse set of molecules, as evidenced by the higher counts across both diversity metrics. This outcome further supports the assertion that Chebyshev scalarization promotes broader exploration of the solution space and facilitates the discovery of molecular candidates that might be overlooked by the linear approach.

*Table 23.* Comparative analysis of diversity outcomes using Molstitch with linear and Chebyshev scalarization .

| Molecular objectives | GSK3$\beta$+JNK3+QED+SA | |
| --- | --- | --- |
| Diversity metrics | BM scaffold | Carbon skeletons |
| MolStitch (w/ Linear) | 3453 | 1664 |
| MolStitch (w/ Chebyshev) | **3836** | **1976** |

In summary, these findings collectively demonstrate that MolStitch is compatible with both linear and Chebyshev scalarization techniques. We think that linear scalarization remains a reasonable choice when the Pareto front is expected to be convex or when one objective is known to be dominant. However, Chebyshev scalarization is better suited for high-dimensional optimization tasks involving multiple, conflicting objectives. In such scenarios, Chebyshev can provide more robust trade-off among competing objectives and leads to the generation of more diverse and well-balanced molecular candidates.

## S. Extended Analysis on Molecular Property and Docking Score Optimization Tasks

### S.1. Additional Experimental Results on the Molecular Property Optimization Task

Given the relevance of the offline MOO benchmark (Xue et al., 2024) to our study, we conducted additional experiments incorporating baseline methods introduced in that work, specifically the use of multiple models in conjunction with COMs and RoMA. The underlying idea of leveraging multiple models closely aligns with our ensemble proxy model setup in MolStitch. As shown in Table 24, the Multiple Models + COMs/RoMA configurations outperformed their single-model counterparts. This result is consistent with our findings throughout the paper, where ensemble-based proxy models generally demonstrated superior performance relative to single-model approaches such as Grad.

*Table 24.* Additional experimental results on the molecular property optimization task, including baselines from the **offline MOO** paper.

| Molecular objectives | GSK3$\beta$+JNK3 | | GSK3$\beta$+JNK3+QED | | GSK3$\beta$+JNK3+QED+SA | |
| --- | --- | --- | --- | --- | --- | --- |
| Method | HV($\uparrow$) | R2($\downarrow$) | HV($\uparrow$) | R2($\downarrow$) | HV($\uparrow$) | R2($\downarrow$) |
| Single Model + COMs | 0.479$\pm$0.063 | 0.877$\pm$0.109 | 0.205$\pm$0.072 | 2.496$\pm$0.288 | 0.171$\pm$0.062 | 4.219$\pm$0.628 |
| Single Model + RoMA | 0.492$\pm$0.091 | 0.843$\pm$0.177 | 0.198$\pm$0.052 | 2.537$\pm$0.269 | 0.169$\pm$0.071 | 4.207$\pm$0.617 |
| Multiple Models + COMs | 0.489$\pm$0.089 | 0.814$\pm$0.117 | 0.211$\pm$0.095 | 2.449$\pm$0.407 | 0.190$\pm$0.061 | 4.157$\pm$0.592 |
| Multiple Models + RoMA | 0.499$\pm$0.081 | 0.812$\pm$0.141 | 0.214$\pm$0.050 | 2.472$\pm$0.247 | 0.188$\pm$0.073 | 3.988$\pm$0.533 |
| REINVENT-BO | 0.472$\pm$0.107 | 0.909$\pm$0.216 | 0.232$\pm$0.086 | 2.385$\pm$0.393 | 0.205$\pm$0.105 | 3.974$\pm$0.895 |
| MolStitch (Ours) | **0.579$\pm$0.070** | **0.698$\pm$0.128** | **0.403$\pm$0.065** | **1.649$\pm$0.259** | **0.352$\pm$0.080** | **2.953$\pm$0.571** |

In addition, motivated by the strong performance of Bayesian optimization (BO) reported in the offline MOO benchmark, we implemented a comparable BO approach using REINVENT as the generative model, which we refer to as REINVENT-BO. As shown in the same table, REINVENT-BO achieved competitive results, indicating its viability as a baseline method for offline molecular optimization tasks. Nevertheless, our MolStitch framework consistently outperformed all these baseline methods, thereby reaffirming the robustness and efficacy of our approach.

### S.2. Advancing MolStitch with State-of-the-Art Techniques for the Molecular Property Optimization Task

To further enhance the capabilities of MolStitch, we explored the integration of both well-established and state-of-the-art techniques, including simulated annealing (Van Laarhoven et al., 1987) and the fragment-RAG(Lee et al., 2024) methods. First, simulated annealing (SA) is a probabilistic optimization algorithm that improves exploration by allowing the acceptance of suboptimal solutions with a certain probability in order to escape local optima. Inspired by this principle, we introduced an SA mechanism into MolStitch by allowing the occasional acceptance of losing molecules during training. As shown in Table 25, MolStitch with SA did not lead to significant performance gains. We attribute this marginal benefit of additional exploration through SA to the effectiveness of our rank-based proxy model in reliably distinguishing between candidate molecules, as well as the intrinsic diversity already promoted by the StitchNet architecture.

We also investigated incorporating ideas from fragment-RAG (f-RAG), which is a recent state-of-the-art framework that introduces the use of hard fragments—explicit structural components used to construct new molecules—and soft fragments, which are injected as embeddings to implicitly guide the generation process. Since StitchNet naturally supports the principles

*Table 25.* Additional experimental results on the molecular property optimization task, integrating well-established and recent techniques.

| Molecular objectives | GSK3$\beta$+JNK3 | | GSK3$\beta$+JNK3+QED | | GSK3$\beta$+JNK3+QED+SA | |
|---|---|---|---|---|---|---|
| Method | HV($\uparrow$) | R2($\downarrow$) | HV($\uparrow$) | R2($\downarrow$) | HV($\uparrow$) | R2($\downarrow$) |
| MolStitch | **0.579**$\pm$**0.070** | 0.698$\pm$0.128 | 0.403$\pm$0.065 | 1.649$\pm$0.259 | 0.352$\pm$0.080 | 2.953$\pm$0.571 |
| MolStitch (w/ SA) | 0.542$\pm$0.107 | 0.774$\pm$0.200 | 0.394$\pm$0.054 | 1.669$\pm$0.162 | 0.361$\pm$0.040 | 2.704$\pm$0.276 |
| MolStitch (w/ f-RAG) | 0.578$\pm$0.069 | **0.694**$\pm$**0.115** | **0.451**$\pm$**0.081** | **1.519**$\pm$**0.277** | **0.412**$\pm$**0.060** | **2.494**$\pm$**0.356** |

of hard fragments through explicit recombination of molecular substructures, it readily accommodates the first component of f-RAG. To incorporate soft fragment guidance, we extended StitchNet to condition on fragment embeddings during the stitching process, enabling implicit control over molecule generation. As demonstrated in Table 25, MolStitch with f-RAG yielded improved performance, highlighting the benefits and effectiveness of soft fragment guidance.

### S.3. Additional Experimental Results on the Docking Score Optimization Task using SMINA

In our main study, we employed QuickVina (QVina) (Alhossary et al., 2015) for the docking score optimization task, which is a widely recognized molecular docking tool derived from AutoDock Vina. Our decision to use QVina was motivated by its widespread adoption and following established practices from prior studies (Lee et al., 2023; Guo & Schwaller, 2024b). While QVina is widely used, other molecular docking tools derived from AutoDock Vina have been developed, among which SMINA (Cieplinski et al., 2023) is particularly notable. Specifically, SMINA offers extended capabilities, including support for customized scoring functions and the ability to enforce specific ligand–receptor interaction constraints.

*Table 26.* Additional experimental results on docking score optimization tasks using **SMINA** as the docking evaluation tool.

| Target protein | parp1 | jak2 | braf | fa7 | 5ht1b |
|---|---|---|---|---|---|
| Method | HV($\uparrow$) | HV($\uparrow$) | HV($\uparrow$) | HV($\uparrow$) | HV($\uparrow$) |
| REINVENT | 0.522$\pm$0.007 | 0.476$\pm$0.014 | 0.503$\pm$0.006 | 0.417$\pm$0.004 | 0.514$\pm$0.003 |
| BootGen | 0.534$\pm$0.006 | 0.498$\pm$0.009 | 0.518$\pm$0.020 | 0.425$\pm$0.007 | 0.530$\pm$0.016 |
| MolStitch (Ours) | **0.550**$\pm$**0.007** | **0.539**$\pm$**0.060** | **0.530**$\pm$**0.012** | **0.450**$\pm$**0.004** | **0.544**$\pm$**0.004** |

*Table 27.* Additional experimental results on docking score optimization tasks using **SMINA** as the docking evaluation tool.

| Target protein | parp1 | jak2 | braf | fa7 | 5ht1b |
|---|---|---|---|---|---|
| Method | R2($\downarrow$) | R2($\downarrow$) | R2($\downarrow$) | R2($\downarrow$) | R2($\downarrow$) |
| REINVENT | 1.408$\pm$0.021 | 1.589$\pm$0.058 | 1.487$\pm$0.025 | 1.804$\pm$0.029 | 1.441$\pm$0.030 |
| BootGen | 1.362$\pm$0.044 | 1.502$\pm$0.044 | 1.444$\pm$0.066 | 1.772$\pm$0.032 | 1.376$\pm$0.080 |
| MolStitch (Ours) | **1.323**$\pm$**0.019** | **1.325**$\pm$**0.177** | **1.397**$\pm$**0.046** | **1.627**$\pm$**0.026** | **1.329**$\pm$**0.022** |

Given its relevance and growing adoption, we conducted additional docking score optimization experiments using SMINA to further evaluate the robustness of our MolStitch framework. As shown in Tables 26 and 27, MolStitch consistently maintained superior performance in terms of both HV and R2 evaluation metrics, even when assessed using SMINA. These findings further validate the generalizability and robustness of our MolStitch across distinct molecular docking tools.

## T. Future Work and Limitations

In this study, we focused on optimizing the properties of small molecules and docking scores for five specific proteins. A natural extension of this work would be to apply our framework to material discovery, particularly for optimizing inorganic molecules, thereby broadening its applicability beyond small molecules. Additionally, while we investigated both full-offline and semi-offline optimization settings, there remains considerable potential to enhance the semi-offline optimization. One promising direction is the use of a behavior policy to improve exploration of chemical space when periodically incorporating new molecule data. This strategy would enable the inclusion of molecules that were not present in the initial offline dataset, leading to more effective integration of newly obtained data. Even in cases where the initial offline dataset contains lower-quality molecules, a behavior policy could progressively improve the quality of the data over time. Moreover, our results suggest that employing multiple proxies yields valuable insights and substantial performance gains in specific cases. As such, future work will focus on further developing and optimizing multiple proxy methods to fully realize their potential in molecular discovery.

## U. Molecule Examples

In this section, we provide visual examples of molecules generated by the fine-tuned generative model, which aims to produce novel molecules that surpass the best-known molecules in the offline dataset. Specifically, we present representative molecules sampled from the Pareto front in the four-objective optimization scenario (QED+SA+JNK3+GSK3$\beta$). Each molecule illustrates a distinct trade-off among these objectives, demonstrating the diverse range of solutions on the Pareto front. These examples emphasize the ability of our framework to explore diverse molecules that effectively balance multiple objectives.

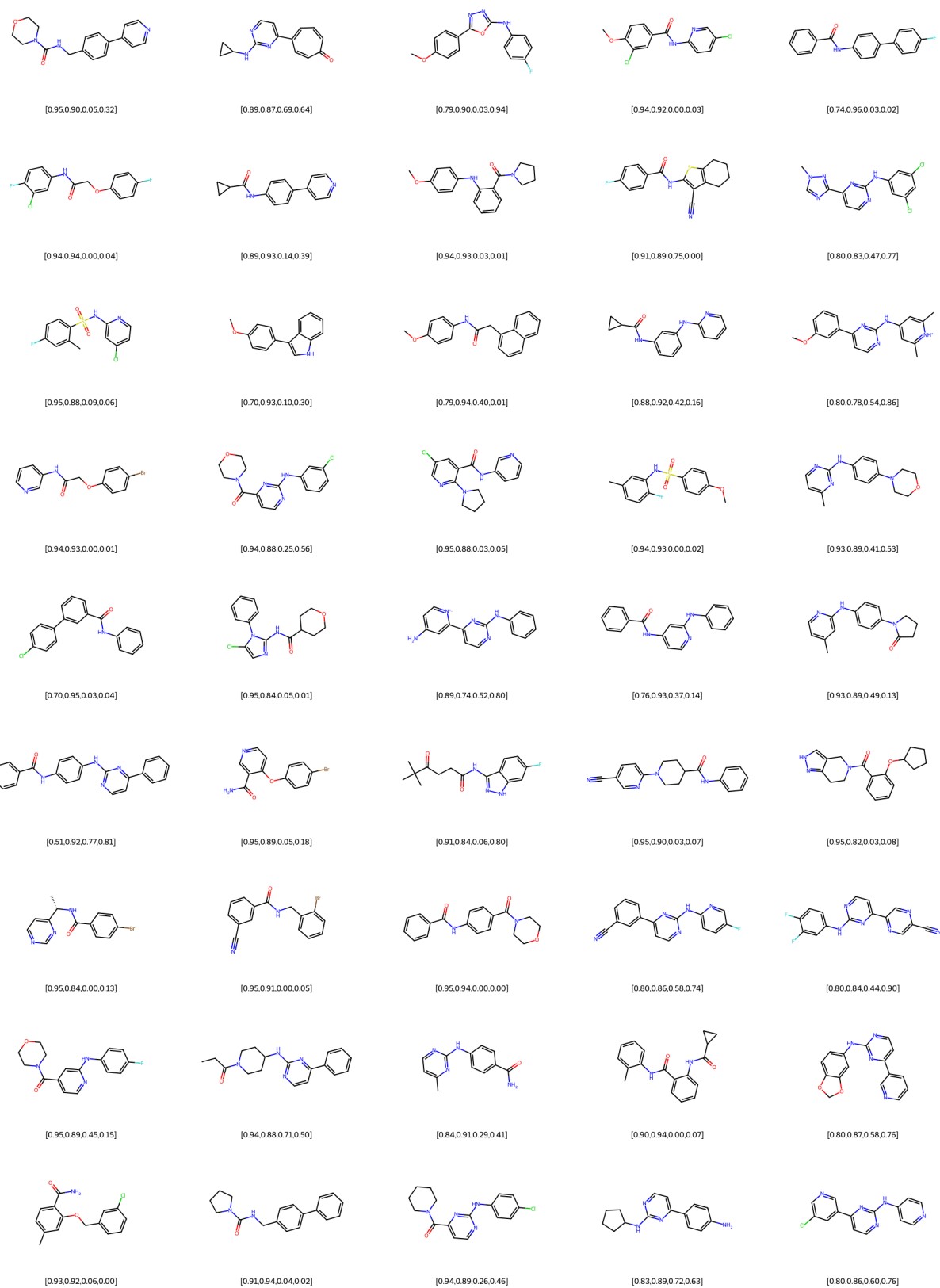

*Figure 18.* Representative molecules sampled from the Pareto front in the four-objective optimization scenario (QED+SA+JNK3+GSK3$\beta$). The numerical scores for each objective are displayed below the respective molecular structures. Each molecule reflects a distinct trade-off among these objectives, highlighting the diverse range of solutions on the Pareto front.

