# OpenReview forum: "Offline Model-based Optimization for Real-World Molecular Discovery"
_ICML.cc/2025/Conference — ICML 2025 poster_

### Official Review · Reviewer_RQBY · 2025-03-08

**Overall Recommendation:** 4

**Summary:**

The authors propose MolStitch as a generative method to generate molecular designs in an offline, multi-objective setting. Their method generates novel 'stitched molecules' that combine the desirable properties of original molecules sampled from the offline dataset. The authors evaluate their method on a number of offline molecular design tasks.

**Claims And Evidence:**

The claims made by the authors are clear and supported by convincing evidence.

**Essential References Not Discussed:**

1. In general, I think most of the essential references have been discussed or included as relevant baselines. Additional references that I think would strengthen the experimental results include:

  - [Simulated Annealing](https://en.wikipedia.org/wiki/Simulated_annealing)
  - [Fragment-RAG](https://arxiv.org/abs/2411.12078) from Lee et al. Proc NeurIPS (2024).
  - [DyNA-PPO](https://openreview.net/forum?id=HklxbgBKvr) from Angermueller et al. Proc ICLR (2020).
  - [GFNSeqEditor](https://openreview.net/forum?id=g0G8DQSBcj) from Ghari et al. Proc NeurIPS (2024).

That being said, I'm well aware that there are many offline MBO algorithms now proposed in the literature, and I think the authors have already demonstrated experimentally that their method works across a wide variety of different tasks. I would more strongly encourage Simulated Annealing and Fragment-RAG to be included as baselines given their similarity with the authors' proposed method. I feel less strongly that DyNA-PPO and GFNSeqEditor would need be included as baselines - appropriate discussion in the Related Work (if not already included) is likely more than sufficient.

**Experimental Designs Or Analyses:**

The experimental design and analysis used by the authors are sound and use standard metrics from the offline optimization and multi-objective optimization literature (e.g., diversity of designs, Pareto fronts, hyper volume indicator, etc.

**Methods And Evaluation Criteria:**

The molecular design tasks that the authors use to evaluate their proposed method are standard and representative of real-world molecular design tasks. I think this proposed methods and chosen evaluation benchmarks are well-motivated to study the problem proposed by the authors.

**Other Comments Or Suggestions:**

None

**Other Strengths And Weaknesses:**

In general, I think this is a well-motivated, well-executed, and well-written submission and lean towards recommending acceptance of this work. There are some additional experiments and associated discussion I recommend that would potentially help strengthen the paper that I detail in my earlier comments, but overall, I think this submission is sound.

### Strengths

2. The idea to use a DPO-/IPO- like framework for "synthetic" molecule priority sampling is interesting, original, and significant to the best of my knowledge.
3. I appreciate the inclusion of batch hybrid learning results using the MolStitch method in the Appendix - this is a very real-world problem formulation and I the strong results of the authors' method in this setting strengthens the contributions of this paper.

### Weaknesses

4. The authors propose a method of objective scalarization via sampling from the Dirichlet distribution. A number of other method exist for scalarization - notably [Chebyshev scalarization](https://arxiv.org/abs/1904.05760) and even uniform sampling - that would be worth including as an ablation study.

**Questions For Authors:**

5. In line 147, the authors assume that the offline dataset $\mathcal{D}$ contains all of the evaluated objective scores for each of the $k$ objectives. In practice, I would imagine that the majority of molecules would only be experimentally evaluated using only a subset of the $k$ objectives in building the offline dataset. How would the method proposed by the authors (or how have others in prior work) adapt to this setting?
6. Could the authors provide some additional details regarding the rule-based crossover operator using in the unsupervised pre-training stage? I am having a hard time understanding how this process should encourage StitchNet to internalize chemical grammar. More explicitly, my understanding is that molecules can have very similar token representations but represent very different molecules, and similar molecules in token space may have very different validity scores.
7. In Section 3.2, the authors mention that they use the oracle score of $m\_{orig}$ as an approximation for $\bar{m}\_{stit}$ because the stitched molecule shares the same molecular fragments as the original molecule. However, I would imagine that there might be functions that depend on the properties of the global molecule, or how the fragments are positioned with respect to one another in the molecule - such properties would be lost through the stitching process and make the approximation that $\mathcal{R}$ is similar for the stitched and original molecules invalid. Is this the case? This is more so a minor clarification question on my part - I understand that the authors have cited prior work to support their argument (lines 215-217, right column), but am not as familiar with these prior literature.
8. Is the loss function in Equation (9) indeed a summation over $m\_{orig}\in\mathcal{D}$, or an expectation value?

**Relation To Broader Scientific Literature:**

In general, I think MolStitch is a meaningful contribution to the offline optimization for molecular discovery literature. The idea of using IPO for preference fine-tuning builds off of [IPO](https://arxiv.org/abs/2310.12036) and [DPO](https://arxiv.org/abs/2305.18290) from the language model literature, and the idea of trajectory annealing/stitching has been explore in prior work (e.g., [DiffStitch](https://arxiv.org/abs/2402.02439), [SSD](https://arxiv.org/abs/2402.07226), [GFNSeqEditor](https://openreview.net/forum?id=g0G8DQSBcj), [Fragment-RAG](https://arxiv.org/abs/2411.12078), Simulated Annealing). The authors also compare their method against relevant baselines in their experimental work.

**Theoretical Claims:**

There are no theoretical claims to check the correctness of.

---

> ### Author Rebuttal · Authors · 2025-04-01
>
> We appreciate your helpful feedback and the opportunity to enhance our manuscript.
> # Q1: Additional references would strengthen the experimental results.
> First, regarding **Simulated Annealing (SA)**, it is a probabilistic optimization algorithm that enhances exploration by occasionally accepting worse solutions with a certain probability. Inspired by this, we incorporated an SA mechanism into our MolStitch by occasionally accepting a losing molecule with a certain probability. Due to character limitations, **we have uploaded the full result tables at the following link** [here](https://tinyurl.com/molstitch). As shown in Table 5, MolStitch w/ SA did not yield noticeable performance improvements. We hypothesize this is because our rank-based proxy already performs well in determining winning and losing molecules, and the additional exploration facilitated by SA is somewhat redundant given StitchNet's diversity.
>
> Second, we found **Fragment-RAG (f-RAG)** to be interesting. Although the official codebase was not available, we attempted to incorporate its core ideas into our framework. Specifically, f-RAG introduces the use of hard fragments—explicit structural components used to construct new molecules—and soft fragments, which are injected as embeddings to implicitly guide generation. StitchNet naturally supports hard fragments through explicit recombination of molecular substructures. To incorporate soft fragment guidance, we extended StitchNet to condition on embeddings of soft fragments during the stitching process. MolStitch w/ f-RAG achieved improved performance, highlighting the benefits of soft fragment guidance.
>
> Third and fourth, while **DyNA-PPO and GFNSeqEditor** are primarily designed for biological sequence design, they offer valuable insights from a model-based optimization perspective. We will include these important references in the related work section.
> # Q2: Include an ablation study with Chebyshev scalarization.
> Thank you for this valuable suggestion. A similar point was also raised by another reviewer, which prompted us to conduct additional experiments. For a detailed discussion of the results, we respectfully refer you to our response to `Reviewer B6bd, Q1`.
> # Q3: In practice, most molecules are evaluated on only a subset of objectives.
> Thank you for this insightful question. Several strategies have been proposed in the literature to address this challenge.
> * One direct approach is **imputation methods**, which estimate missing values from available data [1] or via pseudo-labeling in a semi-supervised manner [2].
> * Another strategy is **multi-task learning** [3], where models are trained to jointly tackle multiple objectives while allowing for missing labels. By sharing knowledge across related objectives, these models can leverage observed objectives to inform the learning of others.
>
> Extending our MolStitch to handle missing objective values would be an intriguing direction for future work. We will include these considerations in the limitations and future work section.
> # Q4: Additional details regarding the rule-based crossover operator.
> We apologize for any confusion. The rule-based crossover operator ensures that child molecules generated from two parent molecules are chemically valid by following predefined chemical rules and constraints (e.g., SMARTS templates). Consequently, the goal of the unsupervised pre-training stage is to train StitchNet to imitate this rule-based crossover operator using a maximum likelihood estimation (MLE) objective, similar to teacher forcing.
>
> In practice, we first generate numerous (parent1, parent2, child) triplets using the rule-based crossover operator. These serve as training examples for StitchNet, which is designed to generate child stitched molecules from given pairs of parent molecules. Although StitchNet may initially produce invalid stitched molecules, it gradually learns to imitate the rule-based crossover through MLE. Therefore, as pre-training progresses, StitchNet becomes increasingly proficient at generating chemically valid stitched molecules.
> # Q5: Assumptions for approximating objective scores of 𝑚̄ₛₜᵢₜ.
> Thank you for this important question. A similar concern was also raised by another reviewer, and we respectfully refer you to our response to `Reviewer tueg, Q3`. Briefly, while we acknowledge the limitations, we tried to mitigate them by enforcing a similarity threshold to ensure stitched molecules retain sufficient structural overlap.
> # Q6: Is the loss function in Eq. (9) a summation or an expectation?
> Thanks for the clarification. Eq.(9) is written as a sum over samples from the finite offline dataset.
>
> [1] Lobato et al. “Multi-objective genetic algorithm for missing data imputation.” Pattern Recognit Lett (2015).
>
> [2] Huang et al. “Offline data-driven evolutionary optimization based on tri-training.” Swarm Evol Comput (2021).
>
> [3] Liu et al. “Structured multi-task learning for molecular property prediction.” AISTATS (2022).

---

> > ### Comment · Reviewer_RQBY · 2025-04-01
> >
> > I thank the authors for their hard work on their rebuttal and manuscript overall. All of my concerns have been sufficiently addressed, and I maintain my initial rating of 4 to indicate that I am in favor of accepting this work.

---

> > > ### Author Response · Authors · 2025-04-04
> > >
> > > We sincerely appreciate your time and effort in reviewing our work. Your valuable feedback has significantly helped us improve our manuscript.

---

### Official Review · Reviewer_tueg · 2025-03-14

**Overall Recommendation:** 3

**Summary:**

This paper introduces the Molecular Stitching (MolStitch) framework, designed to address the molecular discovery problem in an offline setting, where an offline dataset is employed without requiring iterative queries to the oracle function. Particularly, MolStitch operates by leveraging existing molecules from the offline dataset to generate novel stitched molecules that combine desirable properties using the StitchNet model. A rank-based proxy model is then employed to compare molecules in the stitched set, determining which is preferable in each pair. This information is used to fine-tune the generative model through Identity Preference Optimization (IPO). The effectiveness of the MolStitch framework is demonstrated through two key offline MOMO experiments, showcasing its potential in molecular optimization.

**Claims And Evidence:**

The claims in this paper are supported by clear and convincing evidence.

**Essential References Not Discussed:**

No essential related works are absent that are crucial for understanding the key contributions of this paper.

**Experimental Designs Or Analyses:**

I have verified the soundness and validity of the experimental designs and analyses.

**Methods And Evaluation Criteria:**

The proposed method and evaluation criteria are well-suited to the problem.

**Other Comments Or Suggestions:**

It is a minor point, but the authors could consider rearranging the order of stages. For example, Stage 1 could include the pretraining of the generative model and StitchNet, followed by the two proposed stages.

**Other Strengths And Weaknesses:**

Strengths:

- Pre-training on the ZINC dataset helps StitchNet internalize chemical grammar, allowing it to generate chemically valid stitched molecules.
- Introduces a rank-based proxy for molecule evaluation instead of traditional value-based methods, followed by preference optimization to fine-tune the generative model.
- The writing is clear and well-structured.
- Empirical results support the effectiveness of the proposed method.

Weaknesses:

- The paper lacks a related work section in the main text.
- While the mean performance over 10 different seeds is strong, the variance is high.

**Questions For Authors:**

1. The authors should consider including and briefly describing recent works [1] [2] [3] [4] [5] on offline optimization in the Related Works section, particularly [1], as it shares a similar idea of training a rank-based proxy model.

2. Where is the pre-training process of the generative model described in this paper? It would be helpful to clarify this aspect.

3. Using the objective scores of $m_{orig}$ as chemical feedback to approximate the objective scores of $\bar{m}_{stit}$ in Eq.(9) seems questionable. Even minor structural modifications to a molecule can lead to significant variations in certain properties. The authors should provide justification or additional validation for this approximation.

[1] Tan, Rong-Xi, Ke Xue, Shen-Huan Lyu, Haopu Shang, Yao Wang, Yaoyuan Wang, Sheng Fu, and Chao Qian. "Offline Model-Based Optimization by Learning to Rank." arXiv preprint arXiv:2410.11502 (2024).

[2] Dao, Manh Cuong, Phi Le Nguyen, Thao Nguyen Truong, and Trong Nghia Hoang. "Boosting offline optimizers with surrogate sensitivity." arXiv preprint arXiv:2503.04181 (2025).

[3] Nguyen, Tung, Sudhanshu Agrawal, and Aditya Grover. "Expt: Synthetic pretraining for few-shot experimental design." Advances in Neural Information Processing Systems 36 (2023): 45856-45869.

[4] Hoang, Minh, Azza Fadhel, Aryan Deshwal, Janardhan Rao Doppa, and Trong Nghia Hoang. "Learning surrogates for offline black-box optimization via gradient matching." arXiv preprint arXiv:2503.01883 (2025).

[5] Chemingui, Yassine, Aryan Deshwal, Trong Nghia Hoang, and Janardhan Rao Doppa. "Offline model-based optimization via policy-guided gradient search." In Proceedings of the AAAI Conference on Artificial Intelligence, vol. 38, no. 10, pp. 11230-11239. 2024.

**Relation To Broader Scientific Literature:**

This paper addresses the offline multi-objective molecular optimization (MOMO) problem, which has promising applications in drug discovery and molecular design.

**Theoretical Claims:**

This paper does not include any theorems.

---

> ### Author Rebuttal · Authors · 2025-04-01
>
> We are truly grateful for your thoughtful feedback. In the following, we carefully respond to each of your comments.
> # Q1: The authors should include recent works
> We sincerely thank the reviewer for highlighting these important references [1–5], which are indeed highly relevant from the perspective of offline optimization. We fully agree with the significance of these works and will ensure they are properly cited and discussed in the revised manuscript.
>
> Regarding the method presented in [1], RaM, we appreciate the reviewer’s suggestion to emphasize its connection to our work. While we had included RaM as a competing baseline and compared its performance with our MolStitch framework in Tables 1 and 2 of the main manuscript (with additional details in Appendix N), we acknowledge that we had not discussed it explicitly in the related work section.
>
> We are grateful for the reviewer’s suggestion and will make sure to address this omission in the revision to provide a more complete overview of related work. Additionally, since an extra page is permitted for the revised manuscript, we plan to move the main related work section into the main text, while retaining the extended discussion in Appendix.
>
> # Q2: Where is the pre-training process of the generative model described in this paper?
> In this work, we used REINVENT as the main backbone generative model, both for our MolStitch and for all baseline offline optimization methods to ensure fair comparison. REINVENT is an RL-based generative model that produces molecules in an auto-regressive manner.
>
> As part of its standard training pipeline, REINVENT first undergoes a pre-training process, where it learns to generate chemically valid molecules by capturing the underlying chemical grammar. This pre-training process is critical to ensure that the model can produce syntactically valid molecular structures before any optimization takes place.
>
> Following pre-training, REINVENT is further optimized via reinforcement learning, where it receives feedback in the form of reward signals based on the desired molecular objectives. This two-stage pipeline—pre-training followed by fine-tuning—has proven highly effective in the molecular domain, enabling REINVENT to achieve robust performance across a range of molecular optimization tasks.
> We acknowledge that our original manuscript did not provide sufficient detail about the pre-training process for the generative model, and we will include a more comprehensive explanation in the revised manuscript.
>
> # Q3: Even minor structural modifications to a molecule can lead to significant variations in certain properties.
> In our study, StitchNet is trained through self-supervised learning by decomposing a single molecule into two fragments and then recombining them. Because the fragments originate from the same parent molecule, the stitched molecule is likely to preserve essential substructures (core scaffolds) that strongly influence molecular properties. Moreover, our approach is supported by the Similar Property Principle (SPP) [6], a foundational concept in drug discovery and QSAR research, which states that structurally similar molecules often exhibit similar properties. To further ensure sufficient structural similarity, we employed Tanimoto similarity metric between original molecules ($m_{orig}$) and their stitched counterparts ($\bar{m}_{stit}$). Prior research [7] has indicated that molecules with a Tanimoto similarity greater than 0.887 commonly demonstrate similar biological activities.
>
> However, we acknowledge exceptions to this principle, particularly in cases involving stereochemistry or activity cliffs, where minor structural changes can lead to major shifts in molecular properties. To address these limitations, we plan to incorporate advanced fingerprints—such as 3D-aware or chirality-aware descriptors—that capture more detailed structural and spatial information. We will include these considerations in the limitations and future work section of our revised manuscript.
>
> # Q4: Authors could consider rearranging the order of stages.
> Thank you for the helpful suggestion. We agree that rearranging the order of stages would improve the logical flow, and we will revise it accordingly.
>
> [1] Tan et al. “Offline Model-Based Optimization by Learning to Rank.” ICLR (2025).
>
> [2] Dao et al. “Boosting offline optimizers with surrogate sensitivity.” ICML (2024).
>
> [3] Nguyen et al. “Expt: Synthetic pretraining for few-shot experimental design.” NeurIPS (2023).
>
> [4] Hoang et al. “Learning surrogates for offline black-box optimization via gradient matching.” ICML (2024).
>
> [5] Chemingui et al. “Offline model-based optimization via policy-guided gradient search.” AAAI (2024).
>
> [6] O’Boyle, Sayle. “Comparing structural fingerprints using a literature-based similarity benchmark.” J Cheminform (2016).
>
> [7] Cheng et al. “Investigating the correlations among chemical structures, bioactivity profiles and molecular targets.” Bioinformatics (2010).

---

> > ### Comment · Reviewer_tueg · 2025-04-05
> >
> > Thank you for the detailed responses. My concerns have been resolved, and I will support accepting the paper.

---

> > > ### Author Response · Authors · 2025-04-07
> > >
> > > We are truly grateful for the time and effort you dedicated to reviewing our work. Your valuable comments provided us with many insights and significantly helped us enhance the quality of our manuscript. Thank you once again for your thoughtful feedback.

---

### Official Review · Reviewer_fpEX · 2025-03-15

**Overall Recommendation:** 4

**Summary:**

The paper introduces MolStitch, a framework for offline multi-objective molecular optimization (MOMO). Key contributions include StitchNet, which generates "stitched molecules" by combining fragments from an offline dataset; a rank-based proxy model for pairwise molecule evaluation; and preference optimization techniques (e.g., IPO) for fine-tuning the generative model. Priority sampling via a Dirichlet distribution is used to explore diverse trade-offs among objectives. Experiments on molecular property (MPO) and docking score optimization tasks demonstrate improvements in hypervolume (HV) and R2 metrics over baselines, with ablation studies validating the framework’s components. The method addresses challenges in offline settings where wet-lab evaluations are costly and slow.

## update after rebuttal

The authors have addressed my comments including docking score optimization experiments using SMINA, and clarification on how StitchNet, the rank-based proxy contributes to final performance. Therefore, I will raise my score to accept.

**Claims And Evidence:**

The claims are supported by experiments across multiple tasks, ablation studies, and comparisons to diverse baselines. However, the evidence has limitations:
1. Benchmark coverage: it is not clear why recent docking score benchmark SMINAare not included.

2. Component contributions: The improvements from StitchNet vs. other data augmentation methods (e.g., crossover operators) are shown, but the analysis lacks depth in explaining why neural-based stitching outperforms other genetic algorithm alternatives such as Saturn.

3. Rank-based proxy: The proxy’s superiority over score-based variants is demonstrated via accuracy metrics, but its impact on downstream optimization (beyond pairwise classification) is not thoroughly analyzed.

**Essential References Not Discussed:**

- SMINA docking tool: Critical for docking score benchmarks, as highlighted in Ciepliński et al.’s work.
- Recent preference-based molecular design**: Works like below apply preference optimization to molecular optimization but are not cited.

1. Extracting medicinal chemistry intuition via preference machine learning
Oh-Hyeon Choung, Riccardo Vianello, Marwin Segler, Nikolaus Stiefl & José Jiménez-Luna
Nature Communications volume 14, Article number: 6651 (2023)

2. Preference Optimization for Molecular Language Models

**Experimental Designs Or Analyses:**

Semi-offline experiments: Results are deferred to the appendix, limiting insight into this practically relevant setting.
- Backbone model: While REINVENT is justified as a backbone, newer architectures (e.g., GFlowNets, Mamba) are only briefly discussed in appendices.

**Methods And Evaluation Criteria:**

The methods are sensible for offline MOMO, leveraging stitching and preference optimization. However:
- Benchmark choice: The omission of SMINA (a docking benchmark not discussed in related work) is unexplained, weakening the docking score evaluation’s credibility.
- Metrics: HV and R2 are appropriate for multi-objective tasks, but the paper does not clarify why these metrics were prioritized over others (e.g., success rate, diversity scores).

**Other Comments Or Suggestions:**

The paper presents a technically sound framework with thorough empirical validation, but the incremental nature of contributions and benchmark omissions temper enthusiasm. Addressing the questions above could strengthen the case for acceptance.

**Other Strengths And Weaknesses:**

Strengths: The integration of stitching, rank-based evaluation, and preference optimization is novel. The semi-offline experiments and diversity analysis (e.g., Bemis-Murcko scaffolds) are practical contributions.

Weaknesses: The components (stitching, ranking, priority sampling) are largely adaptations of existing ideas, limiting conceptual novelty. The writing is dense, with insufficient intuition for non-experts.

**Questions For Authors:**

1. Benchmark justification: Why was SMINA not used for docking evaluation, despite its relevance in the cited benchmark?

2. StitchNet vs. crossover: How does StitchNet’s neural approach fundamentally differ from genetic algorithm-based crossover operators in exploring chemical space?

3. Component dominance: Are the gains primarily from StitchNet, the rank-based proxy, or their combination? Ablation suggests synergy, but a breakdown would clarify.

**Relation To Broader Scientific Literature:**

The work connects to offline MBO and preference optimization literature but could better contextualize:
- Trajectory stitching: The analogy to RL trajectory stitching is under-explored; differences in molecular vs. RL state spaces are not discussed.
- Preference optimization: While DPO/IPO are applied, their adaptation to molecular design (vs. language models) lacks critical analysis and insights.

**Theoretical Claims:**

The paper does not present theoretical claims, focusing instead on empirical validation. No theoretical issues are noted.

---

> ### Author Rebuttal · Authors · 2025-04-01
>
> We sincerely appreciate your insightful feedback and the opportunity to clarify the key aspects of our study.
> # Q1: Why was SMINA not used for docking evaluation?
> In our original study, we employed QuickVina (QVina) [1] for docking score evaluation, which is a widely recognized molecular docking tool derived from AutoDock Vina. Our decision to use QVina was motivated by its widespread adoption and following established practices from prior studies [2,3].
>
> We appreciate your mention of SMINA [4]. Upon investigation, we found that SMINA is also derived from AutoDock Vina and provides additional capabilities, such as imposing constraints on ligand interactions. Due to its high relevance and in response to your valuable suggestion, we conducted additional docking score optimization experiments using SMINA. Due to character limitations, we have uploaded the full results tables at the following link [here](https://tinyurl.com/molstitch).
>
> As demonstrated in the table, **MolStitch maintained its superior performance even when evaluated using SMINA**. These additional results further validate the robustness of our framework across different docking evaluation tools.
>
> ||parp1|jak2|braf|fa7|5ht1b|
> |-|:-:|:-:|:-:|:-:|:-:|
> ||HV|HV|HV|HV|HV|
> |REINVENT|0.522|0.476|0.503|0.417|0.514|
> |BootGen|0.534|0.498|0.518|0.425|0.530|
> |MolStitch|0.550|0.539|0.530|0.450|0.544|
>
> We sincerely thank you for bringing SMINA to our attention, and we will include these additional results along with a description in our revised manuscript.
> # Q2: What is the novelty of StitchNet, and how does it differ from genetic algorithm-based crossover operators?
> Conventional genetic algorithm-based crossover operators typically rely on hand-crafted, rule-based procedures to recombine fragments of parent molecules. Although these rules incorporate chemical intuition and domain expertise, they remain fixed throughout the search process and do not adapt based on the quality of generated outcomes. Consequently, rule-based crossover operators may explore chemical space less effectively, as they lack mechanisms to refine and favor recombination strategies that produce higher-quality molecules.
>
> In contrast, StitchNet employs a neural network architecture trained via self-supervised learning to adaptively discover effective fragment recombination strategies. **Specifically, StitchNet leverages chemical feedback—using objective scores from an offline dataset—to guide the stitching process**, ensuring that resulting molecules are chemically valid and likely to exhibit desirable properties. This self-supervised learning mechanism allows StitchNet to refine its recombination strategies based on chemical feedback, enabling it to explore promising regions of chemical space.
> # Q3: Are the gains primarily from StitchNet, the rank-based proxy, or their combination?
> Thank you for the insightful question. As noted, the combination of StitchNet and the rank-based proxy is indeed synergistic and central to the effectiveness of our framework. However, we agree that a clearer breakdown of their individual contributions is valuable.
>
> In Appendix C.6 of our manuscript, we provided a comprehensive ablation study that isolates the effects of each component. From these results, we observed that the addition of the rank-based proxy alone yields greater performance improvements compared to the addition of StitchNet alone.
>
> To clarify the ablation study experimental setup:
> * In the **rank-based proxy alone** setting, new molecules are sampled directly from the generative model (`model.sample()`) and evaluated using the rank-based proxy. The generative model is then updated through preference optimization based on this proxy feedback.
> * In the **StitchNet alone** setting, new stitched molecules are generated by StitchNet and evaluated using a score-based proxy that directly estimates their objective scores. The generative model is subsequently updated using these estimated scores as pseudo-rewards.
>
> We believe the rank-based proxy alone performs more effectively than StitchNet alone because it provides robust and reliable feedback, enabling stable and meaningful updates to the generative model. In contrast, although StitchNet alone enhances molecular diversity by producing novel stitched molecules, the score-based proxy often struggles to evaluate them accurately, leading to less reliable feedback. **In essence, the full potential of our framework is achieved when these two components are combined, effectively leveraging their complementary strengths.**
>
> [1] Alhossary et al. “Fast, accurate, and reliable molecular docking with QuickVina 2.” Bioinformatics (2015).
>
> [2] Guo et al. “Saturn: Sample-efficient generative molecular design using memory manipulation.” arXiv (2024).
>
> [3] Lee et al. “Drug Discovery with Dynamic Goal-aware Fragments.” ICML (2024).
>
> [4] Cieplinski et al. “Generative models should at least be able to design molecules that dock well: A new benchmark.” JCIM (2023).

---

### Official Review · Reviewer_B6bd · 2025-03-21

**Overall Recommendation:** 4

**Summary:**

This paper introduces MolStitch, a framework for offline molecular optimization that generates novel molecules by "stitching" fragments from an existing offline dataset, eliminating the need for iterative oracle queries. Inspired by trajectory stitching in offline reinforcement learning, MolStitch uses StitchNet to combine desirable properties from parent molecules and a rank-based proxy to evaluate molecules through pairwise comparisons. To address multi-objective optimization, the framework employs priority sampling with a Dirichlet distribution to explore diverse trade-offs along the Pareto front. Experimental results demonstrate MolStitch's effectiveness in outperforming existing methods across various offline molecular optimization benchmarks.

## update after rebuttal

The authors provided a strong rebuttal with new experiments and detailed clarifications. They showed that MolStitch benefits from Chebyshev scalarization in complex multi-objective settings, both in performance and molecular diversity. They also added comparisons to ICML 2024 baselines, including Multiple Models + COMs/RoMA and REINVENT-BO, where MolStitch remained superior. Finally, their justification of the design choices—molecular stitching, rank-based proxy, and priority sampling—was well-motivated by domain-specific insights. This is why I raised my score to accept.

**Claims And Evidence:**

Claim:  Linear scalarization with priority sampling enables effective exploration of trade-offs in multi-objective optimization.

Issue: While priority sampling with a Dirichlet distribution is effective for generating diverse weight configurations, the reliance on linear scalarization assumes a convex Pareto front. This assumption may not hold in practice, as Pareto fronts in molecular optimization can often be non-convex. The paper does not provide evidence or discussion on how this limitation affects performance in such scenarios. Including additional experiments or theoretical analysis on non-convex Pareto fronts would strengthen this claim and provide a more comprehensive understanding of the framework's capabilities.

**Essential References Not Discussed:**

Yes, the paper misses a key reference: the ICML 2024 paper "Offline Multi-Objective Optimization" by Ke Xue et al. This work is highly relevant because it introduces a wide range of baselines for offline MOO, including methods like Multiple Models + COMs, Multiple Models + RoMA, and Multi-head approaches. These baselines are essential for comparing and evaluating new methods like MolStitch. By not citing this work, the paper misses an opportunity to position MolStitch within the current state-of-the-art and demonstrate its performance against established baselines. Including this reference would provide a stronger foundation for understanding and validating the contributions of MolStitch.

**Experimental Designs Or Analyses:**

The experimental designs and analyses in the paper are sound and valid, with appropriate benchmark datasets, evaluation metrics, and comparisons to state-of-the-art methods. The ablation studies further strengthen the validity of the results.

**Methods And Evaluation Criteria:**

The proposed methods and evaluation criteria are well-aligned with the problem of offline molecular optimization. MolStitch introduces StitchNet to generate novel molecules by combining fragments from an offline dataset and employs a rank-based proxy for robust evaluation through pairwise comparisons. These methods effectively address the challenge of optimizing molecules without repeated oracle queries. The evaluation uses standard benchmarks (e.g., molecular property and docking score optimization) and metrics (e.g., hypervolume, R2.
However, one limitation is the use of linear scalarization for multi-objective optimization, which assumes a convex Pareto front—an assumption that may not always hold in practice.

**Other Comments Or Suggestions:**

No.

**Other Strengths And Weaknesses:**

Strengths:

1- Practical Relevance: The paper addresses a highly relevant real-world problem: offline molecular optimization without costly oracle evaluations. This is significant for applications like drug discovery, where wet-lab experiments are time-consuming and expensive.

2- Clarity and Presentation: The paper is well-written and clearly explains the motivation, methodology, and experimental results. The use of ablation studies and visualizations helps readers understand the contributions and effectiveness of the proposed framework.

3- Strong Empirical Results: The paper demonstrates strong performance on benchmark tasks, outperforming several state-of-the-art methods. This empirical validation strengthens the significance of the work.


Weaknesses:

1- Linear Scalarization Limitation: The reliance on linear scalarization assumes a convex Pareto front, which may not hold in many real-world scenarios. The paper does not explore alternative scalarization techniques or provide evidence of performance on non-convex Pareto fronts, limiting its generalizability.

2- Lack of Comparison with Offline MOO Baselines: The paper does not compare MolStitch with the baselines proposed in the ICML 2024 paper on offline multi-objective optimization. Including these comparisons would provide a clearer picture of how MolStitch performs relative to other state-of-the-art methods in offline MOO.

3- Limited Justification for Combination of Ideas: While the paper combines ideas from offline RL (trajectory stitching), preference optimization (rank-based proxy), and multi-objective optimization (priority sampling), it does not provide a detailed explanation of why this specific combination is the best fit for molecular applications in a multi-objective setting. A more thorough discussion of the rationale behind these choices would strengthen the paper.

**Questions For Authors:**

1- The framework uses linear scalarization, which assumes a convex Pareto front. How does MolStitch perform on problems with non-convex Pareto fronts, and have you explored alternative scalarization techniques (e.g., Chebyshev)?

2- The ICML 2024 offline MOO paper proposes several baselines (e.g., Multiple Models + COMs/RoMA). Why were these not included in your experiments, and how does MolStitch compare to them in terms of hypervolume (HV) and R2 metrics?

3- Can you explain why the combination of trajectory stitching, rank-based proxy, and priority sampling is particularly effective for molecular discovery in a multi-objective setting? Are there domain-specific insights that justify this design?

**Relation To Broader Scientific Literature:**

The paper addresses offline multi-objective optimization (MOO) in molecular discovery, building on prior work in offline single-objective optimization (SOO) and preference optimization (e.g., DPO, IPO). It introduces MolStitch, which combines trajectory stitching from offline RL and a rank-based proxy for robust evaluation. While the individual components are not entirely novel, their integration into offline MOO is timely, especially given the limited work in this area, as highlighted by the recent ICML 2024 "Offline Multi-Objective Optimization". That paper underscores the importance of offline MOO in real-world applications like molecule and protein design, proposing benchmarks to advance the field.

MolStitch’s focus on avoiding costly oracle evaluations makes it a practical and valuable tool for practitioners, even if it does not introduce groundbreaking innovations.

**Theoretical Claims:**

They look good.

---

> ### Author Rebuttal · Authors · 2025-04-01
>
> We are grateful for your thoughtful feedback. Below, we respectfully provide point-by-point responses to each of your comments.
> # Q1: Have you explored the Chebyshev scalarization technique?
> In our original study, we employed linear scalarization because it is one of the most fundamental scalarization techniques, aiming to establish the applicability of the MolStitch framework at a fundamental level.
>
> In response to your suggestion, we conducted additional experiments using MolStitch with Chebyshev scalarization in place of linear scalarization. Due to character limitations, we have uploaded the full results tables at the following link [here](https://tinyurl.com/molstitch).
>
> As shown in the table, **MolStitch w/ Chebyshev** achieved comparable performance to the linear approach in the two-objective setting. Notably, it outperformed the linear approach in the more challenging three-objective and four-objective settings.
> ||JNK3+GSK3β|JNK3+GSK3β+QED|JNK3+GSK3β+QED+SA|
> |-|:-:|:-:|:-:|
> ||HV|HV|HV|
> |w/ Linear|0.579|0.403|0.352|
> |w/ Chebyshev|0.580|0.440|0.397|
>
> We believe this is because in the two-objective setting, the Pareto front remains relatively simple, limiting the observable benefits of Chebyshev. As the number of objectives increases, the Pareto front becomes more complex and non-convex, which allows the advantages of Chebyshev scalarization to become more pronounced. Overall, these results show that MolStitch works well with both linear and advanced Chebyshev scalarization techniques.
> # Q2: Include baselines from the ICML 2024 offline MOO paper (Multiple Models + COMs/RoMA).
> Following your suggestion, we conducted additional experiments with Multiple Models + COMs/RoMA from the offline MOO paper. As shown in the table, **Multiple Models + COMs/RoMA** demonstrated better performance than their single model counterparts. This finding aligns with results throughout our paper, where ensemble proxy generally outperformed single proxy methods such as Grad.
>
> ||JNK3+GSK3β|JNK3+GSK3β+QED|JNK3+GSK3β+QED+SA|
> |-|:-:|:-:|:-:|
> ||HV|HV|HV|
> |Single Model+COMs|0.479|0.205|0.171|
> |Single Model+RoMA|0.492|0.198|0.169|
> |Multiple Models+COMs|0.489|0.211|0.190|
> |Multiple Models+RoMA|0.499|0.214|0.188|
> |REINVENT-BO|0.472|0.232|0.205|
> |MolStitch|0.579|0.403|0.352|
>
> Additionally, inspired by the strong Bayesian optimization (BO) performance in the offline MOO paper, we implemented a comparable BO method using REINVENT as the generative backbone. We refer to this as REINVENT-BO. As shown in the table, **REINVENT-BO** achieved competitive performance, highlighting its potential in this domain. However, our MolStitch framework consistently outperformed all these baselines, thereby reaffirming the superiority and efficacy of our approach.
> # Q3: Can you explain why the combination of trajectory stitching, rank-based proxy, and priority sampling is particularly effective for molecular discovery in a multi-objective setting?
> * Inspired by trajectory stitching, we propose a novel **molecular stitching** operation that recombines fragments from two parent molecules to create new stitched molecules. In molecule design, it is well established that core molecular substructures—often referred to as privileged scaffolds—play a critical role in determining key biological properties [1]. By recombining these scaffolds from diverse parents, molecular stitching creates molecules that inherit beneficial traits, enabling a broader exploration of chemical space and enhancing the discovery of novel, diverse, and high-quality candidates.
>
> * However, evaluating these newly stitched molecules presents a practical challenge: they often fall outside the distribution of the training data, making it difficult for conventional proxy models to reliably approximate their property scores. To address this, we introduce a **rank-based proxy**. Rather than regressing absolute property values, our proxy learns to predict which of two molecules is more likely to be superior with respect to the target properties. This classification-style formulation simplifies the learning task and enhances the robustness of the proxy model.
>
> * In a multi-objective setting, fixed weight configurations often struggle to balance competing objectives like potency and safety. Our **priority sampling** mechanism addresses this by generating diverse weight configurations that emphasize different objectives. This approach encourages the model to maintain a diverse population of candidate molecules that span a wide range of trade-offs. For example, in drug discovery, one scenario may prioritize potency over safety (e.g., cancer treatments), while another may require the opposite (e.g., pediatric applications). Priority sampling enables the exploration of these diverse trade-offs among multiple objectives, providing domain experts with a broader range of candidate molecules.
>
> [1] Welsch et al. “Privileged scaffolds for library design and drug discovery.” Curr Opin Chem Biol. (2010).

---

> > ### Comment · Reviewer_B6bd · 2025-04-02
> >
> > Thank you for your detailed rebuttal and for conducting additional experiments with Chebyshev scalarization. Your efforts to extend the analysis beyond linear scalarization are appreciated.
> >
> > The results suggest that Chebyshev scalarization provides an advantage in higher-dimensional objective spaces, but a more in-depth discussion of why this occurs should be included in the revised paper. You mention that the complexity of the Pareto front plays a role, but it is important to further explain how Chebyshev scalarization specifically handles non-convexity in this setting. For instance, does it lead to a more diverse or well-distributed set of solutions compared to linear scalarization? Clarifying this point would strengthen the discussion and provide better insights into when and why Chebyshev should be preferred.

---

> > > ### Author Response · Authors · 2025-04-04
> > >
> > > We sincerely appreciate for your thoughtful comments. Thanks to you, we have gained valuable insights and learned so much.
> > >
> > > # Q4: More detailed discussion of Chebyshev scalarization.
> > >
> > > To the best of our understanding, Chebyshev scalarization operates by minimizing the maximum weighted deviation from a reference point (e.g., the ideal objective values). Unlike linear scalarization, which combines all objectives into a weighted sum,  Chebyshev scalarization focuses on each objective separately and attempts to minimize the highest discrepancy among them. By targeting the worst-performing objective at each step, Chebyshev aims to balance trade-offs among objectives and can explore non-convex (concave) regions of the Pareto front—areas that linear scalarization often misses due to its inability to bend into such regions. As a result, Chebyshev scalarization tends to produce a more diverse and well-balanced set of solutions. In molecular optimization, this can lead to molecules with diverse scaffolds and balanced performance, even when objectives conflict with each other.
> > >
> > > To empirically assess the advantages of Chebyshev scalarization in the offline multi-objective molecular optimization task, we conducted an additional analysis comparing the diversity achieved by the two scalarization methods. Specifically, we measured the number of unique Bemis-Murcko (BM) scaffolds and carbon skeletons [1] among molecules on the Pareto front, comparing results from MolStitch using Chebyshev versus linear scalarization in the four-objective setting (JNK3+GSK3β+QED+SA). As shown in the table below, Chebyshev scalarization achieved higher diversity across both BM scaffold and carbon skeletons metrics, supporting the assertion that it facilitates more effective exploration of diverse regions within the search space.
> > >
> > > |                     |   BM scaffold   |  Carbon skeletons  |
> > > |---------------------|:---------------:|:------------------:|
> > > | w/ Linear           |      3453       |        1664        |
> > > | w/ Chebyshev        |      3836 |        1976 |
> > >
> > > Lastly, we would like to share our perspective on when each scalarization method might be more appropriate:
> > > * **Linear scalarization** remains effective when the Pareto front is expected to be convex (e.g., in two-objective problems with mild conflict), or when prior domain knowledge indicates that one objective should be prioritized. In these cases, linear scalarization provides a straightforward, intuitive, and interpretable approach to guide the optimization process.
> > > * **Chebyshev scalarization** is well-suited for problems involving high dimensionality or multiple conflicting objectives. In such cases, Chebyshev encourages the exploration of diverse regions within the search space and aims to produce well-balanced solutions that perform reasonably well across all objectives.
> > >
> > > We will incorporate this discussion, along with the additional diversity analysis, into the revised manuscript. Thank you once again for taking the time to review our work and for providing such thoughtful and constructive feedback.
> > >
> > > [1] Bemis GW, Murcko MA. “The properties of known drugs. 1. Molecular frameworks.” J Med Chem (1996).

---

### Decision · Program_Chairs · 2025-05-01

**Decision:**

Accept (poster)

**Comment:**

**Summary**: The paper considers the problem of offline optimization applied to the specific setting of molecular optimization where the search space is the space of molecules. The key idea of the paper is to develop a model that combines fragments from pairs of existing molecules in the offline dataset referred as “molecular stitching”. This acts like a data augmentation strategy as well. Additionally, the paper uses a learning to rank formulation for the surrogate/reward/proxy model. In order to extend the approach to the setting of multi-objective optimization, the paper considers sampling with a dirichlet distribution to explore diverse scalarizations of the objectives.

All the reviewers liked the paper and recognized the practical relevance of the problem setup and the proposed molecular stitching concept and the integration of rank-based preference learning. The authors’ response was convincing for the reviewers during the rebuttal. Overall, I am a bit skeptical of methods that have many moving parts without strong justification/unification. Such methods' performance tend to be brittle and break with small modifications in the problem setting. Having said that, **I am happy to recommend acceptance** based on unanimously supportive comments by the reviewers. I request the authors to update the final version based on discussion in the rebuttal period.